# Modeling and simulation of complex dynamic musculoskeletal architectures

Xiaotian Zhang[1], Fan Kiat Chan[1], Tejaswin Parthasarathy[1] & Mattia Gazzola [1,2]*

Natural creatures, from fish and cephalopods to snakes and birds, combine neural control, sensory feedback and compliant mechanics to effectively operate across dynamic, uncertain environments. In order to facilitate the understanding of the biophysical mechanisms at play and to streamline their potential use in engineering applications, we present here a versatile numerical approach to the simulation of musculoskeletal architectures. It relies on the assembly of heterogenous, active and passive Cosserat rods into dynamic structures that model bones, tendons, ligaments, fibers and muscle connectivity. We demonstrate its utility in a range of problems involving biological and soft robotic scenarios across scales and environments: from the engineering of millimeter-long bio-hybrid robots to the synthesis and reconstruction of complex musculoskeletal systems. The versatility of this methodology offers a framework to aid forward and inverse bioengineering designs as well as fundamental discovery in the functioning of living organisms.

[1] Department of Mechanical Science and Engineering, University of Illinois at Urbana-Champaign, Urbana, IL 61801, USA. [2] National Center for Supercomputing Applications, University of Illinois at Urbana-Champaign, Urbana, IL 61801, USA. *email: mgazzola@illinois.edu

Musculoskeletal systems consist of bones, muscles, tendons, ligaments, and other connective tissues that altogether provide function and structure to natural creatures. One of the most intriguing aspects of these architectures is the often inseparable nexus between actuation and control, topology and mechanics, due to the continuum, non-linear nature of their constitutive elements. As a consequence, and in stark contrast with rigid-body robots, soft creatures can harness a wide range of deformations and structural instabilities to effectively cope with complex, unstructured and dynamic environments[1]. Thus, biological musculoskeletal architectures, due to their intrinsic distributedness, softness and compliance, exhibit the ability to outsource control tasks to their embodiments, an emerging paradigm denoted as morphological computation or mechanical intelligence[1–3].

These considerations have prompted a number of soft robotic investigations in which artificial compliant materials and highly stretchable and shearable elastomeric structures are used in a variety of applications from gripping, grasping, manipulation[4–8] and artificial muscles[9], to a range of robotic creatures[10–19]. More recently, a radically new breed of soft bio-hybrid robots[3,20–24] that combine biological muscles and sensors with artificial scaffolds has been emerging, paving the way to engineer living machines with unique abilities of self-assembly, healing, growth and adaptivity. This technology carries the promise of high impact applications, from biomedicine to manufacturing[25], and of fundamental discovery as it provides a platform to test hypotheses related to the functioning of living organisms[26].

Despite these experimental advances, the modeling and simulation of dynamic musculoskeletal architectures (either biological, artificial or bio-hybrid) has not proceeded at the same pace[27], impairing the broad deployment of soft robotic technology. Biological layouts have been traditionally modeled as mechanical structures composed of springs, dampers and linkages, formulating joint motions into rigid-body dynamic equations[28,29]. Although insightful in many contexts[30,31], this approach is ill-suited to fully capture the dynamics of intrinsically soft-bodied systems such as cephalopods, fish or snakes. On the other side of the spectrum, high-fidelity 3D simulations based on the finite element method (FEM) have been used to model muscles as viscoelastic continuous materials[32,33] and to design soft robotic components[8]. Nonetheless, these methods also exhibit limitations: often prohibitive computational costs, numerical instabilities and loss of accuracy due to distortion of discretization elements, ad-hoc (re-)meshing, and an involved mathematical formulation. As a consequence, FEM have so far been impractical for the simulation of complete musculoskeletal structures and their interaction with the environment.

An attempt to fill the space between rigid-body and FEM models is represented by the 3D lattices of masses and Euler beams of Hiller et al.[34], which represent a balanced compromise between accuracy, robustness, and computational costs, although specialized to monolithic soft bodies. The graphics community has also been active in this space[35–37]. Grinspun and colleagues introduced a popular discrete elastic rods method[38] for the simulation of elastic yet unshearable and unstretchable filaments, and considered their assembly into dense, entangled masses[39]. Pai and colleagues investigated combining together spline-based muscle-strands for the simulation of various human body parts such as hands or ocular muscles[35,36,40–42]. Although numerically efficient, these approaches are specialized to scenarios in which shear, stretch, and/or twist and dynamic effects are unimportant. In general, instead, it is not possible to rule out a priori any particular mode of deformation, especially when complex and compliant architectures interact with unstructured and dynamic environments. Hence the need for efficient, robust and more

broadly applicable solvers. Thus, building on the above methods, we have developed an approach based on assemblies of Cosserat rods[43]. These rods are slender, elastic, and soft filaments that can undergo all modes of deformation and whose dynamics in 3D space can be accurately represented via a one-dimensional mathematical description. Active and passive rods characterized by different material properties are then combined together through appropriate boundary conditions, into dynamic layouts that model the connectivity between bones, tendons, muscles as well as artificial scaffolds interacting with the environment (contact, friction, fluids). The Cosserat model and its (far more popular) unstretchable, unshearable counterpart, the Kirchhoff model[44], spurred by the work of Grinspun and colleagues[38], have led to a number of graphics applications involving elastic ribbons[45,46], woven cloth[47,48], entangled hair and fibers[39,46,49], wire mesh[50], and viscous threads[51]. Moreover, these models found application in physics, biology, and engineering to characterize polymers and DNA[52,53], flagella[26,54], tendrils[55], cables in automotive design[56], and soft robot arms[57]. Nevertheless, almost all studies consider individual filaments, a few consider multiple rods that are generally passive and homogeneous (hair, cloth) and none of them consider dynamic, heterogeneous living architectures capable of undergoing all modes of deformation.

Here, we demonstrate a methodology in which the full dynamics of all these deformation modes (bend, twist, shear, and stretch) is accounted for. We build on our previous work on Cosserat rods and establish a musculoskeletal modeling approach to represent and realistically simulate active, heterogenous biological layouts. We thus demonstrate the utility of our approach to engineer, synthesize and replicate living body architectures through: (1) Engineering of bio-hybrid robots in which we illustrate and exploit the predictive capability of our approach to guide the design and fabrication of bio-hybrid robots, mitigating the need for impeding trial-and-error methods; (2) Synthesis of complex biological systems in which we couple our solver with evolutionary optimization techniques to understand how muscular architectures lead to smooth slithering gaits. This study illustrates how biological layers of complexity can be stripped away to unveil broadly applicable design principles, thus advancing biomimetic applications; (3) Replication of full-scale biological systems in which we employ our solver to reconstruct and actuate feathered wings. This exemplifies our ability to replicate biological systems, mimicking the underlying biomechanics, thus providing accessible ways to study and understand the biophysical functioning of natural creatures in silico.

Overall, this study advances the argument that rod models have a valuable role to play in the modeling of complex active systems, further expanding their range of application in robotics and biology.

## Results

**The human elbow joint**. We first consider the human elbow joint comprised of muscles, tendons and bones (Fig. 1a) to illustrate how rod assemblies are mapped to physiology, dynamics, and morphology. In contrast to a fully compliant system, the elbow joint exhibits both soft and stiff characteristics as well as simplified dynamics and reduced configuration space. Nonetheless, its analysis allows us to verify and calibrate our model against a wealth of readily available data (anatomical and biomechanical), and to relate our description to the widely used Hill model[30] (see Supplementary Note 1). It also serves the purpose of illustrating our representation's level of detail, which can be leveraged to address human patient-specific kinesiological needs.

We reproduce the biceps brachii (Fig. 1b, orange elements) in silico, each head modeled as a bundle of 18 viscoelastic Cosserat

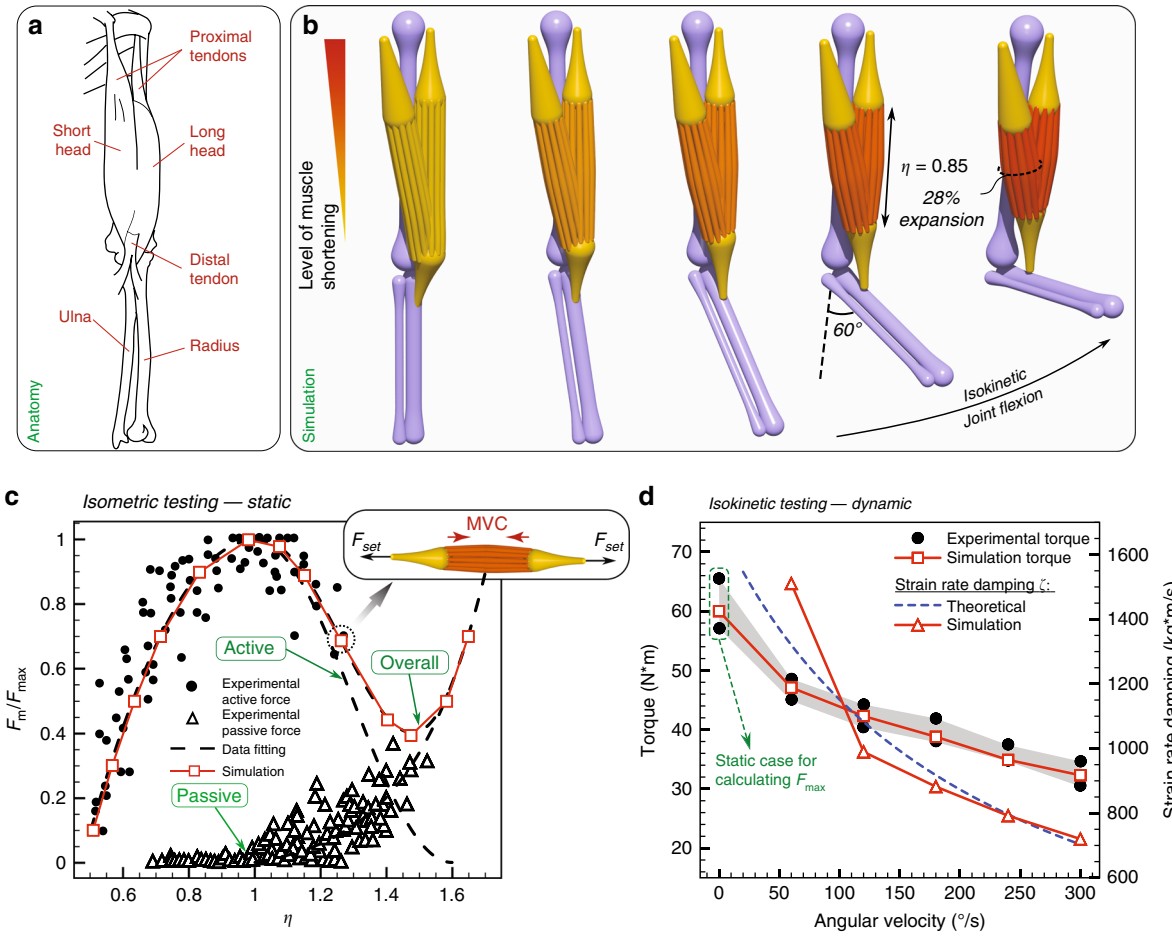

**Fig. 1** Human elbow actuation. **a** Elbow anatomy. **b** Simulation of an elbow composed of three bones (humerus, ulna and radius) and two heads of biceps (short and long head) performing a complete flexion. **c** Experimental data[60] and simulations for active and passive force normalized with peak force ($F_m/F_{max}$) during the isometric exercise ($F_{set}$ mimics the resistance encountered by the muscle and results in its equilibrium length $\eta$). **d** Experimental[61] and simulation torque measurements of the elbow joint (angled at 60°) performing maximal isokinetic concentric flexions at different angular velocities along with the corresponding overall muscle strain rate damping $\zeta$. The numerically determined $\zeta$ (see Supplementary Note 2) are then compared with theoretical estimates based on the Hill model[62]

rods, each rod representing 20 motor units, for a total of 360, in agreement with average physiological measurements[58]. We note that the number of motor units per rod can be varied depending on the desired level of granularity. These rods, whose governing equations can be found in the Methods section, can actively contract and relax. Their characteristic force outputs and twitching times can be directly related to their cross-sectional area through a contractile stress ($\sigma_m$). This allows the muscle to comply with the size principle[59] which relates low-force, slow-twitch activity to smaller motor units, and high-force, fast-twitch activity to larger ones (see Supplementary Note 1). Towards the completion of a full elbow assembly, we consider the humerus, ulna and radius (all of which are slender bones), represented as passive, stiff (effectively rigid) rods with tapering segmental radii (Fig. 1b, purple elements). Similarly, proximal and distal tendons are modeled as tapered passive, but this time elastic, rods (Fig. 1b, yellow elements). We note that bones are not always slender, in which case a mixed representation that includes rigid bodies or FEM should be employed.

The final assembly and its configuration space are achieved by specifying boundary conditions and connectivity among the various elements. Spherical joints (free relative rotations) are used for the muscle–tendon, bone–tendon, and upper arm–shoulder connections. A hinge joint (relative rotations in a prescribed plane) is implemented for the humerus–radius connection, while

a fixed joint (no relative motion) is used for the ulna–radius connection. For the mimicking of pure flexion–extension exercises (as intended here), we do not consider the relative rotation between ulna and radius which occurs during pronation–supination. However, these movements can be modeled by redefining the joint connection to allow for rotations in two perpendicular planes. The full elbow assembly is depicted in Fig. 1b. Details on the modeling and boundary conditions can be found in the Methods section, and biomechanical properties of constitutive elements are summarized in Supplementary Table 1.

We then performed isometric (static) and isokinetic (dynamic) tests for validation against experiments[60,61]. An isometric test is conducted with the biceps muscle performing a maximum voluntary contraction (MVC) against a non-moving handle, so that the elbow joint movement is restricted and the muscle length $\eta$ of the biceps is kept constant. By repeating this exercise for different handle positions, the static force output is mapped to different muscle lengths (Fig. 1c). To perform the test in silico, we use available experimental data (Fig. 1c) to compute polynomial fittings that dictate the muscle active MVC and passive elastic response (Fig. 1c) as functions of its elongation $\eta$ (see Supplementary Note 2). Once these biomechanical properties are determined, we let the muscle (initialized at rest length $\eta = 1$) perform its MVC while applying prescribed external forces $F_{set}$ at its ends. The simulation then dynamically evolves the muscle to

its static equilibrium length $\eta$. By repeating this experiment for various $F_{set}$, we can relate muscle length to static force output (Fig. 1c), confirming a good match between simulations and experiments.

Isokinetic tests instead measure the dynamic torque output of the muscle performing MVC against a handle moving at constant speed (joint flexion in Fig. 1b). When the muscle performs MVC during length-changing actuation, its viscosity introduces damping effects which decrease the static force output. We take into account these effects through a damping coefficient $\zeta$, which is numerically set to match simulated and experimental torque outputs (Fig. 1d). The resulting $\zeta$ was then compared with theoretical estimates[62] (Supplementary Eq. 5) and found to be in reasonable agreement (Fig. 1d).

Our simulations also capture morphological deformations as the joint flexes. Indeed, during contraction the motor units shorten and, due to muscle incompressibility[63], the bicep radius increases (Fig. 1b). Our model accounts for incompressibility (Poisson ratio $\nu = 0.5$) through the local dilatation factor $e$ of Eqs. (3) and (4) (Methods section—mathematical derivation in Gazzola et al.[64]), and prevents rods' interpenetration by checking for collisions among them (Eq. (7)). We measure ~28% increase in biceps cross-sectional area when the elbow forms a 90° angle, consistent with ~30–34% observed experimentally[65].

In summary, we virtually reconstructed a 3D replica of a human elbow joint and, by taking advantage of isometric and isokinetic tests, modeled, calibrated, and validated individual muscle unit's actuation so as to reproduce the dynamic and morphological behavior of this system. In general, this modeling approach presents several advantages relative to the commonly used Hill model: (a) Individual rods (motor units of different sizes) can be selectively recruited or rendered passive (mimicking an injury). As an example, in Supplementary Note 6.2, we propose an assisting device (inspired by the coiled fishing lines of Haines et al.[66]), which converts internal twist into contraction forces to aid restoring the weightlifting abilities of an injured biceps. (b) Compliant muscles can bend, twist and shear to respond realistically to the dynamics of the entire structure and the environment. Indeed, the investigations presented in Supplementary Note 6 find that neglecting twist or shear (disregarded in Kirchhoff or strand models[35,36,40–42]) can have a significant quantitative and qualitative impact, especially when the environment produces three-dimensional, fluctuating and impulsive loads.

**Engineering of bio-hybrid robots**. Next, we employ our solver to guide the design and fabrication of swimming and walking millimeter-long bio-hybrid bots.

For the investigation of swimming bio-hybrid robots, we first solve a forward problem by numerically modeling and simulating the bio-hybrid flagella of Williams et al.[23], the first instantiation of a functionalized PDMS (polydimethylsiloxane) coupled with cultured cardiomyocytes beating in viscous fluids. We create a one-to-one computational replica of the original swimmer of length $L = 1927$ mm: the PDMS substrate is modeled as one passive filament replicating the experimental geometry and material properties[23], while the living component—the cluster of cultured cells—is represented as a small, soft contractile filament connected to the substrate. The swimmer operates in a flow regime characterized by a small Reynolds number ($Re \sim 10^{-2}$) so that hydrodynamic loads can be captured via slender body theory[67]. System details are reported in Fig. 2, and Supplementary Note 3.

As observed in Fig. 2, we obtain good qualitative and quantitative match between simulations and experiments in both the assessment of swimming motion (Fig. 2a) and forward displacement of the bot's center of mass (Fig. 2b).

With a working model in hand, we then tackle the inverse problem of optimizing the bot layout to maximize its swimming speed. In order to identify the optimal design, we couple our solver with the Covariance Matrix Adaptation-Evolution Strategy algorithm (CMA-ES, Hansen et al.[68]). The CMA-ES is a stochastic optimization algorithm that progressively samples generations of parameter vectors (population of bots characterized by different layouts) from a multivariate Gaussian distribution $\mathcal{N}$. While there is no mathematical proof of convergence to global optimum, CMA-ES has proven reliable in dealing with multi-modal, low-dimensional continuous problems[69,70] and has been employed in a range of engineering and biophysical applications[71–74].

We thus let CMA-ES evolve four key parameters that characterize the bot layout—head length, head radius, tail radius and cell location—within prescribed ranges accounting for actual manufacturability. The bot length remains fixed.

The optimization course illustrated in Fig. 2c converges to an optimal solution that improves the original swimmer's maximum speed by a factor of 2.44. The optimal design requires a shorter but wider head, muscle cells attached closer to the head and a ~38% thinner tail (exact parameters reported in Fig. 2). We observe that the longer (due to shorter head) and thinner (thus, more compliant) tail in the optimal design enables larger bending deflection (Fig. 2d), leading to larger forward thrust that accelerates the swimmer, while the optimized head contributes in balancing the imparted angular momentum. We note that the optimizer did not select the lower bound of attainable tail radius, which suggests that balance between flexibility and drag associated with large tail deflection is needed for optimal performance. This approach thus lays the foundations for the next generation of bio-hybrid swimming robot design[24].

In addition to modeling and optimizing a bio-hybrid swimmer, we also tackled the computational design of a bio-hybrid walker, leading to the fabrication and testing of the largest and fastest motile biological machine (biobot) to date[21]. Inheriting the bio-hybrid robot design from a previous demonstration[22], the walker of Pagan-Diaz et al.[21] consists of an asymmetric hydrogel scaffold and skeletal muscle tissues, resembling muscle–tendon–bone relationships found in vivo. The walker operates in a solution bath in which the muscles are suspended and electrically shocked to induce contractions that result in motion due to asymmetry and friction. We modeled this architecture and, targeting a bot length of 14 mm, which is approximately twice the previous largest attempt, used our simulations to design the new scaffold and topological muscle arrangement of the bot. Critical to the design is a new muscle–tissue topology (represented via rod assembly, Fig. 2e) in which a thin strip section connects two rings wrapped around the skeleton legs to transfer muscle contraction forces. The muscle contractile stress ($\sigma_m$) was characterized through benchmark experiments[21], and implemented in simulation as the absolute value of a sinusoidal signal with amplitude $\sigma_m$. Thus, the maximum force $F_m$ that a muscle tissue with cross-sectional area $A_m$ and Young's modulus $E_m$ can exert on the skeleton is expressed as

$$F_m = A_m \left( \gamma \sigma_m - \frac{E_m \epsilon}{1 - \epsilon} \right), \tag{1}$$

where $\gamma = A_{act}/A_m$ is the ratio of active-to-total muscle cross section area (determined in Pagan-Diaz et al.[21]) and $\epsilon$ denotes strain. The second term on the right-hand side captures the elastic response of the deforming filament and its cross-sectional rescaling as it shortens or elongates. Taking into account hydrogel

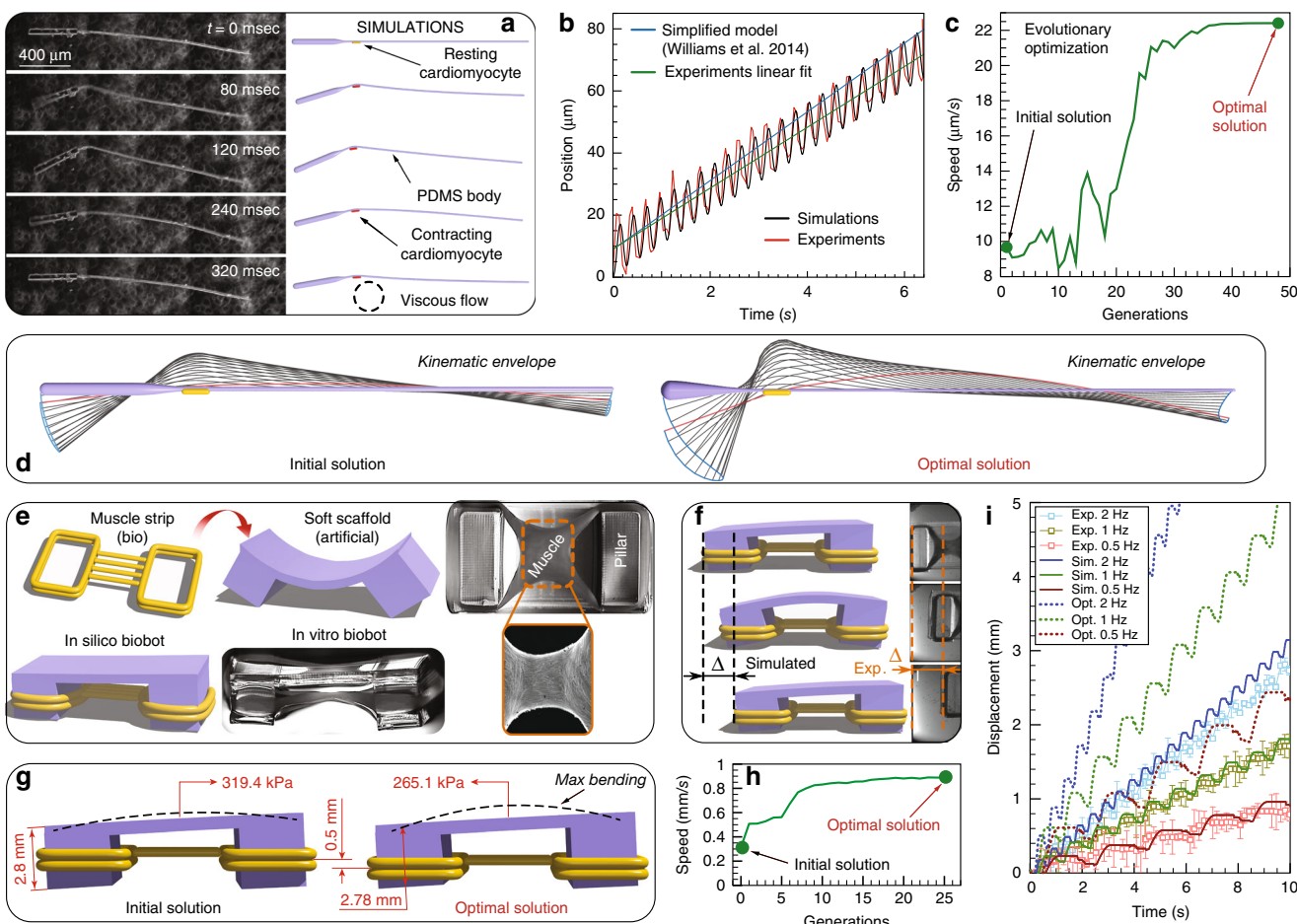

**Fig. 2 Bio-hybrid robotic design. a** One-to-one comparison of the robot with experimental photographs[23] at different stages within one swimming cycle. **b** Axial position of the robot's center monitored over more than 20 cycles compared with experimental data. **c** Optimization course: Convergence to optimal solution is observed after 48 generations. Optimization was constrained with head length within $[0, 1.927]$ mm, head radius within $[4, 40]$ μm, tail radius within $[4, 6.5]$ μm and cell location at any point along the tail. The bot longitudinal dimension is fixed at $L = 1.927$ mm, so that the tail length can be inferred from the head length. The parameter ranges are determined to account for actual manufacturability. **d** Visualization of both the original and the optimal designs showing configurations at rest and midline kinematic envelopes over one period. Original swimmer details: substrate is modeled with 424-μm-long head and 1503-μm-long tail with radii 20 μm and 7 μm respectively. Density $\rho = 0.965$ gcm$^{-3}$, Young's modulus $E = 3.86$ MPa, bending stiffness $EI = 2.427 \times 10^{-9}$ Nmm$^2$, mass $m = 7.364 \times 10^{-7}$ g are set according to Williams et al.[23]. The contractile cell is modeled with radius 10 μm and length 100 μm. The cell is set to produce a contracting force $F = \sigma_m A = 12$ μN with beating frequency $f = 3.6$ Hz [23]. The fluid has dynamic viscosity $\mu = 1.2 \times 10^{-3}$ Pa · s. Optimized swimmer details: substrate has length and radius of 190 μm and 32.3 μm, respectively. The contractile cell is attached 190 μm away from the head and the tail is 4.3 μm thick. **e** Overall design of the walker with yellow elements representing muscle rings and purple elements representing the skeleton. Experimental images adapted from[21]. **f** Simulations versus experiments: Bot displacement over 2 seconds for the actuation frequency 1 Hz. **g** Visualization of initial and optimized design of the walker. **h** Optimization course converges after 25 generations. Optimization was constrained with skeleton's Young's modulus [250–350] kPa, length of the shorter pillar [2.4–3.4] mm and location of muscle strip [0.5–3] mm (distance from ground), all of which are chosen according to manufacturability constraints. **i** Dynamic behavior of simulated (solid lines) and experimental (markers with error bars) walker with muscle contraction at different frequencies, and comparison of walking performance between initial (solid lines) and further optimized design (dashed lines)

properties (Young's modulus $E$) and 3D printing capabilities, we redesigned the target robot layout by varying material stiffnesses, leg lengths, muscle topology, and geometry until a faster design was obtained. This computational blueprint was then experimentally tested and demonstrated in Pagan-Diaz et al.[21] (Fig. 2f). As computationally predicted, the bot is capable of walking under different muscle stimulation frequencies (Fig. 2i), with the maximum velocity being twice that of the previously reported design[22].

Here, we challenge our computational framework to further improve the walker of Pagan-Diaz et al.[21] by optimizing it for speed, a non-trivial task given the non-linear interplay between asymmetric friction, bending stiffness of the skeleton and length

ratio of the two pillars. Thus, after fixing the overall length and width of the walker and the muscle contraction frequency at 2 Hz, we identify three critical parameters: skeleton's Young's modulus, length of the shorter pillar and location of the muscle strip. As can be seen in Fig. 2i, the newly identified optimal solution (Fig. 2g) locomotes at double (~250%) the speed of Pagan-Diaz et al.[21], across stimulation frequencies (2 Hz, 1 Hz, and 0.5 Hz), thanks to a softer connecting bridge coupled with slightly more asymmetric legs (Fig. 2g).

We have shown through these studies that our computational approach is able to capture the physics of cell- and muscle-powered soft robotic systems and further optimize their design for desired performance. This also illustrates how the robustness

and versatility of our solver—coupled with inverse design techniques—can be harnessed to engineer more capable prototypes.

**Synthesis of slithering snakes.** Here we employ our numerical approach to distill design principles and extract broadly applicable architectural motifs from complex biological systems (in this case, a snake with its intricate musculature layout), in favor of engineering manufacturability and biomechanical understanding.

Extensive work has been done on understanding snake locomotion[12,29,75], targeting robotic replicas made of rigid linked elements actuated with servomotors[76,77]. Here we illustrate the viability of a completely soft elastic snake, modeled and computationally designed inspired by real snakes, but effectively actuated via a small number of muscle–tendon groups to achieve smooth undulatory movement. Snakes exhibit a complex architecture made of hundreds of overlapping homologous lateral muscle segments, each spanning across multiple vertebrae (Fig. 3a). Although snakes are equipped with a multitude of muscles to orchestrate a variety of gaits and body deformations, we speculate that only a few and, importantly, overlapping actuators are necessary for effective and smooth forward slithering. We test this hypothesis by considering a simplified snake architecture made of a small number of symmetric and antagonistic lateral muscle–tendon pairs. Then, we let CMA-ES identify locations and actuation patterns, so as to maximize the snake's forward speed. This way architectural motifs are free to emerge, and their performance can be compared with reference simulations[64,78,79] and experimental recordings[75,80]. While previous reference studies were able to realistically replicate various gaits via continuously actuated elastic beams[64,78,79], we emphasize that our goal here is to reveal hidden architectural design principles and expose their function for engineering purposes. This is achieved here via a generic, species-agnostic approach, rather than dissecting in detail the functioning of any specific snake architecture (Supplementary Note 4).

Then, our limbless, soft robot is constituted by a tapered, elastic skeleton modeled as a filament, and to retain the biological analogy, we measure the length of the snake in terms of vertebrae, from 0 (head) to 100 (tail). In our simulation, the three major lateral muscle–tendon groups responsible for locomotion (semi-spinalis-spinalis (SSP-SP), longissimus dorsi (LD) and iliocostalis (IC)) are lumped into a single group—one muscle bundle that intervenes between two tendons (Fig. 3a). Two joints anchor the extrema of these longitudinal actuators along the snake's body at half its radius away from the midline vertebra. While this simplifies the snake's overall architecture, it retains its fundamental components and allows us to test whether overlapping muscle layouts naturally emerge as favorable solutions. Finally, muscles and tendons are 'glued' to the body, hence complying with the same local curvature in response to the full body dynamics. The interaction between the snake and the ground is rendered via anisotropic friction by adopting the model of Gazzola et al.[64] and the experimental friction coefficients and Froude number $Fr$ (ratio between inertial and friction forces) of refs. [75,80].

Given the above musculoskeletal model, we seek to determine the minimal number of actuators, their layout and activation patterns so as to closely reproduce reference simulations[64,78,79] and experimental recordings[75,80]. The body layout is then parametrized so that each antagonistic muscle–tendon group $i = 1, \ldots, n$ (we separately considered the six scenarios $1 \leq n \leq 6$) can arbitrarily span any number of vertebrae between 5 and 95 ($x^i_m$ denotes the starting location and $L^i_m$ the overall length

$0 \leq L^i_m \leq 90$), exert a peak force of magnitude $0 \leq F^i_m \leq 3500\,N$ (corresponding to a local torque between 0 and $\sim 40\,\mathrm{Nm}$, consistent with the range in Gazzola et al.[64]), and is rhythmically activated by the periodic function based on Hu et al.[75] and Gazzola et al.[64]

$$\text{Activity}\,(t) = \begin{cases} F^i_m \cdot \left(0.5 \sin\left[2\pi f\left(t - \phi^i_m\right)\right] + 0.5\right), & t > \phi^i_m \quad \text{right} \\ F^i_m \cdot \left(-0.5 \sin\left[2\pi f\left(t - \phi^i_m\right)\right] + 0.5\right), & t > \phi^i_m \quad \text{left} \end{cases}$$

(2)

where $0 \leq \phi^i_m \leq 1$ is the phase shift, and the activation frequency is set to be constant so as to attain $Fr = 0.1$[64,75]. Therefore, our fully compliant, active structure is characterized through the parameters $x^i_m, L^i_m, F^i_m, \phi^i_m$. The optimal set of parameters that maximizes the average forward speed over one actuation period is again identified via CMA-ES. This set-up allows us to make meaningful comparisons against previous studies[64] which employed the same optimization technique.

By considering snakes with increasing number of muscle pairs ($1 \leq n \leq 6$), separately optimized for speed, we show in Supplementary Note 4 that as few as four soft longitudinal actuators can closely approximate the idealized continuous reference[64], which sets the attainable velocity upper bound. The 37-generation optimization course of this four-muscle architecture is reported in Fig. 3b and illustrates how the average velocity converges to a maximum value that coincides with the upper bound. Thus, a snake bearing merely four muscle groups is shown to perform comparably to the continuous actuation model. The identified design exhibits muscle groups that span roughly 30–40 vertebrae (Fig. 3d). This is in reasonable agreement with biological observations[81] where the snake's major epaxial muscle segments in total span $\sim 27$ vertebrae (Fig. 3a, adapted from Jayne[81]). The phase differences between the muscle groups are depicted in Fig. 3c. Moreover, actuators' overlap (Fig. 3d) is consistently identified as a key feature independent of the number of muscle pairs considered (Supplementary Note 4). Indeed, non-overlapping architectures were systematically discarded by CMA-ES as sub-optimal (up to 60% speed degradation).

Comparing the dynamic behavior of the Gazzola et al.[64] with our identified model, we observe that despite having approximately equal mean forward velocities ($\sim 0.58\,\mathrm{ms}^{-1}$), our musculoskeletal representation exhibits larger oscillation in forward and lateral velocities (Fig. 3f). This stems from the limited number of muscles, and is reflected in the more prominent lateral displacement of the midline kinematics (Fig. 3e). For comparison, we additionally report experimentally recorded midline gaits of a corn snake, the fastest recorded in Hu and Shelley[80] among various species characterized by $Fr \sim 0.1$. The observed gait is found to closely resemble our models (Fig. 3e). It is then remarkable how the careful orchestration of distributed actuation (four longitudinal muscle groups) allows for smooth realistic gaits despite its simplicity. This is in stark contrast with a rigid snake robot counterpart equipped with only four servomotors that would otherwise exhibit a less refined and less smooth motion.

This study illustrates a framework to simplify, test and distill biomechanical principles out of complex biological systems as shown by a fully compliant, realistically slithering and fast snake made of a few simple actuators. Thus, by solving an inverse problem, the musculoskeletal layout for a potential soft robotic snake is identified, guiding its practical design and manufacturing. This is shown to approximate the idealized continuous actuation case, highlighting the role of a natural solution based on overlapping longitudinal actuators.

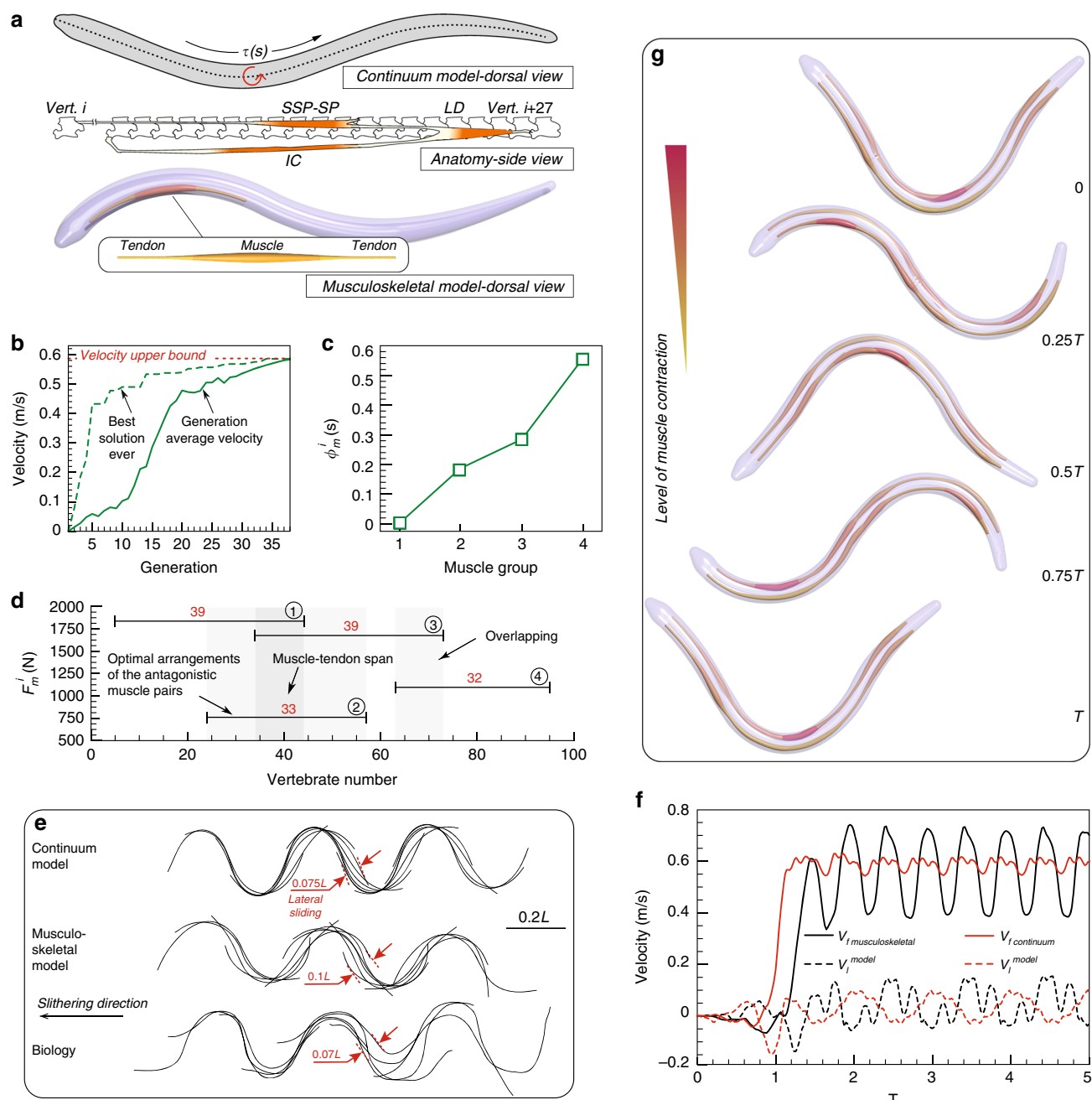

**Fig. 3** Emergent snake muscular architecture. **a** (Top) Continuum modeling of a snake with continuous torque profile along a uniform slender body[64]. (Middle) Sketch of the snake lateral muscle anatomy highlighting the epaxial muscle segment comprised of multiple muscles and tendons (adapted and modified with permission from Jayne[81]). (Bottom) Our simplified muscular snake model consists of a compliant continuum body and antagonistic muscle segments intervened between tendons. **b** The optimization course of the forward velocity as the snake's musculoskeletal structure and activation patterns are evolved by the optimizer. Average velocity (within one generation) converges to the velocity upper bound provided by the model of Gazzola et al.[64]. **c** Identified optimal phase difference $\phi^i_m$ between muscle groups. **d** Identified optimal muscle arrangement: muscle groups' span (detailed exploded view provided in Supplementary Fig. 3) and corresponding peak actuation forces $F^i_m$. **e** Comparison between the fastest gait observed in the continuous reference[64], our musculoskeletal model and experimental recordings of fast snakes characterized by similar Froude number[80] (Scale bar, 0.2 L). **f** Forward and lateral velocity of continuous torque and musculoskeletal models. **g** Undulatory motion of snake slithering over one cycle, illustrating the level of contraction of each muscle at different phases over one actuation period. Settings: snake skeleton is modeled with length $L_s = 1$ m, maximum radius of $R_s = 25$ mm, density $\rho = 1000$ kg m$^{-3}$ and Young's modulus $E = 10$ MPa (silicone rubber of intermediate stiffness)

**Replication of feathered wings**. So far we have studied musculoskeletal layouts that are assembled from two key components: the power source (muscles, cells) and the substrate (bones, elastic bodies), entailing $\mathcal{O}(10)$ rods generating locomotive functions on surfaces or in bulk liquids. In this section, we demonstrate an instance of locomotive strategy that incorporates additional biological structures with critical functionalities—a feathered musculoskeletal bird wing—scaling up our representation to $\mathcal{O}(10^3)$ rods. The study case here serves as an illustration of our solver's ability to qualitatively replicate full-scale biological systems while capturing main traits of the underlying biophysical behavior, thereby providing an accessible tool to understand them in silico.

Numerous studies have been conducted to understand the different biophysical aspects of bird flight, from muscular activation patterns for different flight modes[82] to geometrical and mechanical properties of feathers[83], in relation to thrust generation, drag reduction and sound suppression[84]. Motivated by these investigations, we consider the dynamics of the wing structure of a pigeon (Columba livia). We reconstruct in silico remiges feathers and model the rachis as filaments with bending stiffness $EI$ consistent with[85]. Depending on the length of the feather, ~80−200 barbs are attached to one rachis (Fig. 4a). Each computational barb represents approximately five real ones, with its radius set to match the estimated aggregate bending stiffness[86] (see Supplementary Note 5.1). Overall, 19 feathers are connected to the wing such that the total wing area conforms to biological data[87]. Our computational model then entails a total of ~3000 rods per wing. We consider the four muscles associated with the shoulder and elbow joints to control wing actuation and morphing (Fig. 4b), with biomechanical parameters adapted from the human elbow joint, due to lack of specific measurements. In our four-muscle model, the supracoracoideus–pectoralis pair controls the dorsoventral angle of the shoulder and the biceps–scapulotriceps pair controls the elbow angle during flexion–extension. The temporal evolution of anteroposterior angle then arises from the dynamic interaction between the structure and the environment. Aerodynamic loads are estimated via a reduced order model in which forces scale quadratically with the local body velocity (see Supplementary Note 5.2). While this model cannot capture the complex unsteady aerodynamics associated with flapping flight, it nonetheless provides a preliminary estimate. We underscore the qualitative character of this specific demonstration.

We then set to reproduce the kinematics of wings morphing through a full stroke cycle during takeoff mode. We first initialize our simulated wing in a straight, flat configuration (Fig. 4d) and over the initiation phase, set (arbitrarily) the muscle activation via Supplementary Eq. 6 so as to prepare the wing for the downstroke phase (Fig. 4e, f). During the downstroke and upstroke phase, the muscle actuation patterns (Supplementary Eqs. 7–10) are instead based on experimentally recorded electromyography (EMG) signals[82] (Fig. 4f). Since EMG measurements do not provide the magnitude at which the muscles operate (only their time sequences), we set the muscle actuation force (~10 N, same order of magnitude as in Biewener et al.[88], Supplementary Note 7.4). As can be seen in Fig. 4e, our model captures the temporal evolution of the three joint angles, in qualitative agreement with experimental measurements[82]. This is a non-trivial task given the highly non-linear interplay between muscle actuation, passive structural dynamics and aerodynamic loads. The main discrepancy is observed for the anteroposterior joint angle. This is not surprising since the four muscles have little control on it and its time evolution emerges as a result of the overall system dynamics, rendering it most sensitive to modeling approximations. In this context, it is still notable that despite all the approximations that our model necessarily entails, simulations can qualitatively capture the overall wing behavior, with a maximum joint angle deviation from experiments of ~10°, comparable with measurement variations.

Thus, here we have demonstrated the potential of our method in representing complex, heterogenous biological structures to a high degree of detail for the investigation of locomotive functions.

### Importance of all modes of deformations.
Finally, we note that while the role of different deformation modes may be predicted a priori for simple problems, their significance in more complex heterogenous architectures interacting with uncertain environments may present a challenge. In this light, and to further advance the argument for the need to capture all deformation modes, we extend our investigation to understand in particular the impact of twist and shear (often assumed unimportant) on the architectures presented in this study through numerical twist- and shear-hardening experiments. Findings can be found in Supplementary Note 6, where interaction with the environment (particularly friction) is observed to excite these modes, thereby affecting system response. Additionally, two demonstrations, which functionality critically relies on twist (elbow assistive device) and shear (slithering on uneven terrain) modes, are introduced to underscore the opportunity of modeling these effects.

## Discussion
We have presented a methodology based on the assembly of heterogeneous, active and passive Cosserat rods for the simulation of dynamic musculoskeletal architectures that can undergo all modes of deformation. This approach aims at addressing the lack of rigorous engineering methods in soft robotics and contributes to fill the gap between conventional rigid-body modeling and high-fidelity FEM methods, striking a compromise between versatility, accuracy, robustness, numerical and computational complexity. These favorable features are leveraged for engineering, synthesis and replication of soft-bodied systems, demonstrated here in a number of forward and inverse problems relative to soft robotic and complex biological structures across scales (from ~100 μm to m) and environments (aquatic, terrestrial, and aerial locomotion). Our results illustrate the utility of our approach, establishing it as a promising asset in a broad range of applications, from bioengineering to customized rehabilitation, as well as fundamental discovery in the functioning of living organisms.

## Methods
**The Cosserat rod model.** In the context of composite and soft bodies characterized by large deformations in 3D space, non-linear mechanics, continuous actuation, sensory feedback and interface effects, we propose a dynamic model based on assemblies of Cosserat rods[89]. In this section, we recall the mathematical basis and numerical methods for the simulation of individual elastic, stretchable and shearable filaments[64], and the extensions implemented here to handle complex biological layouts.

**Individual rods.** We mathematically describe a rod (slender body, Fig. 5a) by a centerline $\bar{\mathbf{x}}(s,t) \in \mathbb{R}^3$ and a rotation matrix $\mathbf{Q}(s,t) = \{\bar{\mathbf{d}}_1, \bar{\mathbf{d}}_2, \bar{\mathbf{d}}_3\}^{-1}$ which leads to a general relation between frames for any vector $\mathbf{v}$: $\mathbf{v} = \mathbf{Q}\bar{\mathbf{v}}$, $\bar{\mathbf{v}} = \mathbf{Q}^T\mathbf{v}$, where $\bar{\mathbf{v}}$ denotes a vector in the lab frame and $\mathbf{v}$ is a vector in the local frame. Here $s \in [0, L_0]$ is the material coordinate of a rod of rest length $L_0$, $L$ denotes the deformed filament length and $t$ is time. If the rod is unsheared, $\bar{\mathbf{d}}_3$ points along the centerline tangent $\partial_s \bar{\mathbf{x}} = \bar{\mathbf{x}}_s$ while $\bar{\mathbf{d}}_1$ and $\bar{\mathbf{d}}_2$ span the normal–binormal plane. Shearing and extension shift $\bar{\mathbf{d}}_3$ away from $\bar{\mathbf{x}}_s$, which can be quantified with the shear vector $\boldsymbol{\sigma} = \mathbf{Q}(\bar{\mathbf{x}}_s - \bar{\mathbf{d}}_3) = \mathbf{Q}\bar{\mathbf{x}}_s - \mathbf{d}_3$ in the local frame. The curvature vector $\boldsymbol{\kappa}$ encodes $\mathbf{Q}$'s rotation rate along the material coordinate $\partial_s \mathbf{d}_j = \boldsymbol{\kappa} \times \mathbf{d}_j$, while the angular velocity $\boldsymbol{\omega}$ is defined by $\partial_t \mathbf{d}_j = \boldsymbol{\omega} \times \mathbf{d}_j$. We also define the velocity of the centerline $\bar{\mathbf{v}} = \partial_t \bar{\mathbf{x}}$ and, in the rest configuration, the bending stiffness matrix $\mathbf{B}$, shearing stiffness matrix $\mathbf{S}$, second area moment of inertia $\mathbf{I}$, cross-sectional area $A$ and mass per unit length $\rho$. Then, the dynamics[64] of a soft slender body is described by:

$$\rho A \cdot \partial_t^2 \bar{\mathbf{x}} = \partial_s \left( \frac{\mathbf{Q}^T \mathbf{S} \boldsymbol{\sigma}}{e} \right) + e\bar{\mathbf{f}} \tag{3}$$

$$\begin{aligned}\frac{\rho \mathbf{I}}{e} \cdot \partial_t \boldsymbol{\omega} =\ & \partial_s \left( \frac{\mathbf{B}\boldsymbol{\kappa}}{e^3} \right) + \frac{\boldsymbol{\kappa} \times \mathbf{B}\boldsymbol{\kappa}}{e^3} + \left( \mathbf{Q}\frac{\bar{\mathbf{x}}_s}{e} \times \mathbf{S}\boldsymbol{\sigma} \right) \\ & + \left( \rho \mathbf{I} \cdot \frac{\boldsymbol{\omega}}{e} \right) \times \boldsymbol{\omega} + \frac{\rho \mathbf{I}\boldsymbol{\omega}}{e^2} \cdot \partial_t e + e\mathbf{c}\end{aligned} \tag{4}$$

where Eqs. (3) and (4) represent linear and angular momentum balance at every cross section, $e = |\bar{\mathbf{x}}_s|$ is the local stretching factor, and $\bar{\mathbf{f}}$ and $\mathbf{c}$ are the external force and couple line densities, respectively[64].

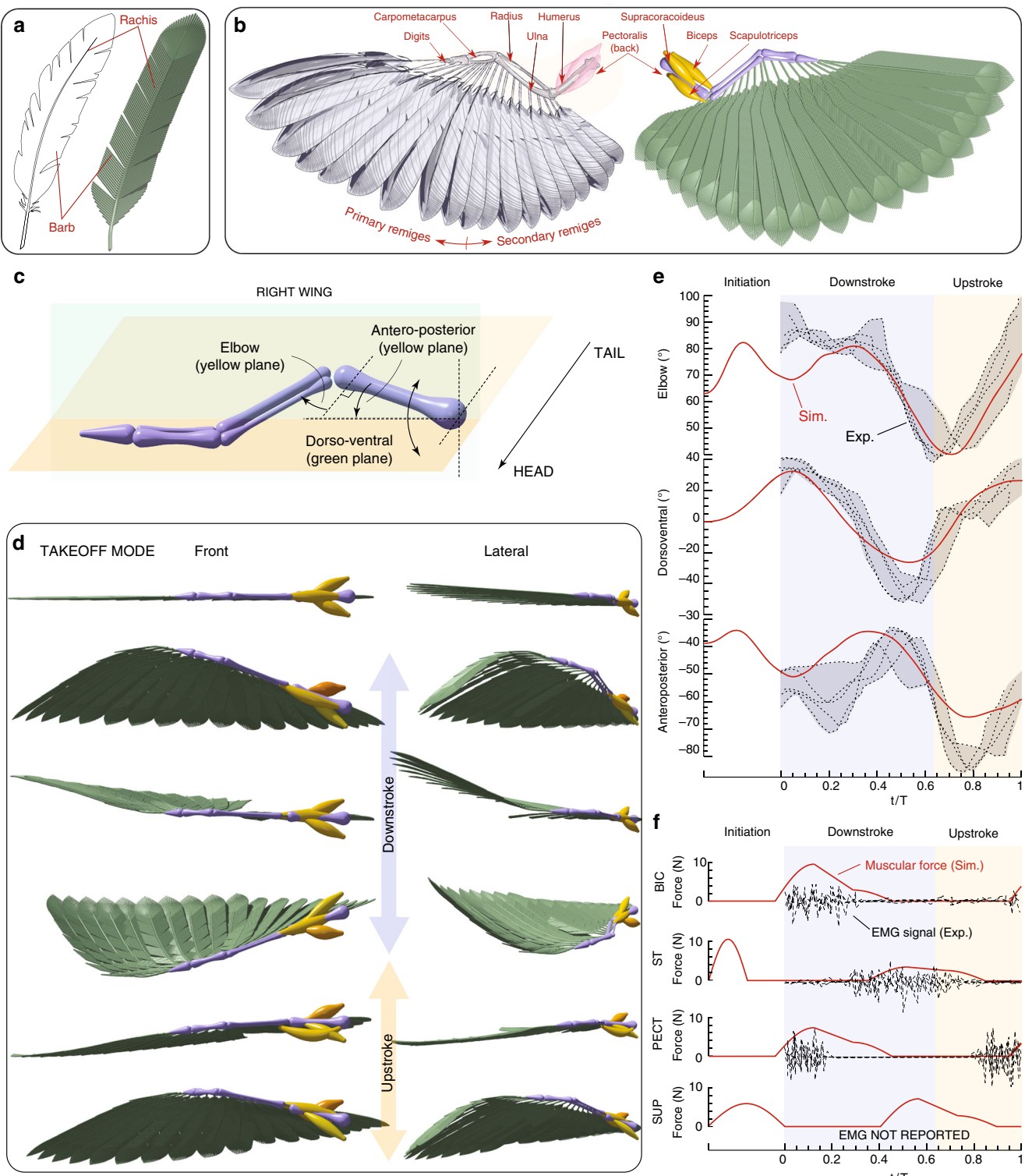

**Fig. 4** Flapping feathered wings. **a** Model of a feather comprised of rachis and barbs. **b** Computational wing (right) consisting of 3171 filaments (close-up visualization from multiple perspectives provided in Supplementary Fig. 5) that mimics the illustration (left) of the wing anatomy. **c** Elbow, dorsoventral and anteroposterior joint angles. **d** Initiation process that lifts the wing from flat position, followed by a single power downstroke and upstroke during the takeoff stage. **e** Joint angle measurements (simulation vs. experiments[82]) for elbow, dorsoventral, and anteroposterior angle. **f** Actuation patterns for four different muscles: biceps (BC), scapulotriceps (ST), pectoralis (PECT), and supracoracoideus (SUP). EMG recordings[82] are represented in black and our simulated muscle activity are represented in red. Settings: Rachis are modeled filaments of radii 1 mm, density $\rho = 2.5 \times 10^{-3}$ g cm$^{-3}$, bending stiffness $EI = 5$ N mm$^2$. Barbs are modeled as filaments of radii 0.6 mm. Total wing area is ~650 cm$^2$. Additional details can be found in Supplementary Note 5

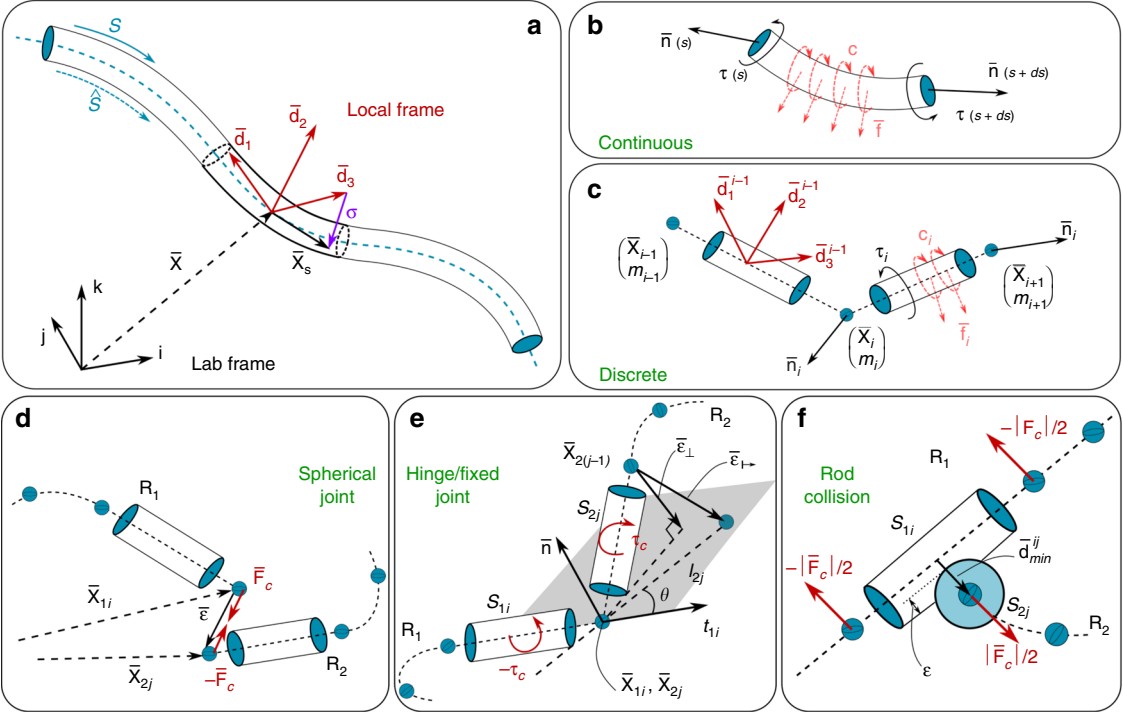

**Fig. 5** The Cosserat rod model. **a** A soft filament described by the centerline coordinate s. **b**, **c** Continuous and discretized model of a rod segment: $\bar{\mathbf{n}}$ and $\boldsymbol{\tau}$ denote the internal force and torque resulting from local deformation, while $\bar{\mathbf{f}}$ and $\mathbf{c}$ represent external force and torque. Assembling filaments. **d** Spherical joint. **e** A hinge joint allows $R_2$ to rotate in a plane defined by $R_1$, while a fixed joint restrict any relative motion between the two adjacent segments. **f** Repulsion forces are enforced when two filaments are in contact

The above continuous representation is discretized into $(n + 1)$ nodes of position $\mathbf{x}_i$ and $n$ connecting cylindrical segments (Fig. 5a–c), so that midline linear displacements are determined by the internal and external forces acting at the nodes (of mass $m_i$), while rotations are accounted for via couples applied to the cylindrical elements. The dynamic behavior of a rod is then computed by integrating the discretized set of equations in time via a second order position Verlet scheme. The details of our numerical implementation can be found in Gazzola et al.[64], together with a rigorous validation against a number of benchmark problems with known analytic solutions as well as experimental investigations involving contact, anisotropic surface friction and highly viscous fluids.

This representation entails a number of advantages: (1) captures 3D dynamics accounting for bend, twist, shear and stretch—fundamental effects in the case of elastomeric or biological materials; (2) continuum actuation, interface and environmental effects can be directly combined with the body dynamics through $\bar{\mathbf{f}}$ and $\mathbf{c}$, rendering the inclusion of (self-)contact, friction, muscular activity, hydrodynamics and adhesion straightforward; (3) its complexity scales linearly with axial resolution, as opposed to cubic for FEM solvers, reducing computing time; (4) the realization that bones, tendons and muscle fiber units are generally slender, provides the rationale for assembling multiple rods with different mechanical properties to model complex musculoskeletal structures.

**Assembly of heterogeneous rods.** In order to assemble multiple active and passive rods into dynamic architectures, it is necessary to prescribe their rules of interaction, via appropriate boundary conditions. In the following we detail our modeling approach to spherical joints, fixed joints, hinges, and rod–rod contact, which are the main interaction modes employed in this study. All other connections that may be employed in this study are derived from these basic ones, and detailed in the SI when relevant.

As a general strategy, we avoid mathematically enforcing 'hard' constraints via Lagrange multipliers as their formulation may be cumbersome, impairing the modularity and versatility of the numerical solver. We instead resort to apply correcting forces and torques through 'soft' displacement–force (or torque) relations. In the following, to simplify the notation, all vectors are expressed in the global coordinate.

**Spherical joint.** A spherical joint (Fig. 5d) approximates the physical connection between two or more filaments that allows them to rotate with respect to each other. One biological instance is a fibrous enthesis where a tendon (or a ligament) is connected to a bone. In our model, a spherical joint is formed by the nodes $i$ and $j$ at the extremities $\mathbf{x}_{1i}$ and $\mathbf{x}_{2j}$ of the filaments $R_1$ and $R_2$, respectively. The external

force $\mathbf{F}_c = k\boldsymbol{\epsilon}$ that holds the filaments' ends connected is proportional to the distance $\boldsymbol{\epsilon} = \mathbf{x}_{2j} - \mathbf{x}_{1i}$. The constant $k$ can thus be interpreted as the joint stiffness.

**Hinge and fixed joint.** A hinge connection is encountered when a rod is confined in a plane defined by another rod. Knee or elbow joints are intuitive examples, as shanks or forearms rotate with only one degree of freedom. In our simulations, a hinge joint (Fig. 5e) is based on the spherical joint model with an additional constraint on the orientation of the end segments $S_{1i}$ and $S_{2j}$ of the connected filaments. The allowed rotational plane is then defined by the tangent $\mathbf{t}_{1i}$ and normal vector $\mathbf{n}$ to the segment $S_{1i}$. To cancel the off-plane vector $\boldsymbol{\epsilon}_\perp$ (that might arise during structure actuation) at the node $\mathbf{x}_{2(j-1)}$, we apply a correcting torque $\boldsymbol{\tau}_c$ proportional to the hinge stiffness through the constant $k$

$$\boldsymbol{\tau}_c = (\mathbf{x}_{2(j-1)} - \mathbf{x}_{2j} + \boldsymbol{\epsilon}_\perp) \times (k\boldsymbol{\epsilon}_\perp), \quad \boldsymbol{\epsilon}_\perp = -[(\mathbf{x}_{2(j-1)} - \mathbf{x}_{2j}) \cdot \mathbf{n}]\mathbf{n}. \quad (5)$$

A fixed joint is evolved from a hinge joint by further constraining the orientation of $S_{2j}$ to a fixed angle $\theta$ relative to $S_{1i}$. A correcting torque is then similarly computed, this time through the off-position vector $\boldsymbol{\epsilon}_\mapsto$

$$\boldsymbol{\tau}_c = (\mathbf{x}_{2(j-1)} - \mathbf{x}_{2j}) \times (k\boldsymbol{\epsilon}_\mapsto), \quad \boldsymbol{\epsilon}_\mapsto = \left[\mathbf{x}_{2j} + l_{2j}(\mathbf{t}_{1i}\cos\theta + \mathbf{b}_{1i}\sin\theta)\right] - \mathbf{x}_{2(j-1)} \quad (6)$$

where $l_{2j}$ is the length $S_{2j}$, and $\mathbf{b}_{1i} = \mathbf{n} \times \mathbf{t}_{1i}$ is the binormal vector. To conform with our Cosserat model implementation (Eqs. (3) and (4)), all the torques are transformed into the local frame and stored in the external couple $\mathbf{c}$.

**Rod collision.** Multiple filaments assembled, for example, in a bundle to form a muscle should not interpenetrate each other. To avoid this and capture contact, we adopt the same approach of Gazzola et al.[64]. Thus, as in the previous sections we introduce additional forces $\mathbf{F}_c$ acting between the discrete elements ($S_{1i}$ and $S_{2j}$, belonging to rods $R_1$ and $R_2$, respectively – Fig. 5f) in contact. To determine whether any two cylindrical elements are in contact, we calculate the minimum distance $d_{\min}^{ij}$ between edges $i, j$ by parameterizing their centerlines $c_i(h) = s_i + h(s_{i+1} - s_i)$ so that $d_{\min}^{ij} = \max_{h_1, h_2 \in [0,1]} ||c_i(h_1) - c_j(h_2)||$. If $d_{\min}^{ij}$ is smaller than the sum of the radii of the two cylinders, then they are in contact and penalty forces are applied to each element as a function of the scalar overlap $\epsilon_{ij} = (r_i + r_j - d_{\min}^{ij})$, where $r_i$ and $r_j$ are the radii of edges $i$ and $j$. If $\epsilon_{ij}$ is smaller than zero, then the two edges are not in contact and no penalty is applied. Denoting as $\mathbf{d}_{\min}^{ij}$ the unit vector pointing from the closest point on edge $i$ to the

closest point on edge $j$, the contact repulsion force is given by

$$\mathbf{F}_c = H(\epsilon_{ij}) \cdot \left[ -k\epsilon_{ij} - \gamma(\mathbf{v}_i - \mathbf{v}_j) \cdot \mathbf{d}_{\min}^{ij} \right] \mathbf{d}_{\min}^{ij} \qquad (7)$$

where $H(\epsilon_{ij})$ is the Heaviside function that ensures a repulsion force is produced only in case of contact ($\epsilon_{ij} \geq 0$). The first term within the square brackets expresses the linear response to the interpenetration distance as modulated by the stiffness $k$, while the second damping term models contact dissipation and is proportional to the coefficient $\gamma$ and the interpenetration velocity $\mathbf{v}_i - \mathbf{v}_j$. In general, the values $k$ and $\gamma$ are related to the maximum forces that muscles or external loads exert on a given structure. For the sake of reproducibility, the employed values are reported in the SI.

**Reporting summary.** Further information on research design is available in the Nature Research Reporting Summary linked to this article.

## Data availability
Data supporting the findings of this manuscript are available from the corresponding author upon reasonable request. A reporting summary for this article is available as a Supplementary Information file.

## Code availability
A non-parallel version of the software used to perform this study is publicly available at Github: https://github.com/mattialab/musculoskeletal-elastica

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

## Acknowledgements

We thank Taher Saif, Rashid Bashir, and Gelson Pagan-Diaz for helpful discussions and the provided experimental images. We also thank Margherita Gazzola for the wing illustration. This study is jointly funded by NSF EFRI C3 SoRo #1830881 (M.G.), NSF/USDA #2019-67021-28989 (M.G.), NSF CAREER #1846752 (M.G.), ONR MURI N00014-19-1-2373 (M.G.), and Strategic Research Initiatives (SRI) program of the University of Illinois at Urbana-Champaign (M.G.). We also thank the Blue Waters project (OCI-0725070, ACI-1238993), a joint effort of the University of Illinois at Urbana-Champaign and its National Center for Supercomputing Applications, for partial support.

## Author contributions

X.Z., F.K.C., T.P. and M.G. designed the research. X.Z., F.K.C. and T.P. performed the research. X.Z., F.K.C., T.P. and M.G. analyzed the data. X.Z., F.K.C., T.P. and M.G. wrote the paper.

## Competing interests

The authors declare no competing interests.
