## [Peer Review File · Nature Communications]

Reviewers' Comments:

Reviewer #1:

Remarks to the Author:

I think overall this is an interesting idea, using ensembles of rod elements to represent complex structures. The examples are well chosen, though the last one—bird wing—is rather sketchy in the validation. Overall, there aren't a lot of details, but it seems roughly in line with what is typical for this journal.

There are several grammatical issues, for example: "represent biological active" and

"This illustrates our method's utility in stripping away layers of complexity that hinder the advancements of biomimetic applications, thus feasible, manufacturable designs"

Also, a spellcheck is needed.

Specific comments:

"Once the biomechanical properties are determined"--are these nonlinear elastic constants for isotropic material composing the muscle? How many such parameters are there? What quantities are being estimated from the "training" data and which are being measured on the resulting biomechanical system? Is the damping (ζ) set by fitting to data?

"geometric rescaling of filament cross-sections"--does this arise naturally from the material's modulus of compression (or a related constant, Poisson ratio)?

Fig. 2d,h--how accurately do you think you have the optimal solution? In Fig. 2d it looks like there are big changes right before the end point.

What is the tail radius for the optimal solution (Fig. 2d)? I would think you would want a more symmetrical motion and also a more flexible tail that has traveling waves of deformation.

A good application of this approach, and one that has been studied with networks of rods previously (works by Tangorra et al., Lauder et al., and Q.Zhu (UCSD) et al.), is a fish pectoral fin—it would be good to give references.

Fig. 4 caption "the illustration (left) adopted from Cheng Li"—is there a reference (e.g. journal article or book) where this can be found?

Fig. 4d and e—what is going before $T = 0$ in these figures? Is there something occurring at negative times?

You mention a "a sinusoidal-like muscle activation pattern"—what is meant by sinusoidal-like? Is it periodic in time? You are only looking at a transient effect in Fig. 4, not periodic.

Reviewer #2:

Remarks to the Author:

The submitted work proposes to use Cosserat (Kirchhoff) elastic rods as a modeling primitive for dynamic simulations of biological systems. Assemblies of active and passive elastic rods are used

to model bones, tendons, and ligaments. The specific choice of rod model is introduced in a cited paper, and instead the submitted work focuses on advancing the argument that rods can serve as an efficient and versatile modeling primitive. The argument is supported by case studies: human elbow joint performing isokinetic concentric flexions; optimization of swimming flagella robot; optimization of a walking robot; optimization of gait for slithering snake; detailed replication of feathered wing.

In the opinion of this referee, the assessment of novelty and impact can be broken down into three parts: (a) the advancement of Cosserat assemblies as a tool for biological modeling, (b) the numerical treatment of Cosserat assemblies, (c) the rigor of the included case studies.

The numerical treatment appears to be sound; it is useful as a resource for reproducing the results (alongside the included source code), and it is not a technical contribution over prior work. As a minor parenthetical remark, it is not immediately clear why bones should be modeled as rods as opposed to, say, rigid bodies or finite elements, since bones are not always slender structures.

An evaluation of the rigor of the included case studies is outside the scope of expertise of this referee.

The case for using Cosserat assemblies in biological modeling is compelling and exciting. The diverse case studies, assembled in one article, support this claim and may increase the impact of rod models. However, the novelty of the argument is less clear due to a previous line of work that appears to have escaped notice:

Pai and his co-workers argued for strand models to model muscle fascicles and tendons, in a line of work from 2004 to present day. See for example:

Dinesh K. Pai, Shinjiro Sueda, and Qi Wei. Simulation of 3D neuro-musculo-skeletal systems with contact, Poster in Advances in Computational Motor Control III, Symposium at the Society for Neuroscience Meeting. San Diego, October 22, 2004.
<https://sites.google.com/site/acmconference/acmcpceedings/2004>

"We are developing a new computational model for neuro-musculo-skeletal simulation.... The model is based on a computational primitive called a "muscle strand." The passive elastic behavior of the strand is based on the theory of Cosserat rods (Rubin, 2000; Pai, 2002). These one-dimensional primitives are well suited for incorporation of muscle activation models along the principal axis of the strand (e.g., (Zajac, 1989; Cheng, Brown, & Loeb, 2000)). "

Shinjiro Sueda, Andrew Kaufman, and Dinesh K. Pai. 2008. Musculotendon simulation for hand animation. *ACM Trans. Graph.* 27, 3, Article 83 (August 2008), 8 pages. DOI:
<https://doi.org/10.1145/1360612.1360682>

"Our simulator is built on two primitives: rigid bodies for bones and spline-based strands for tendons and muscles. Our decision to use strands was motivated by the anatomical structure of real muscle tissue. Muscles consist of fibers, curved in space, which are bundled into groups called fascicles. When a muscle is activated, the fibers contract, which transmits a contractile force directly along each fiber. By using strands in our muscle simulation, we are able to directly model this behavior. Strands allow us to define smooth curves to represent tendons, muscles, or the individual fascicles of each muscle. The forces applied to the strands can be transmitted directly along the curve, and also laterally through constraints. We extend the physically-based spline models ... to include [muscle] activation...."

Wei Q, Sueda S, Pai DK. Physically-based modeling and simulation of extraocular muscles. *Prog Biophys Mol Biol.* 2010;103(2-3):273-283. doi:10.1016/j.pbiomolbio.2010.09.002

"Dynamic simulation of human eye movements, with realistic physical models of extraocular muscles (EOMs), may greatly advance our understanding of the complexities of the oculomotor system and aid in treatment of visuomotor disorders. In this paper we describe the first three dimensional (3D) biomechanical model which can simulate the dynamics of ocular motility at interactive rates. We represent EOMs using "strands", which are physical primitives that can model an EOM's complex nonlinear anatomical and physiological properties. Contact between the EOMs, the globe, and orbital structures can be explicitly modeled....Complex nonlinear EOM physiological properties are incorporated in the computation by explicitly representing the experimental EOM constitutive models....The model is computationally efficient. It provides a good balance between simulation accuracy and speed. EOM motions can be simulated at interactive rates (~ 16 frames/second) on personal computers....Contact between EOMs and the globe is physically handled through sliding constraints, not only for visualization but also for realistically simulating force interaction and lateral sliding movement....Our biomechanical model can simulate extraocular muscle dynamics, which have not been supported by other 3D biomechanical models."

The following paper extends these ideas by developing computationally efficient treatments of tendons:

Prashant Sachdeva, Shinjiro Sueda, Susanne Bradley, Mikhail Fain, and Dinesh K. Pai. 2015. Biomechanical simulation and control of hands and tendinous systems. *ACM Trans. Graph.* 34, 4, Article 42 (July 2015), 10 pages. DOI: <https://doi.org/10.1145/2766987>

It follows that the above line of work should be cited and placed in context of the proposed contributions. It is the opinion of the reviewer that the above line of work began with a focus on Cosserat rod models but steered toward a "strand" model that neglects twist (ignores the state of the material frame). As such, there may be an avenue for the present submission to argue for the role of twist in the modeling primitive, and to demonstrate the importance of twist in the quantitative case studies. Absent such a demonstration, however, it becomes unclear how to differentiate from the above line of work.

As in the present submission, earlier works have also sought to find a compromise between expensive 3D finite elements and reduced models. For instance:

Ye Fan, Joshua Litven, and Dinesh K. Pai. 2014. Active volumetric musculoskeletal systems. *ACM Trans. Graph.* 33, 4, Article 152 (July 2014), 9 pages. DOI: <https://doi.org/10.1145/2601097.2601215>

The case studies presented in the submitted work are interesting and varied. It is not clear whether each case study has been fully contextualized vis a vis the available literature. For instance, the following paper modeled the dynamics of flagellar propulsion using a Cosserat rod model:

M. K. Jawed, N. K. Khouri, F. Da, E. Grinspun, and P. M. Reis. Propulsion and Instability of a Flexible Helical Rod Rotating in a Viscous Fluid. *Phys. Rev. Lett.* 115, 168101. 2015.

Reviewer #3:

Remarks to the Author:

This paper introduces a new way of modeling musculoskeletal movements in animals, using Cosserat rods. As a biologist, I am not a mechanic, and am unable to properly evaluate the full relevance and validity of the mechanical modeling (which forms the underlying basis of the paper). That said, this manuscript was quite exciting to read, as the described tools could be broadly used to tackle innumerable problems in biomechanics, opening up new lines of investigation. It really does seem like this method could revolutionize the field, because biologically-accurate models can

be used to test questions that can't be easily addressed with experimental methods, or can compliment those methods.

Despite my enthusiasm, there are a number of issues both large and small before I can recommend publication.

My major concern involves the non-human animal models in the study. From my perspective, to be convincing the models should closely match the quantitative kinematics of movement of the real biological system. I'm less concerned about matching external forces, which can add a far greater level of complication. This concern arises because for both the snake model and the wing model, the kinematics don't look very realistic. The snake model looks to me like the roundworm *C. elegans*. I believe that the new approach in this manuscript could indeed produce snake-like behavior, but this isn't it. The wing model certainly doesn't match a full stroke, in which flexion at the joints (driven by muscles in each segment) plays a large role. Furthermore, I don't understand the modeling of the lift and drag forces, both of which in a takeoff involve unsteady aerodynamics. Drag was modeled based on the drag coefficient of isolated cylinders, which is certainly not appropriate when the cylinders are close together in the form of a feathered wing. Either stick to the kinematics, or work with a fluid mechanist to model the three-dimensional unsteady flows (which I would not recommend, given the large effort that would take).

Other general issues:

Note that throughout the manuscript, the construction "the Hill's model" is used. Hill's model indeed is widely known and employed, but with an article, it should be termed "the Hill model".

"Patient-specific needs" are mentioned in several places. I don't understand what this means in the context of the work done in this paper, which focuses on biological models. What patients, and for what?

Other specific comments by page:

Page 2

Change "leveraged" to "used". I appreciate that "leveraged" is a widely used term in engineering (particularly in proposals, and often poorly used), but given that this manuscript has heavy mechanics content and leverage has a specific mechanical meaning, it's better to not use this jargon and use a simpler, clearer word.

Spell out FEM.

Page 5

It is a little odd that one of the main muscles modeled in this paper is misspelled throughout the manuscript. Biceps branchii should be biceps brachii.

"with each rod representing motor units." How many motor units? How well does this match the human muscle?

"This allows the muscle to comply with the size principle (49), for which low-force, slow-twitch are associated to small motor units, while high-force, fast-twitch to large ones (see Supplementary Information)." Fix the grammar errors here.

Page 6

Figure 1: Explain what the abbreviations and symbols stand for in the figure legend. The reader shouldn't have to hunt these down in the main text to figure out what they are.

"a fixed joint (element connection that restricts any relative motion) is used for the ulna-radius connection." I'm not sure why this is implemented. In a real human forearm, these two bones are free to twist relative to one another. If they are fixed, then there is no purpose of including both bones; only one element is needed for the model.

Page 7

"We then performed isometric (static) and isokinetic (dynamic) tests for validation against experiments." Whose experiments?

"The simulation data is obtained when the muscle reaches its final equilibrium length η (Fig. 1c)." I don't understand. If this is an isometric simulation, then the muscle's length should not be changing.

Change "discount" to something more appropriate. "decrease"?

"increase at 90 degrees angle" Fix grammar.

Page 9

Where does equation 1 come from?

"Covariance Matrix Adaptaion-Evolution Strategy algorithm" Fix spelling.

"CMA-ES has proven reliable over a range of engineering and biophys- ical problems (54, 55), from fish swimming and schooling (54, 56) to aircraft wake dissipation strategies (57)." Proven reliable in doing what?

"244% improvement in swimmer's maximum speed." Fix grammar.

"with length and radius of 0.19 mm and 32.3 μm , respectively." Use consistent units.

Page 11

"for the next generation bio-hybrid swimming robot design." Add the word "of".

"leading to the fabrication and testing of the largest and fastest biobot to date (19)." I don't know the records, but it seems that a 0.7 mm biobot is not very big. And, what is a biobot?

"Inheriting from previous demonstrations (20)," Inheriting what?

"1.4mm" Always use a space between number and unit (except for temps), here and throughout the manuscript.

"used our simulations to design and dimension the new scaffold" Is scale not part of design? Extra verbiage not needed.

For equation 2, the units don't seem to work out properly.

Page 12

Why does "manufacturability" matter in this paper, which concerns simulations?

"the physics of cell-powered soft robotic systems" I think "muscle-powered" is intended.

Page 13

"Extensive work has been done on understanding snake locomotion (12, 25, 58, 59), typically targeting robotic replicas made of rigid linked elements actuated with servomotors." Of these references, only 59 includes a robotic replica.

"While numerous work has demonstrated the successful gait of a slithering snake (60, 61)" What does that mean? That snakes use a gait that works? Which gait? There is more than one.

Page 14

"We claim that our musculoskeletal representation made of only four overlapping soft longitudinal actuators can closely approximate the idealized continuous reference above, which sets the attainable velocity upper bound (62)." At this point in the manuscript, there is no basis for "claiming" this; I believe the intent here is to "test".

Page 15

In the snake model (shown in Fig 3a), I don't quite understand the how the lateral insertion/originations were modeled. The iliocostalis attaches to the end of the rib, providing a larger moment arm than the semispinalis, which attaches much closer to the midline on the vertebra.

Fig 3a legend: the skeletal drawing was modified from Bruce Jayne's work, and the attribution is both improperly done and wrongly cited.

Page 16

"The identified optimal design exhibit muscle groups that span roughly 30–40 vertebrae (Fig. 3d), agreeing reasonably well with biological observations (63)." A few snakes (mostly arboreal) exhibit this large number, but most species employ a smaller span (<15), at least according to the cited paper. Also, fix the grammar.

"This stems from the discrete, limited number of muscles, and is re- flected in the more prominent lateral sliding of the midline kinematics (Fig. 3f)." Lateral undulation in typical conditions does not include sliding.

"as indicated by the feasibility of a fully compliant, realis- tically slithering snake robot made of a few simple actuators." As far as I'm aware, there were no robots used in this study, only simulations.

"By solving an inverse problem, an optimal musculoskeletal layout is identified," This type of statement will rankle biologists; the model was optimized, but optimality in the biological system is a different story.

"highlighting an ingenious natural solution based on overlapping longitudinal actuators." This is a bit self-congratulatory.

Page 17

"so that the bend- ing stiffness is within biological range (67)." What is the range? Tell us so that we can assess and compare the values.

Change to "four-muscle model".

Page 25

References: There are errors in some of the citations that should be fixed.

Page i

"We are interested in capturing both biomechanics and morphology in musculoskeletal systems, whereby Hill's model and Cosserat theory are leveraged." Hill's model and Cosserat theory are not leveraged, they are simply used.

Page ii

Figure S1: What are CE, PE, and SE? Also, change to "three-element model".

"Each head is made of 360 motor units (9)" What does the reference here mean? Explain.

"we create a biceps head with a cross-sectional area $A = 804.2 \text{ mm}^2$ and 18×10^4 muscle fibers, in agreement with experimental data (10)." In agreement can mean any number of things. Tell us specifically what you mean.

Page iii

Change to "muscle unit's".

Page iii

"we are capable of replicating any musculoskeletal system given the biomechanical properties." Possible, but that remains to be seen. Far too strong of a statement, based on only a few models.

The tendons are tapered (as appears with some of the muscles). What is the effect on the modeling of this tapering?

'antropometry' is misspelled.

Page iv

"see that $\text{fact}(\eta) = \text{Fact}/F_{\text{max}}$ decreases with contraction or extension." Physiologically, muscle contraction refers to the activation of the muscle, which might involve shortening, extension, or no net displacement. I assume that shortening is meant here.

Page ix

In Movie S1, why does the muscle start out in yellow and then change color?

Page x-xi

There are errors in the references.

Response to Reviewer 1:

Modeling and simulation of complex dynamic musculoskeletal architectures

Xiaotian Zhang¹, Fan Kiat Chan¹, Tejaswin Parthasarathy¹ and Mattia Gazzola^{1,2†}

¹Mechanical Sciences and Engineering, University of Illinois at Urbana-Champaign, Urbana, IL 61801, USA

²National Center for Supercomputing Applications, University of Illinois at Urbana-Champaign, Urbana, IL 61801, USA

† To whom correspondence should be addressed; E-mail: mgazzola@illinois.edu.

We thank the reviewer for her/his valuable time, consideration and largely positive assessment. In the following, the comments of the reviewer are listed followed by our responses. All modifications to the manuscript are highlighted in red for the reviewer's convenience.

We hope that the reviewer considers our answers acceptable and the improved manuscript is suitable for publication.

1. I think overall this is an interesting idea, using ensembles of rod elements to represent complex structures. The examples are well chosen, though the last one—bird wing—is rather sketchy in the validation. Overall, there aren't a lot of details, but it seems roughly in line with what is typical for this journal.

We thank the reviewer for her/his interest, for recognizing the novelty of our approach as well as for the comments on our work. We are glad that the reviewer finds our examples well chosen.

Regarding the bird wing, we agree with the reviewer that a more quantitative validation is needed. Subsequently, we deployed a significant effort to improve this section. Towards this we now report a much more comprehensive study: (1) A full stroke cycle is now considered; (2) We now consider available experimental EMG measurements of the wing muscles (Robertson et al. (2012)) as input for our simulations, and (3) the corresponding dynamic wing output is now characterized via all three joint angles (dorso-ventral, antero-posterior and elbow joint) which are then reported and compared with biological data. Therefore, simulated muscles—biceps (BC), scapulotriceps (ST), pectoralis (PECT) and supracoracoideus (SUP) (Fig. 4b, reproduced in the answer to comment #9 for reference)—contract according to the EMG time sequences and with peak forces similar to measurements found in Biewener et al. (1998) (Fig. 4f). As can be seen in the updated Fig. 4d (reproduced here for convenience), when our virtual muscles are actuated in a biologically realistic fashion, the wing as a whole recovers its natural kinematics (Fig. 4d). More details can be found in our answers to comments #8, #9, and #10.

Relative to the level of technical detail, as pointed out by the reviewer, we chose a terse style to avoid diluting the main message, in line with the nature of the journal. Nonetheless, we highlight that:

- (a) Our methods for single rods are described and validated in detail in Gazzola et al. (2018).
- (b) That publication (Gazzola et al. (2018)) is accompanied by a public-domain software *Elastica* on GitHub (link to source code provided here for convenience—<https://github.com/mattialab/elastica>).
- (c) This current work on assemblies of rods will be combined with *Elastica* and distributed as an open-source code on GitHub as well. The individual test cases presented here will be directly available for testing and reproducibility.
- (d) Finally, all main parameters and extensions with respect to Gazzola et al. (2018) are described in the Supplementary Information.

We hope that the reviewer appreciates our efforts in assuring code transparency, dissemination and most importantly, reproducibility.

2. There are several grammatical issues, for example: "represent biological active" and "This illustrates our method's utility in stripping away layers of

complexity that hinder the advancements of biomimetic applications, thus feasible, manufacturable designs". Also, a spellcheck is needed.

We thank the reviewer for her/his careful reading and we apologize for the grammatical and spelling issues that may have caused any confusion. We have thoroughly gone over the manuscript, performed an extensive spellcheck and made the necessary corrections, including the ones suggested by the reviewer.

3. "Once the biomechanical properties are determined"--are these nonlinear elastic constants for isotropic material composing the muscle? How many such parameters are there? What quantities are being estimated from the "training" data and which are being measured on the resulting biomechanical system? Is the damping (zeta) set by fitting to data?

We apologize for our terse presentation, and take the chance here to clarify the above aspects. For a more coherent and structured answer, we break down the reviewer's comments into several points addressing: (a) parameters directly set based on values found in literature, (b) modeling of muscle output forces based on experimental data, and (c) validation procedure. We hope that the reviewer finds this helpful.

(a) Biomechanical properties set based directly on values found in literature

In our simulations, different rods represent different biological elements such as bones, tendons and muscle units. Each element, whether active or passive, is characterized by basic mechanical properties (density ρ , Young's modulus E , etc.) and geometric properties (length, radius, etc.). Active elements like muscles are also able to produce forces upon contraction, hence their active force output function must be defined in addition to the above listed properties (more on this in the next part—modeling of muscle output forces).

In the case of the human elbow, the geometrical and mechanical properties make up 13 parameters in total (see table below). Given the existing large body of literature in studying human physiology, representative average values (for a 1.8 m height individual) have been well characterized. In Table S1 in the Supplementary Information (reproduced here for reference), we report these values and the corresponding literature references. We used the same constant values in our model. All slender elements are considered isotropic and incompressible ($\rho = \text{constant}$). We also note that while an individual rod is modeled as isotropic, a bundle of them (muscle) is not in general, because of contact forces. Finally, we refer to answer #4 for a detailed explanation on how the hyper-elastic behavior of soft tissue is captured in our model through filaments cross-section rescaling.

Table S1: Bio-mechanical properties of muscles, tendons and bones used in simulations.

Parameters	Values	Parameters	Values
Muscle E (11)	16 kPa	Muscle Density (12)	1.06 g/cm ³
Bone E (13)	11 GPa	Bone Density (14)	1.9 g/cm ³
Tendon E (15)	500 MPa	Tendon Density (15)	1.67 g/cm ³
Humerus Length ^a	0.34 m	Humerus Radius ^a	0.0105 m
Radius Length ^a	0.255 m	Radius Radius ^a	0.0105 m
Ulna Length ^a	0.255 m	Ulna Radius ^a	0.0055 m
Poisson Ratio	0.5		

^aDimensions are calculated from the table of human **anthropometry** for a 1.8 m height individual (I).

(b) Modeling of muscle output forces based on experimental data

The parameters listed in the table above are sufficient to represent passive elements such as bones or tendons. Active elements such as muscles are instead also able to produce additional forces upon contraction or extension. Then, their force output as a function of their strain (captured here by $\eta = \text{length} / \text{rest length}$) must be modeled. In our work, we employed experimental data from isometric testing (Fig. 1c reproduced below for reference) to determine the force–strain function for bicep muscle units in a static context. We follow the characterization by Rubenson et al. (2012) which considers the force output as the sum of two components:

- An active contraction force that peaks for a muscle length close to its rest length ($\eta = 1$) and decreases as the muscle shortens ($\eta < 1$) or elongates ($\eta > 1$).
- A passive muscle force that is small for $\eta < 1$, and grows non-linearly as the muscles elongates ($\eta > 1$). This particular behavior arises from the membrane of muscle fibers which are stiff and produce high restoring forces when stretched, while they are in slack mode when compressed. Their effect is purely mechanical and corresponds to an effective increase of the Young's modulus for $\eta > 1$.

(c) Isometric Testing - Static

(d) Isokinetic Testing - Dynamic

Model of active force component

For the active component of muscle force (F_{act}), we employ the experimental data from Rubenson et al. (2012) (Fig. 1c in main text, reproduced here for context) and apply a fourth order polynomial fit as mathematically described in Eq. S3 (reproduced below for reference). The function $f_{act}(\eta)$ gives us the length-dependent active force output via $F_{act} = F_{max} \cdot f_{act}(\eta)$ so that when $\eta = 1$ (rest length), we recover the maximum force $F_{act} = F_{max}$ consistent with Rubenson et al. (2012). Here we chose a fourth order polynomial as it provides good fitting accuracy while avoiding spurious interpolatory oscillations.

$$f_{act}(\eta) = 6.405\eta^4 - 24.42\eta^3 + 29.64\eta^2 - 12.01\eta + 1.385 \quad 0.5 \leq \eta \leq 1.6$$

By dividing F_{max} by the bicep area A at rest, we obtain the maximum contractile stress (σ_m), which is the one used in this work.

Model of passive force component

Similarly, here we set to model the passive component of the muscle force (F_{pass}). We first provide an intuition on this passive component by briefly highlighting the text in Supplementary Information that explains the origin of these passive forces in a muscle.

Page iv, paragraph 1, line 9 (Supplementary Information):

“For $\eta < 1$, the overall muscle force consists of only F_{act} since the parallel elastic component of the membrane wrapped around the muscle fibers is in a slack mode and tension only begins to grow non-linearly for $\eta > 1$ (26). We model this parallel elastic mechanism by constructing a piecewise continuous Young’s modulus function $E_{membrane}(\eta)$ of the filaments that reflects this behavior.”

The parallel elastic mechanism that grows non-linearly for $\eta > 1$ mentioned in Winter (2009) is also reflected in the behavior of the passive component data in Rubenson et al. (2012). We then approximate the parallel elastic mechanism by fitting the experimental data from Rubenson et al. (2012) with a fourth order polynomial to produce an additional non-linear Young’s modulus function $E_{membrane}(\eta)$ that characterizes the behavior of the membrane wrapped around the muscle fibers as mathematically described in Eq. S3 (reproduced below for reference).

$$E_{membrane}(\eta) = \begin{cases} 0, & \eta < 1 \\ \frac{F_{max}}{A} \cdot \left(\frac{3.375\eta^4 - 11.33\eta^3 + 13\eta^2 - 5.05\eta}{\eta - 1} \right), & \eta \geq 1 \end{cases}$$

We note that the numerator in brackets is our fitting polynomial, while the denominator accounts for the rescaling of cross-sectional areas as the muscle elongates (for mass conservation). This Young’s modulus function is then translated into a passive tensile force F_{pass} that contributes to the overall static muscle force $F_{qs}(\eta)$ such that

$$F_{qs}(\eta) = F_{act} + F_{pass} = f_{act}(\eta) \cdot F_{max} + \frac{E_{membrane}(\eta) \cdot A \cdot (\eta - 1)}{\eta}$$

Thus, the effect of the muscles' non-linear elastic response is then reflected in our method via passive forces F_{pass} (shown above) applied to the simulated muscles. Therefore, $E_{\text{membrane}}(\eta)$ is not directly defined in the mechanical property of the muscle-Cosserat rod, but rather taken into account through passive forces F_{pass} . Once $f_{\text{act}}(\eta)$ and $E_{\text{membrane}}(\eta)$ are determined, we assembled a virtual muscle bundle and performed isometric testing, recovering the behavior observed experimentally.

(c) Validation procedure and damping parameter ζ

Isometric test

We confirm the correct behavior of our model by conducting *in silico* an isometric test to recover the Force–Length response of Rubenson et al. (2012) (Fig. 1c). We initialize our simulated muscle at rest length ($\eta = 1$) and apply an external tension force $F_{\text{set}} < F_{\text{max}}$. The muscle then performs its Maximum Voluntary Contraction (MVC) and we let the system evolve dynamically from $\eta=1$ to its equilibrium state. By repeating this exercise for various values of $F_{\text{set}} < F_{\text{max}}$, we can reconstruct the Force–Length behavior, which is then confirmed to agree with experiments (Fig. 1c).

We have updated the manuscript as follows.

Page 7, paragraph 3, line 6:

“To perform the test in silico, we use available experimental data (Fig. 1c) to compute polynomial fittings that dictate the muscle active MVC and passive elastic response (Fig. 1c) as functions of its elongation η (SI). Once these biomechanical properties are determined, we let the muscle (initialized at rest length $\eta = 1$) perform its MVC while applying prescribed external forces $F_{\text{set}} \leq F_{\text{max}}$ at its ends. The simulation then dynamically evolves the muscle to its static equilibrium length η . By repeating this experiment for various F_{set} , we can relate muscle length to static force output. This relation is illustrated in Fig. 1c, confirming good match between simulations and experiments.”

Page v, paragraph 1, line 2 (Supplementary Information):

“In order to perform the test in-silico, we initialize our simulated muscle at rest length ($\eta = 1$) and prescribe different external forces F_{set} (inset of Fig. 1c). The simulation then evolves the biomechanical system dynamically from $\eta = 1$ to its equilibrium state (i.e. until the muscle no longer changes in length). The output of the simulation—final equilibrium length of the muscle—is then measured. We can then reconstruct the overall muscle Force-Length curve, which provides a good approximation to the experimental muscle behavior (Fig. 1c).”

Isokinetic test

We employ data from the isokinetic test to determine the damping parameter ζ , which is important in dynamic settings. Our logic is as follows: for each angular velocity, we determined ζ so as to match the experimentally measured torque (Fig. 1d). Then, to check

whether the numerically obtained ζ are physiologically meaningful, we compare them with the theoretical estimates derived from the Hill model (Eq. S5 reported here for reference).

$$\zeta(v) = \frac{(F_{qs} + a)F_{qs}l_0}{av_0 + vF_{qs}}. \quad (S5)$$

As can be seen in Fig. 1d, both curves are found to agree reasonably well, confirming the soundness of our computational model. We have revised the manuscript to make this more explicit and to avoid any confusion that may arise.

Page 8, paragraph 2, line 4:

“We take into account these effects through a damping coefficient ζ , which is numerically set to match simulated and experimental torque outputs (Fig. 1d). The resulting ζ was then compared with theoretical estimates (58) (Eq. S5) and found to be in reasonable quantitative agreement (Fig. 1d).”

4. "geometric rescaling of filament cross-sections"--does this arise naturally from the material's modulus of compression (or a related constant, Poisson ratio)?

We apologize for the terse explanation on the geometric rescaling of filament cross-sections. We consider incompressible materials (density $\rho = \text{constant}$), so that the Poisson ratio is fixed and constant $\nu = 0.5$. Since we do allow for axial stretching or compression, the cross-sectional area of each computational element has to be rescaled appropriately to conserve mass. We track local dilatation through the local stretching factor $e = \frac{d\bar{s}}{ds} = |\partial_s \bar{x}| = |\bar{x}_s|$, defined as the ratio between the current ($d\bar{s}$) and rest configuration (ds) element length. Thus, at each point in time, the current cross-sectional area (\bar{A}) relates to the rest one A through $\bar{A} = \frac{A}{e}$. All other quantities (bending/twist stiffness \mathbf{B} , shear/stretch \mathbf{S} , second area moment of inertia \mathbf{I} , etc.) that depend on area or element length are then rescaled accordingly, giving rise to the following relations (the bar indicates quantities relative to the current configuration)

$$d\bar{s} = e \cdot ds, \quad \bar{A} = \frac{A}{e}, \quad \bar{\mathbf{I}} = \frac{\mathbf{I}}{e^2}, \quad \bar{\mathbf{B}} = \frac{\mathbf{B}}{e^2}, \quad \bar{\mathbf{S}} = \frac{\mathbf{S}}{e}, \quad \bar{\kappa} = \frac{\kappa}{e}$$

which in turn feed into our linear and angular momentum equations:

$$\begin{aligned} \rho A \cdot \partial_t^2 \bar{\mathbf{x}} &= \partial_s \left(\frac{\mathbf{Q}^T \mathbf{S} \sigma}{e} \right) + e \bar{\mathbf{f}} \\ \frac{\rho \mathbf{I}}{e} \cdot \partial_t \boldsymbol{\omega} &= \partial_s \left(\frac{\mathbf{B} \boldsymbol{\kappa}}{e^3} \right) + \frac{\boldsymbol{\kappa} \times \mathbf{B} \boldsymbol{\kappa}}{e^3} + \left(\mathbf{Q} \frac{\bar{\mathbf{x}}_s}{e} \times \mathbf{S} \sigma \right) \\ &+ \left(\rho \mathbf{I} \cdot \frac{\boldsymbol{\omega}}{e} \right) \times \boldsymbol{\omega} + \frac{\rho \mathbf{I} \boldsymbol{\omega}}{e^2} \cdot \partial_t e + e \mathbf{c} \end{aligned}$$

A step-by-step mathematical derivation can be found in Gazzola et al. (2018). Therefore, the reviewer is correct in that the geometric rescaling arises naturally, due to incompressibility.

We note that the use of a linear **stress**-strain relation ($\sigma = E\epsilon$, $\sigma = F/A$, $\epsilon = \Delta l/l_0$) combined with the dynamic rescaling of the cross-sections during stretching and compression results overall in a non-linear **force**-strain response (because $F = \sigma A$ but A is not constant!). This is quantified in the plot from Gazzola et al. (2018) (reproduced below for convenience), where the authors compared the load responses between our Cosserat rod, Neo-Hookean and Mooney-Rivlin models for a rubber rod with Young's modulus $E = 10^7$ Pa under axial stretching. Our approach is shown to reasonably approximate the above more advanced hyperelastic models within 30% deformation range. More details can be found in Gazzola et al. (2018). It is important to realize this because hyperelastic models are suitable for biological tissues.

We have revised the manuscript to reflect details on the implementation and for readability, redirect readers to the equations and variables (local stretching factor) relevant to achieving incompressibility of material.

Page 8, paragraph 3, line 3:

“Our model accounts for incompressibility (Poisson ratio $\nu = 0.5$) through the local dilatation factor e of Eqs. 3, 4 (Methods Section—mathematical derivation can be found in (61)), and prevents interpenetration by performing collision-check (Eq. 7).”

5. Fig. 2d,h--how accurately do you think you have the optimal solution? In Fig. 2d it looks like there are big changes right before the end point.

We thank the reviewer for raising this point which indeed deserves attention. We then rerun our entire optimization campaign for a larger number of generations and replotted Fig. 2c (reproduced here for reference) in the revised manuscript.

We observe that there are no longer measurable gains in the bot's speed beyond generation 40, which is the first indication that we have achieved the optimal solution. We also compared the average speed of the entire population in each generation to the best solution obtained in the optimization course and found that they are comparable, which further confirms convergence. Finally, we note that CMA-ES, a stochastic search algorithm, was able to recover the original solution in an entirely new optimization campaign (with a different seed for the random number generator), which indicates that the optimum found is robust. We point out that CMA-ES does not mathematically ensure convergence to the global optimum. Nonetheless, from a practical standpoint, CMA-ES has been proven robust and effective in dealing with multi-modal (typically encountered in non-linear mechanics applications) and low-dimensional continuous problems, setting the long-standing standard in the GECCO benchmark suite (Hansen et al. (2009), Hansen et al. (2010) and Nguyen et al. (2017)). We now briefly highlight this in the revised manuscript.

Page 11, paragraph 1, line 2:

“While there is no mathematical proof of convergence to global optimum, CMA-ES has proven reliable in dealing with multi-modal, low-dimensional continuous problems (66, 67) and has been employed in a range of engineering and biophysical applications (68–71).”

6. What is the tail radius for the optimal solution (Fig. 2d)? I would think you would want a more symmetrical motion and also a more flexible tail that has traveling waves of deformation.

We should have been more specific in describing the obtained optimal solution. The optimal tail radius is $4.3\ \mu\text{m}$, which is 19% thinner than the original design, and therefore more flexible. As pointed out by the reviewer, in general a flexible tail that allows traveling waves of deformation is desirable for swimming, and the reviewer’s intuition is confirmed by our optimal solution, which clearly exhibits a preference for a more flexible tail as exemplified by the deformation envelopes of Fig. 2d (reported here for reference).

Nonetheless, a good swimming device needs to balance out opposite effects: larger amplitudes can in fact generate more thrust but also more drag. This balance is captured by the optimizer, which did not select the lower bound of attainable tail radius ($4\ \mu\text{m}$). As a sanity check, we ran a case with the minimum radius and indeed observed degraded performance as shown in the figure below (not reported in main text for the purpose of exposition).

We also wish to point out that tail flexibility and length are to be considered concurrently, since they together determine the wave number that characterizes the resulting traveling waves. Here the length is fixed, so that a small decrease in bending stiffness might not be sufficient to excite the next mode of deformation: A longer tail might instead allow for even thinner radii.

Finally, we agree with the reviewer that a more symmetrical motion might be beneficial. Here we set up our optimization problem to account for actual manufacturability and consistency with the original experimental setup. Symmetric actuation via, for example, antagonistic cardiomyocytes is problematic as these cells contract spontaneously, rendering their control difficult. This might be achievable with other technique or cell types, for example through the use of optogenetic skeletal muscles responsive to different light wavelengths. Here, to aid validation and comparison with experiments (Williams et al. (2014)), we chose to maintain the same asymmetric setup.

In the main text, we now briefly comment on these points raised by the reviewer.

Page 11, paragraph 3, line 3:

*"The cells are attached 190 μm away from the head and the tail is 19% thinner (**4.3 μm**) than the original design."*

Page 11, paragraph 3, line 8:

"We note that the optimizer did not select the lower bound of attainable tail radius, which suggests that balance between flexibility and drag associated with large tail deflection is needed for optimal performance."

7. A good application of this approach, and one that has been studied with networks of rods previously (works by Tangorra et al., Lauder et al., and Q.Zhu (UCSD) et al.), is a fish pectoral fin—it would be good to give references.

We thank the reviewer for referring us to the work of these groups which are indeed relevant to our study. We have then selected two papers (listed below for convenience) which we think are most closely-related and added them in the introduction, where we briefly recapped recent efforts in the field.

- Lauder, George V., et al. "Fish biorobotics: kinematics and hydrodynamics of self-propulsion." *Journal of experimental biology* 210.16 (2007): 2767-2780.
- Zhu, Qiang, and Kourosch Shoele. "Propulsion performance of a skeleton-strengthened fin." *Journal of Experimental Biology* 211.13 (2008): 2087-2100.

Page 2, paragraph 2, line 4:

"These considerations have prompted a number of soft robotic investigations in which artificial compliant materials and highly stretchable and shearable elastomeric structures

are used in a variety of applications from gripping, grasping, manipulation (4–8) and artificial muscles (9), to a zoo of robotic creatures (10) from snakes (11,12), **fish (13–15)** and octopuses (16,17) to insects (18) and bats (19).”

8. Fig. 4 caption “the illustration (left) adopted from Cheng Li”—is there a reference (e.g. journal article or book) where this can be found?

We apologize for the lack of proper reference to the scientific illustration (reproduced below for context) and have updated the manuscript to include the link directing to the original work from Cheng Li (2007). We note that the referred illustration is not part of a scientific publication but instead is an artist’s literature reproduction. The link is provided here for convenience: <http://www.lilycli.com/personal.html#&gid=2&pid=5>

Page 20, fig. 4 caption, line 1:

“(b) Computational wing (right) consisting of 3171 filaments that mimics the illustration (left) adapted from Cheng Li’s artwork (67).”

9. Fig. 4d and e—what is going before $T = 0$ in these figures? Is there something occurring at negative times?

Regarding the initiation phase ($T < 0$) mentioned by the reviewer, we emphasize that it simply allows us to bring the wing from its starting flat configuration into position for downstroke movement. This preparatory phase is numerically necessary but does not correspond to a biologically occurring situation, thus we do not report a comparison with experiments.

We also bring to the reviewer’s attention the updated wing section in the revised manuscript. We recognize the need for a more quantitative and rigorous comparison with biological data and have deployed an extensive effort in that aspect.

Instead of simulating only the downstroke phase as in the original submission, we now simulate one complete stroke cycle of the takeoff flight mode. We retain our four-muscle

model as in the original submission (Fig. 4b as reproduced here for reference) and input electromyography (EMG) recordings (Robertson et al. (2012)) in our simulation. When our virtual muscles are actuated with biologically realistic EMG time sequences and peak forces (similar to peak measurements found in Biewener et al. (1998)) (Fig. 4f), the wing recovers its natural kinematics.

We stress that this is not a trivial feat. Indeed, (1) the wing is a highly non-linear system and (2) we note that the four shoulder muscles in our simulation control only two joint angles during flexion–extension: supracoracoideus–pectoralis pair controls the dorso-

ventral angle of the shoulder and the biceps–scapulotriceps pair controls the elbow angle. The temporal undulation of antero–posterior angle is therefore the direct outcome stemming from the dynamic interaction between the active muscles, and the wing structure. The obtained full stroke kinematics are now reported in Fig. 4e, illustrating a good level of quantitative fidelity relative to biological data. The main discrepancies are observed for the antero-posterior angle, which is not surprising as we have no direct control on it and entirely depends on the structure’s passive response. The fact that we do capture its dynamics (maximum deviation of the antero-posterior angle with experiments is ~ 10 degrees, comparable to measurement variations) is still remarkable especially considering the challenges related to the lack of consistent biological datasets. Indeed, although feathered flapping flight has been extensively studied, the reported kinematics, muscle actuation and morphological data are often disjointed as they might refer to different species/specimens and/or flight conditions. For example, accurate morphological measurements could be found for certain species, while careful kinematics may be reported for others. Given the challenging data landscape, we chose to take a pragmatic and logical approach and tried to build a model as consistent as possible by blending together the most closely related datasets we could find. Nonetheless, despite these uncertainties our modeling approach was able to qualitatively and quantitatively capture the most salient kinematic features.

We have updated the manuscript as follows.

Page 19, paragraph 2, line 1:

“We then set to reproduce the kinematics of wings morphing through a full stroke cycle during takeoff mode. We first initialize our simulated wing in a straight, flat configuration (Fig. 4d) and over the initiation phase, set (arbitrarily) the muscle activation via Eq. S6 so as to prepare the wing for the downstroke phase (Fig. 4e,f). During the downstroke and upstroke phase, the muscle actuation patterns (Eqs. S7–10) are instead based on experimentally recorded electromyography (EMG) signals (80) (Fig. 4f). Since EMG measurements do not provide the magnitude at which the muscles operate (only their time sequences), we set the muscle actuation force (~ 25 N) compatibly with the forces reported in (86) (SI). As can be seen in Fig. 4e, our model realistically captures the temporal evolution of the three joint angles, in agreement with experimental measurements (80). This is a non-trivial task given the highly non-linear interplay between muscle actuation, passive structural dynamics and aerodynamic loads. The main discrepancy is observed for the antero-posterior joint angle. This is not surprising since the four muscles do not directly affect it and its time evolution emerges as a result of the overall system dynamics. Thus, this is the angle most sensitive to modeling approximations. In this context, it is still notable that our simulations can capture it qualitatively, and that the maximum deviation from experiments amounts to ~ 10 degrees, comparable to measurement variations.”

10. You mention a “a sinusoidal-like muscle activation pattern”—what is meant

by sinusoidal-like? Is it periodic in time? You are only looking at a transient effect in Fig. 4, not periodic.

The reviewer is right in that the takeoff is a transient gait. Nonetheless, it is still constituted by a few repeating similar strokes as illustrated in Robertson et al. (2012). Hence, our modeling via a periodic signal. We also note that in the revised manuscript, we now prescribe our muscle actuation patterns based on EMG recordings. Therefore, the original sinusoidal function has been replaced with a piece-wise periodic function (Fig. 4f) to account for that. The explicit definitions (Eqns. S6–9) is reported in the Supplementary Information.

We thank the reviewer for her/his careful evaluation and critical comments which helped us in improving our manuscript. We believe it is now a significantly stronger report. We hope the reviewer finds our answers satisfactory and the revised manuscript is suitable for publication.

References used in this response:

- A. Biewener, W. R. Corning, and B. W. Tobalske. In vivo pectoralis muscle force-length behavior during level flight in pigeons (*columba livia*). *Journal of Experimental Biology*, 201(24):3293–3307, 1998.
- M. Gazzola, L. H. Dudte, A. G. McCormick, and L. Mahadevan. Forward and inverse problems in the mechanics of soft filaments. *Royal Society Open Science*, 5(6), 2018.
- N. Hansen. Benchmarking a bi-population cma-es on the bbob-2009 function testbed. In *Proceedings of the 11th Annual Conference Companion on Genetic and Evolutionary Computation Conference: Late Breaking Papers*, pages 2389–2396. ACM, 2009.
- N. Hansen and R. Ros. Benchmarking a weighted negative covariance matrix update on the bbob-2010 noiseless testbed. In *Proceedings of the 12th annual conference companion on Genetic and evolutionary computation*, pages 1673–1680. ACM, 2010.
- A.V. Hill. The heat of shortening and the dynamic constants of muscle. *Proceedings of the Royal Society of London B: Biological Sciences*, 126(843):136–195, 1938.
- G. V. Lauder, E. J. Anderson, J. Tangorra, and P. Madden. Fish biorobotics: kinematics and hydrodynamics of self-propulsion. *Journal of Experimental Biology*, 210:2767–2780, 2007.
- C. L. Li, “Bird wing anatomy,” <http://www.lilycli.com/personal.html>, 2007. [Online; accessed 3-April-2019].
- D. M. Nguyen and N. Hansen. Benchmarking cmaes-apop on the bbob noiseless testbed. In *Proceedings of the Genetic and Evolutionary Computation Conference Companion*, pages 1756–1763. ACM, 2017.
- M. B. Robertson and A. A. Biewener, “Muscle function during takeoff and landing flight in the pigeon (*columba livia*),” *Journal of Experimental Biology*, vol. 215, no. 23, pp. 4104–4114, 2012.
- J. Rubenson, N. Pires, H. Loi, G. Pinniger, and D. Shannon. On the ascent: the soleus operating length is conserved to the ascending limb of the force–length curve across gait mechanics in humans. *Journal of Experimental Biology*, 215(20):3539–3551, 2012.
- B. Williams, S. Anand, J. Rajagopalan, and M. Saif. A self-propelled biohybrid swimmer at low reynolds number. *Nature communications*, 5, 2014.
- A. Winter. *Biomechanics and motor control of human movement*. John Wiley & Sons, 2009.
- Q. Zhu and K. Shoele. Propulsion performance of a skeleton-strengthened fin. *Journal of Experimental Biology*, 211(13):2087–2100, 2008.

Response to Reviewer 2:

Modeling and simulation of complex dynamic musculoskeletal architectures

Xiaotian Zhang¹, Fan Kiat Chan¹, Tejaswin Parthasarathy¹ and Mattia Gazzola^{1,2†}

¹Mechanical Sciences and Engineering, University of Illinois at Urbana-Champaign, Urbana, IL 61801, USA

²National Center for Supercomputing Applications, University of Illinois at Urbana-Champaign, Urbana, IL 61801, USA

† To whom correspondence should be addressed; E-mail: mgazzola@illinois.edu.

We thank the reviewer for her/his valuable time, consideration and largely positive assessment. In the following, the comments of the reviewer are listed followed by our responses. All modifications to the manuscript are highlighted in red for the reviewer's convenience.

We hope that the reviewer considers our answers acceptable and the improved manuscript is suitable for publication.

The submitted work proposes to use Cosserat (Kirchhoff) elastic rods as a modeling primitive for dynamic simulations of biological systems. Assemblies of active and passive elastic rods are used to model bones, tendons, and ligaments. The specific choice of rod model is introduced in a cited paper, and instead the submitted work focuses on advancing the argument that rods can serve as an efficient and versatile modeling primitive. The argument is supported by case studies: human elbow joint performing isokinetic concentric flexions; optimization of swimming flagella robot; optimization of a walking robot; optimization of gait for slithering snake; detailed replication of feathered wing.

In the opinion of this referee, the assessment of novelty and impact can be broken down into three parts: (a) the advancement of Cosserat assemblies as a tool for biological modeling, (b) the numerical treatment of Cosserat assemblies, (c) the rigor of the included case studies.

The numerical treatment appears to be sound; it is useful as a resource for reproducing the results (alongside the included source code), and it is not a technical contribution over prior work. As a minor parenthetical remark, it is not immediately clear why bones should be modeled as rods as opposed to, say, rigid bodies or finite elements, since bones are not always slender structures.

An evaluation of the rigor of the included case studies is outside the scope of expertise of this referee.

The case for using Cosserat assemblies in biological modeling is compelling and exciting. The diverse case studies, assembled in one article, support this claim and may increase the impact of rod models.

We thank the reviewer for the overall positive assessment and for acknowledging the motivation and novelty of our work. Below we address the reviewer's comments and in particular, we contextualize the contribution of our work in light of the suggested literature (comment #1). Indeed, the mentioned articles are relevant to the present work. We now refer to them in the main text, and we apologize for missing this important body of work.

Before delving into the review of these papers, we wish to address here the remark relative to the rigid-body or Finite Element Method modeling of bones. Indeed, as the reviewer rightly points out, bones are not always slender and hence in general should be modeled using FEM or as rigid bodies. In this work, we chose to maintain a uniform mathematical representation throughout the manuscript (to avoid potential confusion arising from the use of mixed models). Given the specific geometry of the bones considered here (humerus, ulna and radius), the choice of using Cosserat rods is justified (all these bones are indeed slender). Nonetheless, the use of rigid body representations would be advantageous in terms of computing efficiency. We note that the extension to a mixed environment of rigid objects and Cosserat rod assemblies is straightforward and

will be included in our open source software. We have also been considering mixed Cosserat-FEM representations, especially in the context of bio-hybrid robots coupled with swelling hydrogels. Such an extension requires an in-depth numerical analysis and is beyond the scope of this work, although it is definitely of interest.

To explicitly address this remark, we add the following sentence to the main text.

Page 6, Paragraph 1, line 2:

*“Towards the completion of a full elbow assembly, we consider the humerus, ulna and radius (**all of which are slender bones**) which are represented as passive, rigid rods with tapering segmental radii (Fig. 1b, purple elements). Similarly, proximal and distal tendons are modeled as tapered passive, but this time elastic, rods (Fig. 1b, yellow elements). **We note that bones are not always slender, in which case a mixed representation that includes rigid bodies or Finite Element Methods should be employed.**”*

1) However, the novelty of the argument is less clear due to a previous line of work that appears to have escaped notice:

Pai and his co-workers argued for strand models to model muscle fascicles and tendons, in a line of work from 2004 to present day. See for example:

Dinesh K. Pai, Shinjiro Sueda, and Qi Wei. Simulation of 3D neuro-musculo-skeletal systems with contact, Poster in Advances in Computational Motor Control III, Symposium at the Society for Neuroscience Meeting. San Diego, October 22, 2004. <https://sites.google.com/site/acmcconference/acmcproceedings/2004>

“We are developing a new computational model for neuro-musculo-skeletal simulation.... The model is based on a computational primitive called a “muscle strand.” The passive elastic behavior of the strand is based on the theory of Cosserat rods (Rubin, 2000; Pai, 2002). These one-dimensional primitives are well suited for incorporation of muscle activation models along the principal axis of the strand (e.g., (Zajac, 1989; Cheng, Brown, & Loeb, 2000)).”

Shinjiro Sueda, Andrew Kaufman, and Dinesh K. Pai. 2008. Musculotendon simulation for hand animation. ACM Trans. Graph. 27, 3, Article 83 (August 2008), 8 pages. DOI:<https://doi.org/10.1145/1360612.1360682>

“Our simulator is built on two primitives: rigid bodies for bones and spline-based strands for tendons and muscles. Our decision to use strands was motivated by the anatomical structure of real muscle tissue. Muscles consist of fibers, curved in space, which are bundled into groups called fascicles. When a muscle is activated, the fibers contract, which transmits a contractile force directly along each fiber. By

using strands in our muscle simulation, we are able to directly model this behavior. Strands allow us to define smooth curves to represent tendons, muscles, or the individual fascicles of each muscle. The forces applied to the strands can be transmitted directly along the curve, and also laterally through constraints. We extend the physically-based spline models ... to include [muscle] activation....”

Wei Q, Sueda S, Pai DK. Physically-based modeling and simulation of extraocular muscles. *Prog Biophys Mol Biol.* 2010;103(2-3):273–283. doi:10.1016/j.pbiomolbio.2010.09.002

“Dynamic simulation of human eye movements, with realistic physical models of extraocular muscles (EOMs), may greatly advance our understanding of the complexities of the oculomotor system and aid in treatment of visuomotor disorders. In this paper we describe the first three dimensional (3D) biomechanical model which can simulate the dynamics of ocular motility at interactive rates. We represent EOMs using “strands”, which are physical primitives that can model an EOM's complex nonlinear anatomical and physiological properties. Contact between the EOMs, the globe, and orbital structures can be explicitly modeled.....Complex nonlinear EOM physiological properties are incorporated in the computation by explicitly representing the experimental EOM constitutive models.....The model is computationally efficient. It provides a good balance between simulation accuracy and speed. EOM motions can be simulated at interactive rates (~ 16 frames/second) on personal computers....Contact between EOMs and the globe is physically handled through sliding constraints, not only for visualization but also for realistically simulating force interaction and lateral sliding movement....Our biomechanical model can simulate extraocular muscle dynamics, which have not been supported by other 3D biomechanical models.”

The following paper extends these ideas by developing computationally efficient treatments of tendons:

Prashant Sachdeva, Shinjiro Sueda, Susanne Bradley, Mikhail Fain, and Dinesh K. Pai. 2015. Biomechanical simulation and control of hands and tendinous systems. *ACM Trans. Graph.* 34, 4, Article 42 (July 2015), 10 pages. DOI: <https://doi.org/10.1145/2766987>

It follows that the above line of work should be cited and placed in context of the proposed contributions. It is the opinion of the reviewer that the above line of work began with a focus on Cosserat rod models but steered toward a “strand” model that neglects twist (ignores the state of the material frame). As such, there may be an avenue for the present submission to argue for the role of twist in the modeling primitive, and to demonstrate the importance of twist in the quantitative case

studies. Absent such a demonstration, however, it becomes unclear how to differentiate from the above line of work.

We thank the reviewer for bringing the above line of work to our notice. Indeed, we now refer to this series in the updated manuscript to contextualize our work. We would argue that our work is not in contrast or incremental relative to these previous investigations. Instead, it further strengthens the compelling case of modeling complex architectures via 1D models, and complements and extends these early investigations in several important ways. We highlight here what we consider the three most distinctive characters, and further expand on them in the following paragraphs:

1. From a modeling perspective, one major difference, as also pointed out by the reviewer, is the use of dynamic Cosserat rods to fully account for inertia and all six modes of deformation at every cross section (including twisting, shearing and stretching). The importance of this aspect is now demonstrated in additional test cases reported in our detailed response below.
2. In terms of scope and validation, our approach is designed to be generally applicable beyond human musculoskeletal architectures (which is instead the main focus of the series of works suggested by the reviewer). Indeed, we demonstrate the viability of our techniques in a range of applications that include slithering soft robots, bio-hybrid motile devices and full-scale biological organisms.
3. Finally, we demonstrate the use of our simulation framework in conjunction with an evolutionary optimization procedure as a tool not only to improve an existing solution (see section concerning bio-hybrid bots), but also to understand and gain insight into the function of specific musculoskeletal traits, often cluttered by the complexity of living organisms (see section concerning the slithering soft robot and the demonstration of their ingenious natural solution based on overlapping longitudinal actuators). From this point of view, our work illustrates a path to distill and synthesize hidden biomechanical principles by virtue of an inverse design approach.

After clarifying what we consider our main distinctive contributions, we now review more in detail the suggested literature, to highlight important differences. Furthermore, we present two more study cases to show the crucial, quantitative role of shear and twist in biological actuation and thus demonstrate and justify our efforts in including them in our model, as requested by the reviewer.

1. The work that shares the most similarity, in terms of the modeling primitive, is (Pai, 2002). Pai considered Cosserat representations using embedded material frames for muscles, albeit quasi-statically as revealed by equations (1) and (5) in (Pai, 2002), where temporal derivative terms are absent. Moreover, bend and twist were modeled, but not shear and stretch. All these choices are justified in the suture simulation scenario considered in (Pai, 2002). Nevertheless, as we target a broad range of applications and timescales (see sections on fast motions of slithering soft robot and

wings), it is important to additionally capture inertia as well as all mechanical modes, including shear, stretch, twist.

2. Successive works, such as (Pai et al., 2004) and (Sueda, et al., 2008), adopted a specialized spline-based geometric representation of rods, called ‘muscle strands’, having intrinsic elastic properties. This representation is numerically convenient and efficient as well as accurate (Remion et al., 1999). The motivation for such a representation, which is capable of producing forces along the principal-axis (i.e. centerline) of the rod (Pai, 2004) stems from the works of (Zajac, 1989) and (Cheng et al., 2000). These works suggest a scale independent modeling of muscular contractile forces—from the sarcomere to the muscle fiber, to even the entire recruitment group (Zajac, 1989; Winter, 2009)—allowing for the representation of entire muscle fascicles as strands. Again, and importantly, these works above do not explicitly account for the role of shear, twist and stretch (some of these works explicitly constrain rods to be stretch-free). While one can discount such dynamics in the case of human skeletal muscles (citing separation of scales in the strain magnitude), this is not possible in general, heterogenous, living architectures or in other types of human muscles (such as cardiac muscles (Nakatani, 2011) and (Roche et al., 2014), where twist is important).

To further illustrate the importance of shear and twist, and as requested by the reviewer, we included two new studies in the manuscript. In the first one, we consider the case of rehabilitation of an impaired/injured human arm using artificial, assistive muscles tailored to injuries of varying levels of severity. We mimic an injury in our elbow-joint setup of Figure 1 (main text) by impairing the ability of a portion of the biceps to contract (modeled in-silico by rendering some muscle fibers as passive). To regain functionality of the arm, we enhance the impaired biological muscle with an external artificial muscle (shown in Figure S2 of the SI, reproduced below for convenience), modeled as a soft (i.e. stretchable), highly twisted Cosserat filament. This device can generate longitudinal contractile forces by releasing stored twist into stretch and is inspired by the mechanical response shown by the highly-coiled fishing lines of (Haines et al., 2016). Figure S2(a), illustrates the utility of this device. A healthy elbow is shown to be able to lift a weight placed at the end of the radius, while an injured one (50% less contractile stresses) cannot do so. Thus, we apply our artificial muscle (Figure S2(d)) to compensate for the injury and restore the patient’s ability to lift the weight (shown by tracking the joint angles in Figure S2(b)). The tightly coiled muscle generates longitudinal contraction as its internal twist is released (this is experimentally achieved by running a thin electric wire along the soft coiled element: upon heating, the radius of the coil expands, releasing twist which in turn is converted to contraction). As can be seen, by modulating the rate of twist release, different restoring forces can be generated so as to approximately match the patient’s needs—more force is generated the faster we release twist. This is shown in Figure S2 (c), with twist release rate (and hence forces) increasing in the direction of arrows. This example highlights the importance of twist in a setting that involves integration of artificial compliant elements with human biological musculoskeletal architectures.

This, in turn, might prove useful in a number of applications from customized rehabilitation devices to effective, enhancing exoskeletons.

Figure S2: Rehabilitation of a human elbow: (a) While an injury to our elbow joint setup impairs its ability to generate forces and hence lift a weight, it can be compensated for by applying external artificial muscles which provides restoring forces by releasing internal twist. The magnitude of these restoring forces is modulated by the rate of twist release. We show this by plotting the (b) elbow angles and (c) artificial muscle forces for three different release rates (dotted, dashed and solid lines). For reference, the healthy (100% strength) and injured (50% strength) cases are also depicted in the former. (d) Shows the release of twist (black lines, represented by internal curvature k_3), achieved by increasing the filament radius (blue lines) for the case of highest twist release. (Inset) shows the structure of the artificial muscle at different instants.

We have created a new section in the SI detailing the working of the artificial muscle and other specifics of this study, which might be of interest to the reviewer.

The second representative study relates to the kinematics of the full-scale feathered wings. First, we note that we have revised this section to include a more rigorous, quantitative analysis. In particular, we 1) consider a full stroke cycle now; (2) consider available experimental EMG measurements of the wing muscles (Robertson et al. (2012)) as input for our simulations, and (3) characterize the corresponding dynamic wing output via all three joint angles (dorso-ventral, antero-posterior and elbow joint) which are then reported and compared with biological data (Robertson et. al, 2012). Thus, realistic muscular activations, material properties, anatomy and morphology dictate the complex output kinematics (Robertson et. al, 2012). Our numerical approach, capturing all deformation modes, is successful in replicating such

kinematics. These results can be found in the (updated) figure 4 of the main text, shown below for convenience.

Comparison with experimental wing kinematics and muscle activation forces

To test the importance of shear/twist in this setting we perform a numerical hardening experiment wherein we change the material properties affecting shear and twist only, while retaining the original muscular activation and scale, and observe the resulting wing kinematics. If shear/twist is indeed unimportant, we expect to obtain almost identical kinematics. The results of this experiment, achieved by changing the shear modulus (G) of the muscles to 25 times the actual value (hence stiffening twist and shear modes 25 times), is detailed in Fig. S3 which we reproduce below for convenience. Here we have tracked the joint angles of the muscles, which serve as a proxy for the full wing kinematics. As seen from the figure (a–c), the wing kinematics differ significantly, indicating that modeling shear/twist is important in such a biophysical context.

Figure S3: Importance of shear/twist: In our fully feathered wing setup, we probe the importance of capturing shear/twist modes by artificially shear stiffening the wing (changing G to 25 times its original/physical value) and observe the resulting kinematics, by tracking the instantaneous joint angles. The resulting kinematics (shown in dashed red lines) differ significantly from the original (solid red lines) and experimental (shaded black region) measurements.

Once again, these details are added to a new section in the SI, along with the above image.

These two examples motivate the inclusion of shear, stretch and twist physics. The point we wish to highlight then, is that *a priori* knowledge of the anatomy, morphology and mechanics of the muscle in specific cases allows for the adoption of convenient and efficient modeling primitives (Pai, 2002; Pai et al., 2004)—but this specialization is not necessarily applicable in a broader biological context or even more so when artificial and biological components are brought together.

Along the same lines, later works (Sueda et al., 2011; Sachdeva et al., 2015) concentrate on application of muscle-strand-rigid body primitives to constrained motions of musculotendon simulations in the context of graphics applications, using a novel Eulerian-on-Lagrangian approach for algorithmic efficiency. This work is also rooted in un-shearable, un-stretchable, un-twistable rods, while also assuming quasi-static behavior (for a subset of rods). A simplified force-strain response based on physiological data is used for the muscles. The authors motivate this choice based on the fact that the Hill model might suffer from numerical stability issues. We note instead that our elbow joint example is consistent with the Hill model, and we demonstrated its realistic behavior both in static (isometric test) and dynamic (isokinetic test) settings. Such a demonstration is not shown in the line of work of (Sueda et al., 2011; Sachdeva et al., 2015).

3. Lastly, regarding constraints of motion between muscle strands and the environment (be it other muscle strands or rigid objects), the above line of work (Pai et al., 2004; Sueda et al., 2008, 2011) resorts to solving a constrained optimization problem (with inclusion of some additional correction forces, chosen based on representation), which necessitates a sparse-matrix solve. This increases the algorithmic complexity to $O(n^2)$ (maybe even $O(n^3)$ in some cases (Sueda et al., 2008)) for sparse LU factorization. Rather, in our case, we simply use dynamic coupling (similar in spirit to the correction forces above) to include penalty forces, which helps us retain an $O(n)$ method. This issue of scalability is important in faithfully reproducing full scale biological systems entailing thousands of rods and millions of degrees of freedom as in our wing example.

In conclusion, the explanation and examples shown above points to the generality of our modeling approach. Rather than specializing the Cosserat rod theory for one particular scenario (such as human musculotendon systems, as pointed out by the reviewer), we chose to represent generic slender elements that can undergo arbitrary bending, stretching, shearing and twisting to model a spectrum of phenomena, both natural and engineered. This is seen in the varied demonstrations shown in our work—be it in modeling the intricacies of human (and non-human) musculoskeletal systems, designing bio-inspired engineered devices with living components, understanding and distilling operational biomechanical principles of a complex biological system such as a snake and faithful replication of biological systems (shown for a bird wing, from modeling the barbs in a single feather to aggregations of muscles, bones and feathers)—which lends to its novelty.

Once again, we thank the reviewer for helping us place our work in the context of the referred works, and helping us further identify open questions pertaining to the modeling of complex musculoskeletal architectures. We believe that our demonstration of artificial muscle-based rehabilitation of the human elbow joint and the role of shear/twist in a feathered wing provides a compelling argument for the inclusion of twist (and shear) in

the modeling primitive. Following the reviewer's comments, we cite and place our work in context relative to the above line of work in the main text:

Page 3, Paragraph 1, line 4:

"The graphics community has also been active in this space (37–39). Pai and colleagues introduced spline-based muscle-strand primitives for the simulation of various human body parts such as hands or ocular muscles (37,38,40–42). Although numerically efficient, this approach is highly specialized for scenarios in which shear, stretch, twist and dynamic effects are unimportant. Inspired by these and motivated by the need for efficient, robust and more generally applicable solvers to study arbitrary composite and dynamic musculoskeletal systems, we have developed an approach based on assemblies of Cosserat rods (43)"

Page 4, paragraph 2, line 1:

"Here, we apply a more general methodology in which the full dynamics of all deformation modes (bending, twist, shear, stretch) is accounted for. We capitalize on our previous work on Cosserat models and establish a novel musculoskeletal modeling approach to realistically represent active, heterogenous biological layouts, thereby providing new opportunities to realistically represent and unravel their functioning, actuation and control strategies."

As in the present submission, earlier works have also sought to find a compromise between expensive 3D finite elements and reduced models. For instance:

Ye Fan, Joshua Litven, and Dinesh K. Pai. 2014. Active volumetric musculoskeletal systems. ACM Trans. Graph. 33, 4, Article 152 (July 2014), 9 pages. DOI: <https://doi.org/10.1145/2601097.2601215>

We thank the reviewer for referring us to the above article, which we have added to our list of references. The above work focuses on efficiently simulating musculoskeletal systems with high volumetric deformations in which contacts and constraints can be easily modeled. Discretization was done on a Eulerian grid, and a deformation map was used to track musculoskeletal motions. An activation map was used to achieve graphically realistic muscle activations. Simulations of ocular muscles, general soft tissues and a boxer arm in this work prove that the algorithm captures realistic muscle motion and deformation, although a verification against physiological actuation is missing, as acknowledged by the authors themselves:

Page 1, Section 1, Paragraph 2:

"Muscles are controlled by neural activation, but representing the behavior of active muscles remains challenging, despite a century of progress in muscle biomechanics. For example, the widely used "Hill-type" models [Zajac 1989] can have large errors (about 50%), even in highly controlled laboratory settings [Herzog 2004]. The behavior of active muscles in vivo is even less understood, due to motor unit diversity within muscle, and other issues such as motor unit recruitment, cycling, bistability of firing rates, etc."

Page 3, Section 4, Paragraph 1:

“When an active muscle contracts, there is an internal reorganization of its mass mediated by molecular motors which cause myofilaments to slide relative to each other. As mentioned in Sec. 1, the actual dynamics of active muscle is complex and not very well understood. Worse, there are significant differences between the properties of different muscles and between individuals. Rather than attempt to model these details, we seek a practical but realistic activation model for use in graphics.”

Page 8, Section 9, Paragraph 1:

“Our muscle activation model is clearly a major simplification, especially for partial activation, but as mentioned in Sec. 1, the alternatives have their own problems. A muscle’s behavior is influenced by its fiber architecture, which is not modeled explicitly in our approach. We chose not model the architecture because the required data are extremely difficult to obtain; most previous work in this area (e.g., [Agur et al. 2003]) are not subject-specific and required painstaking cadaver dissections. More recent work could produce subject specific architecture in vivo using MRI (e.g., [Levin et al. 2011b]), but this work is at an early stage”

A direct one-to-one comparison between the approaches adopted in our work and the work above (along with the closely-related works of (Fan et. al, 2013, Levin et. al, 2011)) would not be very informative as the modeling approaches and the examples presented are very different (i.e. pure Eulerian vs. pure Lagrangian). Nevertheless, we add that for the cases involving human muscles (a central theme of both the works), our Lagrangian approach is natural for modeling and activation purposes as it directly stems from the fibrous nature of muscles. This is also recognized by the authors of the referred work, who aim at *“practical but realistic”* muscle activations for graphics. In any case, such Eulerian-grid based deformation map techniques present computational advantages and can potentially be integrated with Lagrangian approaches (leading to a mixed Eulerian-Lagrangian algorithm), opening an interesting avenue for future work.

The case studies presented in the submitted work are interesting and varied. It is not clear whether each case study has been fully contextualized vis a vis the available literature. For instance, the following paper modeled the dynamics of flagellar propulsion using a Cosserat rod model:

M. K. Jawed, N. K. Khouri, F. Da, E. Grinspun, and P. M. Reis. Propulsion and Instability of a Flexible Helical Rod Rotating in a Viscous Fluid. Phys. Rev. Lett. 115, 168101. 2015.

We thank the reviewer for referring us to this article, which we have now added to the list of our references. Based on the feedback of other reviewers, we have further expanded the context relative to the wing and snake cases.

Finally, we thank the reviewer for her/his careful evaluation and critical comments which helped us in improving our manuscript significantly. We hope the reviewer finds our answer satisfactory and the revised manuscript suitable for publication.

References used in this response:

- E. J. Cheng, I. E. Brown, and G. E. Loeb. Virtual muscle: a computational approach to understanding the effects of muscle properties on motor control. *Journal of neuroscience methods*, 101(2):117–130, 2000.
- Y. Fan, J. Litven, D. I. Levin, and D. K. Pai. Eulerian-on-lagrangian simulation. *ACM Transactions on Graphics (TOG)*, 32(3):22, 2013.
- Y. Fan, J. Litven, and D. K. Pai. Active volumetric musculoskeletal systems. *ACM Transactions on Graphics (TOG)*, 33(4):152, 2014.
- C. S. Haines, N. Li, G. M. Spinks, A. E. Aliev, J. Di, and R. H. Baughman. New twist on artificial muscles. *Proceedings of the National Academy of Sciences*, 113(42):11709–11716, 2016.
- D. I. Levin, J. Litven, G. L. Jones, S. Sueda, and D. K. Pai. Eulerian solid simulation with contact. *ACM Transactions on Graphics (TOG)*, 30(4):36, 2011.
- D. Li, S. Sueda, D. R. Neog, and D. K. Pai. Thin skin elastodynamics. *ACM Transactions on Graphics (TOG)*, 32(4):49, 2013.
- S. Nakatani. Left ventricular rotation and twist: why should we learn? *Journal of cardiovascular ultrasound*, 19(1):1–6, 2011.
- D. K. Pai. Strands: Interactive simulation of thin solids using cosserat models. In *Computer Graphics Forum*, volume 21, pages 347–352. Wiley Online Library, 2002.
- D. K. Pai, D. I. Levin, and Y. Fan. Eulerian solids for soft tissue and more. In *ACM SIGGRAPH 2014 Courses*, page 22. ACM, 2014.
- D. K. Pai, S. Sueda, and Q. Wei. Simulation of 3d neuro-musculo-skeletal systems with contact. In *Advances in Computational Motor Control III*, Symposium at the Society for Neuroscience Meeting, San Diego, CA, 2004.
- Y. Remion, J.-M. Nourrit, and D. Gillard. Dynamic animation of spline like objects. In *Proceedings of the WSCG'1999 Conference*, pages 426–432. Citeseer, 1999.
- M. B. Robertson and A. A. Biewener. Muscle function during takeoff and landing flight in the pigeon (*columba livia*). *Journal of Experimental Biology*, 215(23):4104–4114, 2012.
- E. T. Roche, R. Wohlfarth, J. T. Overvelde, N. V. Vasilyev, F. A. Pigula, D. J. Mooney, K. Bertoldi, and C. J. Walsh. A bioinspired soft actuated material. *Advanced Materials*, 26(8):1200–1206, 2014.
- P. Sachdeva, S. Sueda, S. Bradley, M. Fain, and D. K. Pai. Biomechanical simulation and control of hands and tendinous systems. *ACM Transactions on Graphics (TOG)*, 34(4):42, 2015.
- S. Sueda, G. L. Jones, D. I. Levin, and D. K. Pai. Large-scale dynamic simulation of highly constrained strands. *ACM Transactions on Graphics (TOG)*, 30(4):39, 2011.
- S. Sueda, A. Kaufman, and D. K. Pai. Musculotendon simulation for hand animation. *ACM Transactions on Graphics (TOG)*, 27(3):83, 2008.
- Q. Wei, S. Sueda, and D. K. Pai. Physically-based modeling and simulation of extraocular muscles. *Progress in biophysics and molecular biology*, 103(2-3):273–283, 2010.
- D. A. Winter. *Biomechanics and motor control of human movement*. John Wiley & Sons, 2009.
- F. E. Zajac. Muscle and tendon: properties, models, scaling, and application to biomechanics and motor control. *Critical reviews in biomedical engineering*, 17(4):359–411, 1989.

Response to Reviewer 3:

Modeling and simulation of complex dynamic musculoskeletal architectures

Xiaotian Zhang¹, Fan Kiat Chan¹, Tejaswin Parthasarathy¹ and Mattia Gazzola^{1,2†}

¹Mechanical Sciences and Engineering, University of Illinois at Urbana-Champaign, Urbana, IL 61801, USA

²National Center for Supercomputing Applications, University of Illinois at Urbana-Champaign, Urbana, IL 61801, USA

† To whom correspondence should be addressed; E-mail: mgazzola@illinois.edu.

We thank the reviewer for her/his valuable time, consideration and largely positive assessment. In the following, the comments of the reviewer are listed followed by our responses. All modifications to the manuscript are highlighted in red for the reviewer's convenience.

We hope that the reviewer considers our answers acceptable and the improved manuscript is suitable for publication.

This paper introduces a new way of modeling musculoskeletal movements in animals, using Cosserat rods. As a biologist, I am not a mechanic, and am unable to properly evaluate the full relevance and validity of the mechanical modeling (which forms the underlying basis of the paper). That said, this manuscript was quite exciting to read, as the described tools could be broadly used to tackle innumerable problems in biomechanics, opening up new lines of investigation. It really does seem like this method could revolutionize the field, because biologically-accurate models can be used to test questions that can't be easily addressed with experimental methods, or can compliment those methods.

Despite my enthusiasm, there are a number of issues both large and small before I can recommend publication.

We are grateful to the reviewer for her/his enthusiasm in the current work and for recognizing its novelty and the potential contribution it can offer to the relevant fields. We have deployed a substantial effort to fully address the reviewer's comments, with special focus on the quantitative comparison with biological kinematics for the cases of snake slithering and flapping wings. Our answers are detailed in the following.

My major concern involves the non-human animal models in the study. From my perspective, to be convincing the models should closely match the quantitative kinematics of movement of the real biological system. I'm less concerned about matching external forces, which can add a far greater level of complication. This concern arises because for both the snake model and the wing model, the kinematics don't look very realistic. The snake model looks to me like the roundworm *C. elegans*. I believe that the new approach in this manuscript could indeed produce snake-like behavior, but this isn't it. The wing model certainly doesn't match a full stroke, in which flexion at the joints (driven by muscles in each segment) plays a large role. Furthermore, I don't understand the modeling of the lift and drag forces, both of which in a takeoff involve unsteady aerodynamics. Drag was modeled based on the drag coefficient of isolated cylinders, which is certainly not appropriate when the cylinders are close together in the form of a feathered wing. Either stick to the kinematics, or work with a fluid mechanist to model the three-dimensional unsteady flows (which I would not recommend, given the large effort that would take).

We appreciate the reviewer's comment on the importance of further quantification to comment on the ability of our solver. Following the reviewer's suggestion, we now provide a detailed comparison with biologically occurring kinematics in both the snake and flapping wing cases. We break down the reviewer's comment into two parts—snake and wing model—and address them separately as detailed below. We hope that the reviewer finds the improved manuscript satisfactory.

This concern arises because for both the snake model and the wing model, the kinematics don't look very realistic. The snake model looks to me like the roundworm C. elegans. I believe that the new approach in this manuscript could indeed produce snake-like behavior, but this isn't it.

Here we should have better specified the biological conditions our snake study pertains to. Indeed, as correctly pointed out by the reviewer, snakes exhibit a variety of gaits, many of which are dissimilar from the one reported in our manuscript. Nonetheless, this can also be found in nature and has been reported and characterized in (Hu et al., 2012), from which we extracted the image below. That work considers juvenile milk snakes, and experimentally measures and reports the kinematics relative to different behaviors for the same specimen. We note that snake locomotion in these experiments are characterized by the Froude number $Fr \sim 0.1$ as in our numerical experiments. Moreover, in our revised version, not only do we maintain a biologically realistic Froude number but also employ experimentally measured friction coefficients and animal lengths (Hu et al., 2009), to prevent any unintended numerical artifact related to rescaling.

As can be noted, gaits (b) and (c) corresponding to "leisure" locomotion (as termed by (Hu et al., 2012)) on a rough and smooth surface, respectively, look quite distinct from the "C.elegans" type gait mentioned by the reviewer and reported in our manuscript. Nonetheless, gait (a) corresponding to a *sprinting snake* on a rough surface looks remarkably similar to our simulations. To aid visual comparison we digitized the gait kinematic envelope and now report it in Fig. 3 of the main manuscript (and below for reference). We emphasize that the sprinting behavior corresponds to the target of our

optimization procedure. Indeed we evolved gaits that *maximize speed*, and without any further constraints, the optimizer finds that the optimal computational solution closely resembles the sprinting snake of (Hu et al., 2012).

The optimizer however does not tell us why this gait is optimal for speed or why it is not encountered in other behavioral settings, which presumably operate according to different tradeoffs (for example, a “leisure” gait might be more energetically efficient). This is beyond the scope of this work, although it represents an avenue of future research.

We updated the figure caption, and the text surrounding the image to reflect this new comparison/agreement with experimental data:

Page 17, Figure 3 caption (main text):

“(e) Comparison of gaits and lateral displacements between the continuous reference (61), our musculoskeletal model and experiments (77) for “sprinting” snakes.”

Page 17, paragraph 1, line 5 (main text):

“This stems from the limited number of muscles, and is reflected in the more prominent lateral displacement of the midline kinematics (Fig. 3e). We additionally show experimentally recorded midline gaits of a “sprinting” milk snake (77) ($Fr = 0.1$) which, when locomoting at high speed (sprinting), closely resemble our optimally fast model. It is remarkable how the careful orchestration of distributed actuation (four longitudinal muscle groups) allows for smooth realistic gaits despite its simplicity.”

We thank the reviewer for encouraging us to improve context and compare against real snake centerline kinematics, which has also opened up interesting questions about the

optimality of such gaits. We believe this comparison makes yet another argument for our modeling approach of animal musculoskeletal architectures and its combination with evolutionary optimization strategies.

The wing model certainly doesn't match a full stroke, in which flexion at the joints (driven by muscles in each segment) plays a large role. Furthermore, I don't understand the modeling of the lift and drag forces, both of which in a takeoff involve unsteady aerodynamics. Drag was modeled based on the drag coefficient of isolated cylinders, which is certainly not appropriate when the cylinders are close together in the form of a feathered wing. Either stick to the kinematics, or work with a fluid mechanist to model the three-dimensional unsteady flows (which I would not recommend, given the large effort that would take).

We agree with the reviewer that this section deserves a more rigorous treatment, and a better kinematic characterization to highlight the capability of our solver. Towards this we now report a much more comprehensive study: (1) A full stroke cycle is now considered; (2) We now consider available experimental EMG measurements of the wing muscles (Robertson et al. (2012)) as input for our simulations, and (3) the corresponding dynamic wing output is now characterized via all three joint angles (dorso-ventral, antero-posterior and elbow joint) which are then reported and compared with biological data. Therefore, in our model, simulated muscles—biceps (BC), scapulotriceps (ST), pectoralis (PECT) and supracoracoideus (SUP) (Fig. 4b, reproduced here for reference)—contract according to the EMG time sequences and with peak forces similar to measurements found in Biewener et al. (1998) (Fig. 4f). As can be seen in the updated Fig. 4d (reproduced here for convenience), when our virtual muscles are actuated in a biologically realistic fashion, the wing as a whole recovers its natural kinematics (Fig. 4d)

We stress that this is not a trivial feat. Indeed, (1) the wing is a highly non-linear system and (2) we note that the four shoulder muscles in our simulation control only two joint angles during flexion–extension: supracoracoideus–pectoralis pair controls the dorso-ventral angle of the shoulder and the biceps–scapulotriceps pair controls the elbow angle. The temporal undulation of antero–posterior angle is therefore the direct outcome stemming from the dynamic interaction between the active muscles and the wing structure. The obtained full stroke kinematics are now reported in Fig. 4e, illustrating a good level of quantitative fidelity relative to biological data. The main discrepancies are observed for the antero-posterior angle, which is not surprising as we have no direct control on it and entirely depends on the structure's passive response. Nonetheless, the fact that we do capture its overall dynamics (maximum deviation of the antero-posterior angle with experiments is ~10 degrees, comparable to measurement variations) is still remarkable especially considering the challenges related to the lack of consistent biological datasets. Indeed, although feathered flapping flight has been extensively studied, the reported kinematics, muscle actuation and morphological data are often disjointed as they might refer to different species/specimens and/or flight conditions. For example, accurate morphological measurements could be found for certain species, while careful kinematics may be reported for others. Given the challenging data landscape, we

chose to take a pragmatic and logical approach and tried to build a model as consistent as possible by blending together the most closely related datasets we could find. Nonetheless, despite these uncertainties our modeling approach was able to qualitatively and quantitatively capture the most salient kinematic features.

We thank the reviewer for suggesting a sharper focus on kinematics, which led to the improved section and (we believe) a more convincing case on the capacity of our methodology in replicating realistic gait dynamics. We updated the text and figure in the wing section to reflect this new comparison.

Page 18, paragraph 2, line 14 (main text):

“In our four-muscle model, the supracoracoideus–pectoralis pair controls the dorso-ventral angle of the shoulder and the biceps–scapulotriceps pair controls the elbow angle during flexion–extension. The temporal evolution of antero-posterior angle then arises from the dynamic interaction between the structure and the environment.”

Page 20, Fig. 4 caption (main text):

“(d) Initiation process that lifts the wing from flat position, followed by a single power downstroke and upstroke during the takeoff stage. (e) Joint angle measurements (simulation vs. experiments (80)) for elbow, dorso-ventral and antero-posterior angle. (f) Actuation patterns for four different muscles: biceps (BC), scapulotriceps (ST), pectoralis (PECT), and supracoracoideus (SUP). EMG recordings (80) are represented in black and our simulated muscle activity are represented in red.”

Page 19, paragraph 2, line 1 (main text):

“We then set to reproduce the kinematics of wings morphing through a full stroke cycle during takeoff mode. We first initialize our simulated wing in a straight, flat configuration (Fig. 4d) and over the initiation phase, set (arbitrarily) the muscle activation via Eq. S6 so as to prepare the wing for the downstroke phase (Fig. 4e,f). During the downstroke and upstroke phase, the muscle actuation patterns (Eqs. S7–10) are instead based on experimentally recorded electromyography (EMG) signals (80) (Fig. 4f). Since EMG measurements do not provide the magnitude at which the muscles operate (only their time sequences), we set the muscle actuation force (~ 25 N) compatibly with the forces reported in (86) (SI). As can be seen in Fig. 4e, our model realistically captures the temporal evolution of the three joint angles, in agreement with experimental measurements (80). This is a non-trivial task given the highly non-linear interplay between muscle actuation, passive structural dynamics and aerodynamic loads. The main discrepancy is observed for the antero-posterior joint angle. This is not surprising since the four muscles do not directly affect it and its time evolution emerges as a result of the overall system dynamics. Thus, this is the angle most sensitive to modeling approximations. In this context, it is still notable that our simulations can capture it qualitatively, and that the maximum deviation from experiments amounts to ~ 10 degrees, comparable to measurement variations.”

Relative to the aerodynamics forces, we removed the original computation of the total lift and drag forces generated by the wings, as suggested by the reviewer. Nonetheless, we feel that completely ignoring fluid forces would be even more of an approximation than employing a simple, low order model in which loads scale quadratically with the local velocity \mathbf{v} . At least this model captures the presence of the fluid and inherent non-linearities (through \mathbf{v}^2) at moderate speeds (as in a biological case), thus providing a preliminary, although crude, estimate. We note that upon actuation based on biologically realistic muscle forces and wing morphology, we recover realistic wing kinematics, which would have been difficult if the fluid was entirely misrepresented. We do acknowledge, though, that this is an important aspect to be improved on and it will be the subject of

further research. We thus added a sentence stating this choice, while clearly highlighting its limitations.

Page 19, paragraph 3, line 16 (main text)

“Aerodynamic loads are estimated via a reduced order model in which forces scale quadratically with the local body velocity (SI). While this model cannot capture the complex unsteady aerodynamics associated with flapping flight, it nonetheless provides a preliminary estimate.”

Other general issues:

We thank the reviewer for her/his diligent and careful critique on the current work. We address the reviewer’s comments on other general issues as follow.

1. *Note that throughout the manuscript, the construction “the Hill’s model” is used. Hill’s model indeed is widely known and employed, but with an article, it should be termed “the Hill model”.*

We thank the reviewer for pointing this out. In order to also address comment (41), we refer throughout the manuscript to “the Hill three-element model” or simply “the Hill model”.

2. *“Patient-specific needs” are mentioned in several places. I don’t understand what this means in the context of the work done in this paper, which focuses on biological models. What patients, and for what?*

We apologize for this tangential comment. We now substantiate the claim that our approach can prove useful in addressing patient-specific needs by providing a new study case (in the SI) in which we illustrate how an elbow injury—damaged biceps brachii—can be compensated for via artificial muscles tailored to injuries of varying levels of severity. The simulated injury impairs the ability of a portion of the biceps to contract. To regain functionality of the arm, we enhance the impaired biological muscle by attaching an external artificial muscle (shown in Figure S2 of the SI, reproduced below for convenience), modeled as a soft (i.e. stretchable), highly twisted Cosserat filament. This device can generate longitudinal contractile forces by converting twist into stretch and is inspired by the work of (Haines et al., 2014, 2016) on the mechanical response of tightly coiled fishing lines. The figure below (and Movie 7), illustrates the utility of this device. A healthy elbow is shown to be able to lift a weight placed at the end of the radius, while an injured one (50% less contractile stresses) cannot do so. Thus, we apply our artificial muscle (Fig. S2d) to compensate for the injury and restore the patient’s ability to lift the weight. The tightly coiled muscle generates longitudinal contraction as its internal twist is released (this is experimentally achieved by running a thin electric wire along the soft coiled element—upon heating, the radius of the coil expands, releasing twist which in turn is converted to contraction). As can be seen, by modulating the rate of twist release,

different restoring forces can be generated so as to approximately match the patient's needs.

Figure S2: Rehabilitation of a human elbow: (a) While an injury to our elbow joint setup impairs its ability to generate forces and hence lift a weight, it can be compensated for by applying external artificial muscles which provides restoring forces by releasing internal twist. The magnitude of these restoring forces is modulated by the rate of twist release. We show this by plotting the **(b)** elbow angles and **(c)** artificial muscle forces for three different release rates (dotted, dashed and solid lines). For reference, the healthy (100% strength) and injured (50% strength) cases are also depicted in the former. **(d)** shows the release of twist (black lines, represented by internal curvature k_3), achieved by increasing the filament radius (blue lines) for the case of highest twist release. (Inset) shows the structure of the artificial muscle at different instants.

We have created a new section in the SI detailing the working of the artificial muscle and other specifics of this rehabilitation study, which might be of interest to the reviewer. We also refer to it in the main text to clarify our statement regarding “patient-specific needs”.

3. Change “leveraged” to “used”. I appreciate that “leveraged” is a widely used term in engineering (particularly in proposals, and often poorly used), but given that this manuscript has heavy mechanics content and leverage has a specific mechanical meaning, it's better to not use this jargon and use a simpler, clearer word.

We agree that “leverage” has indeed been generously used. We have updated the text to reflect the updated statement by replacing “leveraged” with “used” throughout the

manuscript.

4. Spell out FEM.

We have updated the text to spell out in full Finite Element Method.

5. It is a little odd that one of the main muscles modeled in this paper is misspelled throughout the manuscript. Biceps brachii should be biceps brachii.

We apologize for the misspelling of biceps brachii. We have fixed the spelling throughout the manuscript.

6. “with each rod representing motor units.” How many motor units? How well does this match the human muscle?

We apologize for our terse style intended to avoid diluting the main text. We now refer the reader to the Supplementary Information for more details on the modeling of biceps brachii muscle, which has been updated to incorporate the reviewer’s comment.

Page 5, paragraph 4, line 1 (main text):

“We reproduce in-silico the biceps brachii (Fig. 1b, orange elements) modeled as a bundle of viscoelastic Cosserat rods or filaments with each rod representing motor units (SI).”

In the Supplementary Information, we now elaborate in detail the muscle modeling approach, where each head of the biceps brachii is modeled as a bundle of 18 Cosserat rods with each representing 20 motor units. This in total provides each head with 360 motor units, which is consistent with the human muscle within the estimated range for number of motor units in the biceps brachii as reported in Brown (1988). While the number of motor units is a physiological, well-defined quantity, their grouping into 18 contractile model filaments that can be autonomously recruited is a modeling approximation. The level of course-graining can be adjusted as needed and can even match all 360 physical units, although this will increase computational costs.

Page ii, paragraph 2, line 1 (Supplementary Information):

“The replication of the two-headed muscle of an elbow joint, the biceps brachii (Fig. S1a), may provide an intuition of our muscle modeling approach. Each head is made of 360 motor units based on (9) and equally-distributed among $N = 18$ filaments. While the number of motor units is a physiological, well-defined quantity, their grouping into 18 contractile model filaments that can be autonomously recruited is a modeling approximation. The level of course-graining can be adjusted depending on the desired

level of detail and can even match all 360 physical units, although this will increase computational costs.”

7. “This allows the muscle to comply with the size principle (49), for which low-force, slow-twitch are associated to small motor units, while high-force, fast-twitch to large ones (see Supplementary Information).” Fix the grammar errors here.

We have now fixed the grammar.

8. Figure 1: Explain what the abbreviations and symbols stand for in the figure legend. The reader shouldn't have to hunt these down in the main text to figure out what they are.

We have updated the legends Fig. 1 (reproduced here for reference) as well as the caption to reflect more details on the data presented and improve readability.

Page 6, Fig 1 caption (main text):

“Fig. 1. Validation of the approach. (a) Elbow anatomy. (b) Simulation of an elbow composed of three bones (humerus, ulna and radius) and two heads of biceps (short and long head) performing a complete flexion. (c) Experimental data (56) and simulations for active and passive force normalized with peak force (F_m/F_{max}) during isometric exercise (F_{set} mimics the resistance encountered by the muscle and results in its equilibrium length η). (d) Experimental (57) and simulation torque measurements of elbow joint (angled at 60°) performing maximal isokinetic concentric flexions at different angular velocities along with its corresponding damping coefficient ζ . The numerically determined ζ (SI) are then compared with theoretical estimates based on the Hill model (58).”

9. “a fixed joint (element connection that restricts any relative motion) is used

for the ulna–radius connection.” I’m not sure why this is implemented. In a real human forearm, these two bones are free to twist relative to one another. If they are fixed, then there is no purpose of including both bones; only one element is needed for the model.

We agree with the reviewer that in a real human forearm, the ulna and radius are free to twist relative to each other. In general, the elbow and forearm have two degrees of freedom: flexion–extension for the lifting of the forearm and pronation–supination for the rotation of the forearm about its longitudinal axis (figure adapted from Moriwaki et al. (2009) shown below for context).

In our validation study of the elbow joint, and in particular for the activity of elbow flexion–extension as conducted in Rubenson et al. (2012) and Gauthier et al. (2001), no relative motion of ulna–radius is involved (hence, no pronation–supination is present). Therefore, we chose a pragmatic approach to constrain the ulna–radius connection with a fixed joint for simplicity as well as for faithful reproduction of the experimental setup for comparison. As the reviewer pointed out, it is true that there is no purpose of including both bones in the case of a fixed joint. However, our framework is meant to be generally applicable and reconfigurable via the introduction of different boundary conditions for the assembly of these bones. By redefining the joint connection, our framework can be readily used to study the pronation–supination movement for the forearm.

We thank the reviewer for pointing this out and have updated the manuscript to reflect the rationale behind the constraints between these two bones as well as highlighting the readiness of our framework in dealing with all modes of elbow motion.

Page 7, paragraph 2, line 6 (main text):

“For the mimicking of pure flexion–extension exercises (as intended here), we do not consider the relative rotation between ulna and radius which occurs during pronation–supination. However, these movements can be modelled by redefining the joint connection to allow for rotations in two perpendicular planes.”

10. “We then performed isometric (static) and isokinetic (dynamic) tests for validation against experiments.” Whose experiments?

We compared our isometric (static) data against experimental data from Rubenson et al. (2012) and compared our isokinetic (dynamic) data against experimental data from Gauthier et al. (2001). The manuscript has been updated to reference the relevant work.

Page 7, paragraph 3, line 1 (main text):

“We then performed isometric (static) and isokinetic (dynamic) tests for validation against experiments (56, 57).”

11. “The simulation data is obtained when the muscle reaches its final equilibrium length η (Fig. 1c).” I don’t understand. If this is an isometric simulation, then the muscle’s length should not be changing.

The reviewer is correct in that physically, in an isometric test, the muscle length remains constant. This is achieved experimentally by repeating the test with the handle in different positions. Each position dictates the muscle length. The handle does not move, thus providing an external force equal and contrary to the one produced by the muscle. In order to obtain the equilibrium muscle length η when the muscle is producing a certain output force F_m/F_{max} , we initialize our simulated muscle at the rest length $\eta = 1$ and prescribe an external force F_{set} representative of the external force (non-moving handle) counteracting the contraction force output during maximum voluntary contraction (MVC). The simulation then dynamically evolves the system towards its equilibrium static state (where F_{set} balances out F_m/F_{max}) to arrive to the equilibrium muscle length η . By repeating this exercise for various values of F_{set} , we reconstruct the Force–Length relation of Fig. 1c, which characterizes and validates the behavior of our simulated muscle during isometric exercise. We have updated the manuscript to clarify the validation procedure.

Page 7, paragraph 3, line 6 (main text):

“To perform the test in silico, we use available experimental data (Fig. 1c) to compute polynomial fittings that dictate the muscle active MVC and passive elastic response (Fig.

1c) as functions of its elongation η (SI). Once these biomechanical properties are determined, we let the muscle (initialized at rest length $\eta = 1$) perform its MVC while applying prescribed external forces $F_{set} \leq F_{max}$ at its ends. The simulation then dynamically evolves the muscle to its static equilibrium length η . By repeating this experiment for various F_{set} , we can relate muscle length to static force output. This relation is illustrated in Fig. 1c, confirming good match between simulations and experiments.”

Page v, paragraph 1, line 3 (Supplementary Information):

“In order to perform the test in-silico, we initialize our simulated muscle at rest length ($\eta = 1$) and prescribe different external forces F_{set} (inset of Fig. 1c). The simulation then evolves the biomechanical system dynamically from $\eta = 1$ to its equilibrium state (i.e. until the muscle no longer changes in length). The output of the simulation—final equilibrium length of the muscle—is then measured. We can then reconstruct the overall muscle Force-Length curve, which provides a good approximation to the experimental muscle behavior (Fig. 1c).”

12. Change “discount” to something more appropriate. “decrease”?

We modified the text accordingly.

13. “increase at 90 degrees angle” Fix grammar.

We have updated and fixed the grammatical error.

14. Where does equation 1 come from?

In order for our swimmer to interact with the environment, we need to couple rods’ mechanics with a model that captures the forces on the swimmer. Our swimmer operates in the Stokes flow regime, where the Reynolds number is low ($Re \sim 10^{-2}$ in this case). The low Reynolds flow condition combined with the slenderness of our swimmer permits the use of slender-body theory in approximating the generated hydrodynamic loads on the body. Equation 1 arises from the asymptotic expansion in terms of radius-to-length ratio, and is formally derived in Cox (1970).

We have updated the manuscript to refer the reader to the Supplementary Information for details on slender body theory and include references to work relevant to the theory.

15. “Covariance Matrix Adaptaion-Evolution Strategy algorithm” Fix spelling.

We have now fixed the misspelling.

16. “CMA-ES has proven reliable over a range of engineering and biophysical problems (54, 55), from fish swimming and schooling (54, 56) to aircraft wake dissipation strategies (57).” Proven reliable in doing what?

We now clarify this point. The algorithm has a long history of successfully dealing with multi-modal (typically encountered in non-linear mechanics applications) and low-dimensional continuous problems. This has been extensively demonstrated in the works cited in the manuscript which exemplify how CMA-ES has been used for optimality studies in a range of applications and physical settings. Moreover, CMA-ES has been long setting the standard in the GECCO benchmark suite (Hansen et al. (2009), Hansen et al. (2010) and Nguyen et al. (2017)). This tests the success rate (i.e. the average probability) of a given optimization algorithm on a number of problems with known solution and characterized by different mathematical properties (multi-modal, non-separable, noisy, etc.). The CMA-ES has been consistently unbeatable in the past decade. We now briefly highlight this in the revised manuscript.

Page 11, paragraph 1, line 2 (main text):

“While there is no mathematical proof of convergence to global optimum, CMA-ES has proven reliable in dealing with multi-modal, low-dimensional continuous problems (66, 67) and has been employed in a range of engineering and biophysical applications (68–71).”

17. “244% improvement in swimmer’s maximum speed.” Fix grammar.

We have fixed the grammar in the manuscript.

18. “with length and radius of 0.19 mm and 32.3 μm , respectively.” Use consistent units.

We apologize for the inconsistency in units. The manuscript has been updated as suggested.

Page 11, paragraph 3, line 2 (main text):

*“The optimal design requires a shorter but wider head, with length and radius of **190 μm** and 32.3 μm , respectively.”*

19. “for the next generation bio-hybrid swimming robot design.” Add the word “of”.

We have updated the manuscript as suggested.

20. “leading to the fabrication and testing of the largest and fastest biobot to date (19).” I don’t know the records, but it seems that a 0.7 mm biobot is not very big. And, what is a biobot?

We apologize for the loose usage of the term “biobot” which is now replaced with “bio-hybrid robot” or “biological machines”. Bio-hybrid robots in general refer to machines that combine biological elements (muscle cells or neurons) with artificial scaffolds, typically to achieve locomotion. In the introduction we briefly introduce this concept.

Page 2, paragraph 2, line 5 (main text):

“More recently, a radically new breed of soft bio-hybrid robots (3,20–23) that combine biological muscles and sensors with artificial scaffolds has been emerging, paving the way to engineer living machines with unique abilities of self-assembly, healing, growth and adaptivity. This technology carries the promise of high impact applications, from biomedicine to manufacturing (24), and of fundamental discovery as it provides a platform to test hypotheses related to the functioning of living organisms (25).”

These robots are generally small (mm scale) due to the limited amount of force that *in vitro*, stem cells derived muscle can produce. Because of this, the scaffold design and mechanics is critical. Reference (19) does not refer to the swimming bot, but to the walker which is ~14 mm long (contrary to 1.4 mm mentioned in the original submission, which was a typing error). To the best of our knowledge, this is the largest reported to date. We have updated the manuscript as follow:

Page 12, paragraph 2, line 1 (main text):

*“In addition to modeling and optimizing a bio-hybrid swimmer, we also tackled the computational design of a bio-hybrid walker, leading to the fabrication and testing of the largest and fastest **motile biological machine (biobot)** to date (21).”*

21. “Inheriting from previous demonstrations (20),” Inheriting what?

We meant to refer to the robot physical design (scaffold geometry and muscle attachment) of Raman et al. (2016). We have revised the manuscript as follow.

Page 12, paragraph 1, line 1 (main text):

*“Inheriting the bio-hybrid robot design from previous demonstration (22), the walker of (21) consists of an asymmetric hydrogel scaffold and skeletal muscle tissues, resembling muscle–tendon–bone relationships found *in vivo*.”*

22. “1.4mm” Always use a space between number and unit (except for temps), here and throughout the manuscript.

We thank the reviewer for pointing this out, which brought to our attention the error made in specifying the dimension of the bot in original submission (14 mm instead of 1.4 mm). We have fixed this accordingly.

Page 12, paragraph 1, line 5 (main text):

*“We modeled this architecture and, targeting a bot length of **14 mm** which is approximately twice the previous largest attempt, used our simulations to design the new scaffold and topological muscle arrangement of the bot.”*

23. “used our simulations to design and dimension the new scaffold” Is scale not part of design? Extra verbiage not needed.

We have revised the manuscript accordingly.

24. For equation 2, the units don’t seem to work out properly.

We have double-checked the consistency of the units in Equation 2 and we believe they are correct. We work out its units below. Given

$$F_m = A_m \left(\gamma \sigma_m - \frac{E_m \epsilon}{1 - \epsilon} \right)$$

force F_m has the units of Newtons [N]. The muscle tissue cross-sectional area A_m is measured in $[m^2]$ and $\gamma = A_{act} / A_m$ has no units. The muscle contractile stress σ_m has units of $[N/m^2]$ and Young’s modulus of the scaffold has units of Pascals $[N/m^2]$. Strain ϵ has no units. Therefore, both terms $\gamma A_m \sigma_m$ and $A_m E_m \epsilon / (1 - \epsilon)$ result in units of Newtons, which is consistent with the output units we expect—force F_m .

25. Why does “manufacturability” matter in this paper, which concerns simulations?

We can indeed simulate a range of architectures, even structures that cannot be physically fabricated with current technology. Nonetheless, we believe that any computational engineering framework must take into account the experimental conditions and constraints (including fabrication) of the system it aims to model. All too often computational efforts quickly diverge from reality, limiting their impact and utility. For this reason, throughout the paper, we tried to validate our methods and, whenever possible, test their predictive capacity by fabricating and testing our designs. This is the reason why

we were careful to account for manufacturability constraints (for example, appendages that are too thin cannot be produced).

26. “the physics of cell-powered soft robotic systems” I think “muscle-powered” is intended.

The reviewer is correct in that “muscle-powered” should be used when referring to the walker robot while “cell-powered” is the more appropriate term for the swimmer demonstration. We have then updated the manuscript to make more explicit the intended concluding statement.

Page 13, paragraph 2, line 1 (main text):

*“We have shown through these studies that our computational approach is able to accurately capture the physics of **cell- and muscle-powered soft robotic systems** and further optimize their design for desired performance.”*

27. “Extensive work has been done on understanding snake locomotion (12, 25, 58, 59), typically targeting robotic replicas made of rigid linked elements actuated with servomotors.” Of these references, only 59 includes a robotic replica.

We have now separated the references between biological studies and robotic cases, as well as added a new reference in the latter case (Wright (2007)). The new paragraph now reads:

Page 13, paragraph 3, line 1 (main text):

“Extensive work has been done on understanding snake locomotion (12, 28, 72), targeting robotic replicas made of rigid linked elements actuated with servomotors (73, 74).”

28. “While numerous work has demonstrated the successful gait of a slithering snake (60, 61)” What does that mean? That snakes use a gait that works? Which gait? There is more than one.

Our original sentence is indeed not properly stated. We modified it as:

Page 14, paragraph 1, line 8 (main text):

“While previous studies modeled the entire snake as a single elastic beam (75, 76) and were able to replicate various gaits, we emphasize that our goal here is to deconstruct its complex internal biomechanical functioning to reveal hidden design principles.”

29. *“We claim that our musculoskeletal representation made of only four overlapping soft longitudinal actuators can closely approximate the idealized continuous reference above, which sets the attainable velocity upper bound (62).” At this point in the manuscript, there is no basis for “claiming” this; I believe the intent here is to “test”.*

We agree that “claim” is not the appropriate term, and modified the sentence as:

Page 15, paragraph 2, line 16 (main text):

*“We **hypothesize** that our musculoskeletal representation made of only four overlapping soft longitudinal actuators can closely approximate the idealized continuous reference above, which sets the attainable velocity upper bound (61).”*

30. *In the snake model (shown in Fig 3a), I don’t quite understand the how the lateral insertion/originations were modeled. The iliocostalis attaches to the end of the rib, providing a larger moment arm than the semispinalis, which attaches much closer to the midline on the vertebra.*

We should have better outlined the simplifications inherent in our model and the rationale behind them. The main goal of this section is not necessarily to faithfully reproduce the snake anatomy, rather test its design principle based on overlapping longitudinal muscles. We then simplified the actual snake architecture by lumping semispinalis-spinalis (SSP-SP), longissimus dorsi (LD) and iliocostalis (IC) into an individual muscle group intervened between the tendons as shown in the inset of the bottom image in Fig. 3a (reproduced here for reference). This simplifies the problem, while retaining the fundamental character (overlapping of long longitudinal muscles) of this architecture. We now clarify this.

Page 14, paragraph 2, line 5 (main text):

*“**In our simulation, the three major lateral muscle–tendon groups responsible for locomotion (semispinalis-spinalis (SSP-SP), longissimus dorsi (LD) and iliocostalis (IC)) are represented by a single group**—one muscle bundle of radius 6 mm spanning across 12 vertebrae and intervened between two tendons of radii 3 mm and adjustable lengths (Fig. 3a). Two hinge joints anchor the extrema of these longitudinal actuators along the snake’s body at half its radius away from the midline vertebra. While this simplifies the snake’s overall architecture, it retains its fundamental design principle and allows us to test the functioning of overlapping longitudinal muscle and compare with idealized elastic beam cases (61, 76).”*

31. Fig 3a legend: the skeletal drawing was modified from Bruce Jayne’s work, and the attribution is both improperly done and wrongly cited.

The reviewer is right and the mentioned image is adapted from Bruce Jayne’s work (Jayne (1988)). We thought we had explicitly stated it, and sincerely apologize about this. We now amend that by mentioning the image source both in the text, in the figure and in the caption.

Page 16, Fig. 3a caption (main text):

“(Middle) Sketch of the snake lateral muscle anatomy (adapted from Bruce Jayne’s (77) illustration of the epaxial muscle segment comprised of multiple muscles and tendons).”

32. “The identified optimal design exhibit muscle groups that span roughly 30–40 vertebrae (Fig. 3d), agreeing reasonably well with biological observations (63).” A few snakes (mostly arboreal) exhibit this large number, but most species employ a smaller span (<15), at least according to the cited paper. Also, fix the grammar.

We should have better clarified this. Since our simplified soft snake models the three major muscles as a single entity, we compare with the major span of epaxial muscle segments, which overall encompasses ~27 vertebrae as reported in Jayne (1988). We have then clarified this aspect and fixed the grammar.

Page 15, paragraph 3, line 4 (main text):

“The identified optimal design exhibits muscle groups that span roughly 30–40 vertebrae (Fig. 3d). This is in reasonable agreement with biological observations where the snake’s major epaxial muscle segments in total span ~ 27 vertebrae (79).”

33. “This stems from the discrete, limited number of muscles, and is reflected in the more prominent lateral sliding of the midline kinematics (Fig. 3f).” Lateral undulation in typical conditions does not include sliding.

Here we meant to refer to the lateral displacement of the gait envelope. We apologize for the confusing use of “sliding” and have updated the text as well as Fig. 3e as follows:

Page 17, paragraph 1, line 5 (main text):

“This stems from the limited number of muscles, and is reflected in the more prominent lateral displacement of the midline kinematics (Fig. 3e).”

34. “as indicated by the feasibility of a fully compliant, realistically slithering snake robot made of a few simple actuators.” As far as I’m aware, there were no robots used in this study, only simulations.

We meant to refer to a computational snake robot, which can serve as a blueprint for an actual robot with few actuators. We have clarified this in our manuscript as follows:

Page 17, paragraph 2, line 1 (main text):

“Thus, this study illustrates a framework to simplify, test and distill biomechanical principles out of complex biological systems as indicated by the feasibility of a fully compliant, realistically slithering and fast snake made of a few simple actuators.”

35. “By solving an inverse problem, an optimal musculoskeletal layout is identified,” This type of statement will rankle biologists; the model was optimized, but optimality in the biological system is a different story.

We agree with the reviewer and refined our statement accordingly.

Page 17, paragraph 2, line 3 (main text):

*“By solving an inverse problem, the musculoskeletal layout for **a potential soft robotic snake** is identified, guiding its practical design and manufacturing.”*

36. “so that the bending stiffness is within biological range (67).” What is the range? Tell us so that we can assess and compare the values.

Regarding the bending stiffness of rachis, we set our Young's modulus ($E = 2.5$ MPa) and radius (2.5 mm) which results in a bending stiffness of 7.67×10^{-5} MPa, approximately consistent with values reported for *Columba livia* in Wang et al. (2012). The barbs, however, are modeled differently since they interlock with one another to increase their effective bending stiffness (Sullivan et al. (2016)). Moreover, the number of barbs differ significantly between feathers of the same species and between species. In view of the challenges in finding accurate data (on the scale of a single barb) and consistent measurements for the specific species, we took a pragmatic engineering approach and approximate five barbs as a single computational representative rod in our simulation. Subsequently, we choose the properties of our computational barb such that it conserves the feather area (consistent with data from Wang et al. (2012)) and approximates the estimated aggregate bending stiffness, which is chosen to be ~ 10 times larger than the numbers reported in Sullivan et al. (2016) (to account for the 5 barbs and their interlocking). We have updated the text in the manuscript to redirect the reader to the Supplementary Information, where a section reflecting these modeling details can be found.

Page 18, paragraph 2, line 4 (main text):

*“Motivated by these investigations, we consider the dynamics of the wing structure of a pigeon (*Columba livia*). We reconstruct in-silico remiges feathers of lengths ranging from 7 to 15 cm, and model rachis ($E = 2.5$ MPa, $\rho = 2.5 \times 10^{-3}$ g/cm³) as filaments of radii 2.5 mm so that the bending stiffness is consistent with (83). Using fixed joints, 200 barbs are attached to one rachis (Fig. 4a), producing feather of total area 28 cm² similar to (83). Each computational barb represents approximately five real ones, and its radius (1.5 mm) is set to match area and estimated aggregate bending stiffness (84) (SI). Overall, 19 feathers are connected to the wing skeleton, comprised of six bones, for a total of ~ 3000 rods per wing.”*

Page ix, paragraph 2, line 1 (Supplementary Information):

“In the feathered wing example in the main text, we model the geometrical and material properties of the rachis to exactly correspond to those found in nature (22). This is difficult to do in the case of barbs. Indeed, the number of barbs differ significantly between species and between feathers within a species (23). Moreover, barbs interlock with one another to increase their effective stiffness (24). In view of all these difficulties, we adopt an engineering approximation and lump 5 biological barbs into a single computational one while conserving the rachis' barb count (23). The properties of our computational barb are chosen so as to conserve the feather area (22) and the estimated aggregate bending stiffness (chosen to be 10 times larger than the numbers reported in (24)). The geometrical and mechanical properties of the feathered wing example are given in Table S3.”

37. Change to “four-muscle model”.

We have updated the manuscript as suggested by the reviewer.

38. References: There are errors in some of the citations that should be fixed.

We went through the manuscript and fixed the references.

39. “We are interested in capturing both biomechanics and morphology in musculoskeletal systems, whereby Hill’s model and Cosserat theory are leveraged.” Hill’s model and Cosserat theory are not leveraged, they are simply used.

We have updated the text to reflect the updated statement by replacing “leveraged” with “used”.

40. Figure S1: What are CE, PE, and SE? Also, change to “three-element model”.

We apologize for the lack of details in the figure. We have updated the caption to reflect the corresponding element in the Fig. S1 (reproduced here for reference).

Page ii, Fig. S1 caption (Supplementary Information):

“Figure S1: (a) Hill three-element model—contractile (CE), parallel elastic (PE) and serial elastic (SE) elements—and simulation of a head of biceps brachii containing 18 filaments, each made up of 20 motor units. (b) Size principle relating muscle twitch time and peak force to motor unit size, based on (4).”

41. “Each head is made of 360 motor units (9)” What does the reference here mean? Explain.

As mentioned in our response to comment #6, the reference (Brown (1988)) here pertains to the estimated number of motor units in each head of the biceps brachii, which we model as 360 motor units per head based on the study of **Brown (1988)**. We have updated the manuscript to motivate the need for cited reference.

Page ii, paragraph 2, line 2 (Supplementary Information):

“Each head is made of 360 motor units **based on (9)** and equally-distributed among $N = 18$ filaments.”

42. “we create a biceps head with a cross-sectional area $A = 804.2 \text{ mm}^2$ and 18×10^4 muscle fibers, in agreement with experimental data (10).” In agreement can mean any number of things. Tell us specifically what you mean.

Given the data reported in Klein et al.(2003) (table reported here for reference), the average fiber area between young and old men is $A_f = 4413.5 \mu\text{m}^2$. Assuming that each of our filament correspond to 10,000 fibers (round number for simplicity), this gives a total of 180,000 fibers. We use 18 filaments (a convenient number due to its cylindrical packing properties) leading to the overall muscle cross-sectional area of 804.2 mm^2 , which is within the range of average measurements.

Variable	Young	Old
Biceps area (mm^2)	$1,223 \pm 222$	899 ± 208 *
Type II fiber %	61.5 ± 3.4	59 ± 7.1
% Type II area	74.2 ± 7.2	60.6 ± 8.2 *
Type I fiber area (μm^2)	$4,384 \pm 1,018$	$3,883 \pm 656$
Type II fiber area (μm^2)	$5,229 \pm 973$	$3,976 \pm 624$ *
Type II/type I area ratio	1.21 ± 0.14	1.03 ± 0.11 *
Mean fiber area (μm^2)	$4,895 \pm 974$	$3,932 \pm 592$
No. of fibers ($\times 10^3$)	253.6 ± 40.4	234.3 ± 67.4

Page ii, paragraph 2, line 7 (Supplementary Information):

“Relating filament diameter to muscle fiber count in each motor unit (assuming an average fiber diameter of $75 \mu\text{m}$ (10)), we create a biceps head with a cross-sectional

*area $A = 804.2 \text{ mm}^2$ and 18×10^4 muscle fibers, approximately consistent with **the cross-sectional area and muscle fiber count reported in (10)***

43. *Change to “muscle unit’s”.*

We have updated the manuscript.

44. *“we are capable of replicating any musculoskeletal system given the biomechanical properties.” Possible, but that remains to be seen. Far too strong of a statement, based on only a few models.*

We softened our statement as “we are capable of replicating complex musculoskeletal systems given their biomechanical properties as long as they are mostly comprised of slender elements.”

45. *The tendons are tapered (as appears with some of the muscles). What is the effect on the modeling of this tapering?*

The tapering of tendons and bones is motivated by biological morphology. Since we prescribe a constant Young’s modulus, the tapering then affects the stiffness of the structure at different locations (varying radius affects bending stiffness). This implies that the ends of the tendons are more susceptible to bending due to its reduced cross-sectional area. We did not systematically quantify the effect of tapering as we were more concerned with building a geometrically plausible model. Nonetheless, in this particular case (elbow), we do not expect this to play a significant role as tendons (which are tapered) are significantly stiffer than muscle tissue (which is the only element that undergoes significant deformations) and mostly serve the purpose of communicating tensile stresses from the biceps to the bones.

46. *‘antropometry’ is misspelled.*

We have fixed the spelling error.

47. *“see that $fact(\eta) = Fact/F_{max}$ decreases with contraction or extension.” Physiologically, muscle contraction refers to the activation of the muscle, which might involve shortening, extension, or no net displacement. I assume that shortening is meant here.*

Indeed, we are referring to shortening here. We have modified the text accordingly.

48. In Movie S1, why does the muscle start out in yellow and then change color?

The color reflects the level of muscle contraction in Movie S1. When the muscle is at rest length and no contraction occurs, it is yellow. As the muscle shortens, it slowly turns red to reflect the level of contraction. Similarly, this is reflected in Fig. 1b in the main text (reproduced here for convenience), where snapshots at different point in time from the video are taken.

49. There are errors in the references.

We fixed this in the revised manuscript.

Finally, we thank the reviewer for her/his careful evaluation and critical comments which helped us in improving our manuscript significantly. We believe this resulted in a stronger and clearer report, and hope the reviewer finds our answer satisfactory and the revised manuscript is suitable for publication.

References used in this response:

- A. Biewener, W. R. Corning, and B. W. Tobalske. In vivo pectoralis muscle force-length behavior during level flight in pigeons (*Columba livia*). *Journal of Experimental Biology*, 201(24):3293–3307, 1998.
- W. F. Brown, M. J. Strong, and R. Snow. Methods for estimating numbers of motor units in biceps-brachialis muscles and losses of motor units with aging. *Muscle & nerve*, 11(5):423–432, 1988.
- G. Cicconofri and A. DeSimone. A study of snake-like locomotion through the analysis of a flexible robot model. *Proc. R. Soc. A*, 471(2184):20150054, 2015.
- R. Cox. The motion of long slender bodies in a viscous fluid part 1. general theory. *Journal of Fluid mechanics*, 44(04):791–810, 1970.
- A. Gauthier, D. Davenne, A. Martin, and J. Van Hoecke. Time of day effects on isometric and isokinetic torque developed during elbow flexion in humans. *European Journal of Applied Physiology*, 84(3):249–252, 2001.
- C. Haines, M. Lima, N. Li, G. Spinks, J. Foroughi, J. W. Madden, S. Kim, S. Fang, M. Jung de Andrade, F. Goktepe, O. Goktepe, S. Mirvakili, S. Naficy, X. Lepř o, J. Oh, M. Kozlov, S. Kim, X. Xu, B. Swedlove, G. Wallace, and R. Baughman. Artificial muscles from fishing line and sewing thread. *Science*, 343(6173):868–872, 2014.
- C. Haines, N. Li, G. Spinks, A. Aliev, J. Di, and R. Baughman. New twist on artificial muscles. *Proceedings of the National Academy of Sciences*, page 201605273, 2016.
- N. Hansen. Benchmarking a bi-population cma-es on the bbob-2009 function testbed. In *Proceedings of the 11th Annual Conference Companion on Genetic and Evolutionary Computation Conference: Late Breaking Papers*, pages 2389–2396. ACM, 2009.
- N. Hansen and R. Ros. Benchmarking a weighted negative covariance matrix update on the bbob-2010 noiseless testbed. In *Proceedings of the 12th annual conference companion on Genetic and evolutionary computation*, pages 1673–1680. ACM, 2010.
- D. L. Hu, J. Nirody, T. Scott, and M. Shelley, “The mechanics of slithering locomotion,” *Proceedings of the National Academy of Sciences*, vol. 106, no. 25, pp. 10081–10085, 2009.
- D. L. Hu and M. Shelley. Slithering locomotion. In *Natural locomotion in fluids and on surfaces*, pages 117–135. Springer, 2012.
- B. C. Jayne. Muscular mechanisms of snake locomotion: an electromyographic study of lateral undulation of the florida banded water snake (*Nerodia fasciata*) and the yellow rat snake (*Elaphe obsoleta*). *Journal of Morphology*, 197(2):159–181, 1988.
- C. S. Klein, G. D. Marsh, R. J. Petrella, and C. L. Rice. Muscle fiber number in the biceps brachii muscle of young and old men. *Muscle & nerve*, 28(1):62–68, 2003.
- Y. Moriwaki, N. Sakai, Y. Sawae, and T. Murakami. Development of forearm models based on human musculoskeletal system. *Journal of Biomechanical Science and Engineering*, 4(1):153–164, 2009.
- D. M. Nguyen and N. Hansen. Benchmarking cmaes-apop on the bbob noiseless testbed. In *Proceedings of the Genetic and Evolutionary Computation Conference Companion*, pages 1756–1763. ACM, 2017.
- R. Raman, C. Cvetkovic, S. Uzel, R. Platt, P. Sengupta, R. Kamm, and R. Bashir. Optogenetic skeletal muscle-powered adaptive biological machines. *Proceedings of the National Academy of Sciences*, 113(13):3497–3502, 2016.
- M. B. Robertson and A. A. Biewener, “Muscle function during takeoff and landing flight in the pigeon (*Columba livia*),” *Journal of Experimental Biology*, vol. 215, no. 23, pp. 4104–4114, 2012.
- J. Rubenson, N. Pires, H. Loi, G. Pinniger, and D. Shannon. On the ascent: the soleus operating length is conserved to the ascending limb of the force-length curve across gait mechanics in humans. *Journal of Experimental Biology*, 215(20):3539–3551, 2012.
- Sullivan, T. N., Pissarenko, A., Herrera, S. A., Kisailus, D., Lubarda, V. A., & Meyers, M. A. (2016). A lightweight, biological structure with tailored stiffness: The feather vane. *Acta biomaterialia*, 41, 27-39.

- X. Wang, R. Nudds, C. Palmer, and G. Dyke. Size scaling and stiffness of avian primary feathers: implications for the flight of mesozoic birds. *Journal of evolutionary biology*, 25(3):547–555, 2012.
- C. Wright, A. Johnson, A. Peck, Z. McCord, A. Naaktgeboren, P. Gianfortoni, M. Gonzalez-Rivero, R. Hatton, and H. Choset. Design of a modular snake robot. In *Intelligent Robots and Systems, 2007. IROS 2007. IEEE/RSJ International Conference on*, pages 2609–2614. IEEE, 2007.

Reviewers' Comments:

Reviewer #1:

Remarks to the Author:

I am satisfied with the revision.

Reviewer #2:

Remarks to the Author:

Thank you for the very thoughtful and thorough response to the reviews, and for the improvements to the manuscript.

First, a big picture question. From a bird's-eye perspective, this paper's goal is to advance the argument that numerical models of elastic rods have a valuable role in modeling an assortment of biological systems as well as hybrid synthetic-biological systems. In my opinion this is an argument worth putting forward. The challenge is in arguing that a specific model has hit the "sweet spot." On the one hand, there are numerical models that are simpler than that employed in the submission. For instance, the strands of Pai, which in several of the papers neglect twist, are well suited for simulating muscle strands and tendons. On the other hand, other models may be more general than the numerical model employed in the submission. For instance, Finite Elements may be applied to modeling non-slender biological structures such as the skull, hip bone, liver, etc. Therefore, the "best" model to use depends on the priorities---versatility in applications, simplicity of implementation and setup, computational efficiency, numerical accuracy, etc. To make the case that one specific numerical modeling choice is the "best" can be challenging. For instance, is a Cosserat model needed or will a Kirchhoff rod suffice? The senior author's own prior paper in Royal Society Open Science makes arguments for relevance of shear and extension models, and if those arguments apply to any (or all) of the case studies examined in this submission, this may be good to point out in some detail and to provide clarity on whether all case studies are equally impacted.

Here are some more specific remarks:

Please accurately contextualize prior work on numerical models wherever possible. Two passages that stood out to me, for instance:

(a) In the revised text, line 48, I found it slightly unnatural to say "Inspired by", since this line of work was cited only in the revised submission.

(b) Citation 45 demonstrated simulation of trees, plectonemes, helices, and the model was used subsequently to simulate hair, fur, flagella, transoceanic communication cables, plant tendrils, etc., with good agreement to laboratory experiments, so "rendering of elastic ribbons" seems overly restrictive. As an aside, the line of work begun in reference 45 also included consideration of complex rod assemblies and numerical treatment of contact, e.g., "Adaptive Nonlinearity for Collisions in Complex Rod Assemblies" and related works, which may be relevant here.

The tests that have been added in the recent revision consider simultaneously the shear and twist. By stiffening both shear and twist modes, different results are obtained for the feather. This advances the argument that neglecting simultaneously both shear and twist limits the range of biological systems that can be captured. It would be interesting to determine whether (solely) shearing of the material cross sections is a necessary ingredient (as opposed to accounting for twist).

A very positive contribution of the submitted work is that the source code for both the numerical model and the specific application examples will be made accessible to the readers.

Thank you again for your thoughtful consideration of this review.

Eitan Grinspun

Reviewer #3:

Remarks to the Author:

Clearly the authors have made a large effort to improve the manuscript, including additional analyses and models. Generally I have a positive view of the manuscript, but I have a few remaining issues before I would recommend it for publication.

Some of my major comments are in regard to the snake modeling:

1) From line 250: "Although snakes are equipped with a multitude of muscles to orchestrate a variety of gaits and body deformations, we speculate that only a few are necessary for effective forward slithering. We test this hypothesis by reducing the system complexity—in favor of engineering feasibility—via the symmetrical arrangement of only four antagonistic, lateral muscle-tendon groups whose location and actuation are determined using the CMA-ES optimization procedure."

I disagree with this idea. The model presented in this study represents a reduced-order model of the snake, and by itself it can indeed provide some insight into what may be fundamental to snake locomotion. But it doesn't test the idea that only certain muscles are necessary for certain gaits or body deformations. To do so would require a complete model that accurately represents every aspect of the musculoskeletal system, and then one could erode away components systematically to see what is essential. Or, one could take the real animal and do the same thing, say for instance by dissecting away muscles (not advised) or paralyzing muscles systematically to see what their effect is. Real anatomy has complex interactions that can't always be predicted from simple models, so such an approach is required to test the idea rigorously. As an example to this approach, using ablation of appendages, see:

Aerial manoeuvrability in wingless gliding ants (*Cephalotes atratus*)
Stephen P. Yanoviak, Yonatan Munk, Mike Kaspari and Robert Dudley
Proc. R. Soc. B 2010 277, 2199-2204

2) I still don't understand the snake model. It is stated that it "allows us to test the functioning of overlapping longitudinal muscle", which implies that there the model actuators are overlapping. I don't see the overlap in Figure 3g. In fact I can't even tell how many virtual muscles there are. On each side of the body, it appears to be three, and they are all in series. If so, why isn't this described explicitly in the manuscript? Somewhere, I would like to see a more detailed figure that shows exactly what the layout is of the virtual muscles within the body. The same is true for the bird wing — it's hard to see in figure 4b where exactly the connections are, and to what elements.

3) Perhaps a slightly more minor point, but the comparison to the snake in Hu & Shelley 2012 isn't the greatest — that book chapter only provides the most superficial of analyses of the snake's movements. What is described as 'sprinting' in that chapter is not even characterized in the simplest of terms (as far as I can tell, the forward speed, amplitudes, and even snake numbers and lengths are not provided). As such, there's no way to evaluate what they mean by 'sprinting' or 'leisurely', which by the way, are not accepted terms in snake locomotion. (Sprinting is a running, used by animals with legs. How slow or fast is 'leisurely'? I naturally walk fast, and my 'leisurely' stroll might be a brisk walk for someone else.) It would be a disservice to propagate the term 'sprinting' for these reasons. This also brings up an ancillary point — Hu and Shelley's book chapter might have been more biologically informed if they had involved a biologist in the work;

similarly, this current study might have been better served if a biologist had been involved. (I also advise the converse: I have seen biologists whose work delves into engineering, who don't involve engineers and whose work would be better served if they had crossed over disciplines to collaborate.)

Other specific comments by line:

Page 1 Change 'feedbacks' to 'feedback'.

Page 1 "we present here a versatile and robust numerical approach to the simulation of arbitrary musculoskeletal architectures"

Delete arbitrary—if one were to use this approach, one would not choose the architectures arbitrarily.

5 Change 'rigid body robots' to 'rigid-body robots'.

14 "to a zoo of robotic creatures" A zoo is a specific thing, which is not a collection of robots; what is meant here is 'range'?

24 "as mechanical structures comprised of springs"

Here and elsewhere throughout the manuscript, change this either to 'composed of' or 'comprising'. See here for an explanation of the difference:
<https://www.grammarly.com/blog/comprise-vs-compose/>

34 Remove the capital letter from 'Finite Element Method' and add 'the' at the beginning.

37 Change 'discretization elements' distortion' to 'distortion of discretization elements'.

42 "This represents..."

Here and throughout the rest of the manuscript, the word 'this' should usually not be used as a noun, because it is ambiguous. Here, does the word mean paper, Euler beams, 3D lattices, or something else?

50 There's no need to italicize 'assemblies'.

65 "We capitalize on our previous work" I believe that 'build' is intended rather than 'capitalize', unless you are referring to making money off of your previous work.

82 "to study and understand in-silico biophysical functioning of natural creatures."

In silico should not be hyphenated, here and elsewhere in the manuscript. Also, the term should fall at the end in this sentence.

89 "It also serves the purpose of illustrating our representation's level of detail, which can be leveraged to address patient-specific kinesiological needs."

Because a large portion of this paper deals with non-human organisms and this is the first mention of patients, it should be noted that this refers to human patients.

93 "viscoelastic Cosserat rods or filaments with each rod representing motor units (SI)."

Does each rod always represent more than one motor unit? Explain more detail here.

94 Remove the capitalization from Sections.

Figure 1b. Change 'Level of muscle contraction' to 'Level of muscle shortening'. Same in figure 3.

127 Change to 'confirming a good match'.

139 Remove the capitalization from Sections.

140 What is a collision-check, and why is it hyphenated?

158 Remove the capitalization from Polydimethylsiloxane.

185 "The above parameter ranges are determined to account for actual manufacturability (23)."

Manufacturing by the authors, in this paper? Or for future work? Distinguish. Also, why does this statement need a citation? Do we need to read that paper to understand the use of parameter ranges in manufacturing?

201 Change to 'a previous demonstration'.

205 Change to '14 mm,'

209 Most researchers in biomechanics report muscle contractile stress in units of MPa. Please convert.

217 Change to 'leg length'.

235 "to accurately capture the physics of cell- and muscle-powered soft robotic systems"

240 I recommend changing 'exploit' to 'use' or 'employ'.

263 "represented by a single group—one muscle bundle of radius 6 mm spanning across 12 vertebrae and intervened between two tendons of radii 3 mm and adjustable lengths (Fig. 3a)."

The usage of 'intervened' seems off, though I can't put my finger on exactly why so. Perhaps it's because the only usage I've seen of this in anatomical lingo is as 'intervene', as in the groups intervenes between two tendons. One suggestion is something like, "represented by a single group—one muscle bundle of radius 6 mm spanning across 12 vertebrae; this group intervenes between two tendons of radii 3 mm and adjustable lengths (Fig. 3a)."

Figure 3. I don't think it's necessary to name Bruce Jayne in the figure and the legend. Instead, indicate that it is modified with permission from the publisher (assuming that has been done).

Also, change 'antagonistic muscle segments intervened between tendons' to 'antagonistic muscle segments that intervene between tendons'.

293 "The identified optimal design exhibits muscle groups that span roughly 30–40 vertebrae (Fig. 3d). This is in reasonable agreement with biological observations where the snake's major epaxial muscle segments in total span ~27 vertebrae (79)."

I scanned through the cited paper, and can't seem to understand where this number 27 comes from. That paper shows very clearly that there is a large amount of variation among snake species in the extent of coverage of the epaxial musculature; it would be better to simply make this

statement and provide a range reported in the paper.

301 "We additionally show experimentally recorded midline gaits"

'Show' seems to be an odd word choice here; it seems that the previous data are being compared to results in this study.

306 "four servomotors that would exhibit a rather clunky motion."

'rather clunky' is subjective; what is more precisely meant?

311 "This is shown to approximate the idealized continuous actuation case, highlighting an ingenious natural solution based on overlapping longitudinal actuators."

The use of 'ingenious' to describe one's own work is a bit self-congratulatory; for the authors' own reputational sake, I'd suggest to tone this down.

322 "Several studies have been conducted to understand the different biophysical aspects of bird flight"

A large number studies have been conducted, not just a few.

324 Change to 'to thrust generation'.

327 Change to 'and model the rachis'.

329 Change to 'producing feathers of total area 28 cm², similar'.

330 "is set to match area" The area of what?

336 Change 'dorso-ventral' to 'dorsoventral', here and elsewhere.

380 Change to 'feedback'.

517,520 Capitalize 'Bioinspiration' and 'Chrysopelea'. And the species name should be italicized. Please check the remaining citations for errors; there are errors in the supplementary references as well, which might not be caught by the journal's editors.

Response to Reviewer 2:

Modeling and simulation of complex dynamic musculoskeletal architectures

Xiaotian Zhang¹, Fan Kiat Chan¹, Tejaswin Parthasarathy¹ and Mattia Gazzola^{1,2†}

¹Mechanical Sciences and Engineering, University of Illinois at Urbana-Champaign, Urbana, IL 61801, USA

²National Center for Supercomputing Applications, University of Illinois at Urbana-Champaign, Urbana, IL 61801, USA

† To whom correspondence should be addressed; E-mail: mgazzola@illinois.edu.

We thank the reviewer for his valuable time, consideration and largely positive assessment. In the following, the comments of the reviewer are listed followed by our responses. All modifications to the manuscript are highlighted in red for the reviewer's convenience.

While reviewing the manuscript, we noticed that we softened, instead of hardening, twist and shear modes in the corresponding wing study that we included in the SI. We re-run that calculation using the correct value, and updated the SI accordingly. As the Editor and the Reviewers will see, this correction does not compromise in any way the reach of our work, and all previous general considerations remain correct. This is further confirmed through the additional tests requested by the reviewer. Nonetheless, we report it for correctness.

We hope that the reviewer considers our answers acceptable and the improved manuscript is suitable for publication.

Thank you for the very thoughtful and thorough response to the reviews, and for the improvements to the manuscript.

We are grateful to the reviewer for recognizing the effort and finding the revision thoughtful and thorough. We address the reviewer's remarks as below.

1. First, a big picture question. From a bird's-eye perspective, this paper's goal is to advance the argument that numerical models of elastic rods have a valuable role in modeling an assortment of biological systems as well as hybrid synthetic-biological systems. In my opinion this is an argument worth putting forward. The challenge is in arguing that a specific model has hit the "sweet spot." On the one hand, there are numerical models that are simpler than that employed in the submission. For instance, the strands of Pai, which in several of the papers neglect twist, are well suited for simulating muscle strands and tendons. On the other hand, other models may be more general than the numerical model employed in the submission. For instance, Finite Elements may be applied to modeling non-slender biological structures such as the skull, hip bone, liver, etc. Therefore, the "best" model to use depends on the priorities—versatility in applications, simplicity of implementation and setup, computational efficiency, numerical accuracy, etc. To make the case that one specific numerical modeling choice is the "best" can be challenging. For instance, is a Cosserat model needed or will a Kirchhoff rod suffice? The senior author's own prior paper in Royal Society Open Science makes arguments for relevance of shear and extension models, and if those arguments apply to any (or all) of the case studies examined in this submission, this may be good to point out in some detail and to provide clarity on whether all case studies are equally impacted.

We confirm the reviewer's understanding that one of the main goals of our work is to advance the argument that elastic rod models are valuable, practical tools for engineering design and fundamental biophysical discovery. We do not claim that our tool is the "best" numerical model, and we apologize if our narration has, in any way, painted this impression. Indeed, the notion of "best" is very much context specific, as rightly pointed out by the reviewer. For example, if we know *a priori* that a certain physical setup does not involve stretching or shearing, a Kirchhoff model is a perfectly good option. Nonetheless, it is not always possible to know *a priori* which modes of deformation will be excited and will be important. This is particularly true in robotics or biology since (a) musculoskeletal architectures can be rather intricate and the overall system dynamics arises from the non-linear interaction of its many elements, and (b) robots and animals typically operate in uncertain and/or dynamic environments, which can elicit modes of deformation deemed unimportant in well-controlled settings. These aspects become even more important when elastomeric or biological (i.e. soft) materials are involved. An example may guide our intuition. Let us consider a simple undulatory filament actuated by a planar torque wave. Given this setup, we know that bending modes dominate, and therefore we might decide to model it via a Kirchhoff rod model. But then, if this device is

slithering on an uneven terrain, or swimming in a turbulent flow, the choice of Kirchhoff model (vs. Cosserat for example) might lead to quantitatively and qualitatively different results. In fact, turbulence as well as impact and friction, can excite other modes of deformation including twist and shear due to their three-dimensional, intermittent and impulsive nature. On the other hand, a viscous fluid environment, due to its highly dissipative character, is not likely to elicit these modes, as illustrated by the good quantitative agreement reported for example in Jawed et al. (2015), and confirmed by our own studies (see below).

Therefore, as suggested by the reviewer, we expanded our analysis of the impact of shear and twist to all cases in the manuscript, and further complement it with proof-of-concept studies aimed at illustrating potential opportunities related to the use of highly shearable and twistable materials. This analysis confirms that architectures operating in complex environments are susceptible to all modes of deformations, and that neglecting some of them can lead to significantly different performance. Below we briefly summarize the main results, and a case-by-case detailed description is presented in answer #3 and in the SI. First, we highlight the importance of the environment by contrasting viscous flows and solid contact/friction. If we consider the original bio-hybrid swimmer of Fig. 2d (left) in the main text, hardening of shear and twist, either individually or simultaneously, cause no discernible difference. As mentioned above, this is not surprising: indeed cell-powered actuation is akin to a highly localized torque that only causes bending; there is nothing that breaks the 2D symmetry of the system (i.e. no 3D flow), and the viscous fluid is highly dissipative so that modes of deformation are dampened, rather than excited. On the other hand, if we consider a slithering snake, hardening shear/twist does produce a significant difference, with lateral gait displacement increased up to 60% and forward speed reduced as much as ~10% (see Fig. S9 in the SI and corresponding section in answer #3). This is due to the slip-and-stick nature of friction, which causes local fluctuations in the contact forces. The friction environment is found to excite shear modes in particular. We know from Timoshenko beam theory that shear causes a beam to deflect more than bending alone. This, translated into our snake setup, causes the local normal forces exerted by the snake on the substrate to be modulated differently in a shearable material, leading to different speeds. This is in agreement with biological (Hu et al. (2009)) and robotic (Rafsanjani et al. (2018)) studies in which friction manipulation is a key factor in slithering locomotion performance. A similar effect is even more prominent in the case of our bio-hybrid walker in which speed differences up to ~43% are observed. As the complexity of robotic or animal architectures increase, such effects may be further exacerbated.

Moreover, we report two study cases designed to specifically highlight the roles (and potential opportunities) related to shear and twist in separate and different contexts (locomotion—new—and human kinesiological rehabilitation—already present in the previous review):

1. We consider a slithering snake moving on an uneven terrain, and show how modulating shearing modes alone can determine success or failure in overcoming a slope.
2. We designed an assistive device which entirely relies on twist modes, and that is inspired by recent advancement in artificial muscle fabrication. This device is shown to approximately restore the ability of an injured elbow joint to flex.

These examples provide the physical rationale for enriching the family of elastic rod models with an approach able to capture the dynamics of assemblies of Cosserat rods in which all modes of deformations can be activated. As long as the architectural elements are slender, this method is accurate and robust and allows us to quantitatively test the importance of various deformation modes, either ruling them out or quantifying their impact. Either way, this will advance the understanding of a given system, potentially providing new design opportunities. Thus, our method is not better/worse or in contrast with previous models. Instead, it complements them by increasing the range of possible applications. Moreover, it still retains the desirable properties of elastic rod models, providing a more practical approach than FEM which are computationally expensive and have so far struggled with highly deformable, heterogenous and dynamic architectures.

We conclude that this discussion is centered on how our approach is positioned with respect to the numerical and modeling literature. Nonetheless, we wish to highlight that there are several elements of novelty in this work.

1. We extensively validate our models against a number of experiments, from human anatomy to bio-hybrid systems and full feathered wings, both in quasi-static and dynamic settings. Although validation studies are sometimes downplayed, we firmly believe that the careful, quantitative demonstration of these methods in complex, real life scenarios is powerful in attracting the interest of the engineering community. In that, this is a valuable contribution to advance the cause of elastic rod models in general.
2. Still related to the dissemination aspect, our software will be made openly available (an aspect appreciated by all three reviewers).
3. We demonstrate the use and robustness of our approach in the context of bio-hybrid robotic design, an emerging field currently severely hindered by the lack of computational design tools (Ceylan et al. (2017)).
4. We couple our methods with an optimization technique, and illustrate its use to rigorously flesh out a design paradigm based on overlapping actuators, and inspired by biological snakes.
5. We illustrate fidelity in replicating full-scale organisms, by reconstructing and actuating feathered wings based on actual biological recordings.

These are all elements that appeal to the robotics and biology communities, as also acknowledged by the other two reviewers (who presumably belong to those two communities given their comments).

To reflect this discussion, we updated the introduction as follows:

Page 3, line 44 (introduction):

“The graphics community has also been active in this space (37–39). Grinspun and colleagues introduced a popular discrete elastic rods method (40) for the simulations of elastic yet unshearable and unstretchable filaments, and considered their assembly into dense, entangled masses (41). Pai and colleagues investigated combining together spline-based muscle-strands for the simulation of various human body parts such as hands or ocular muscles (37, 38, 42–44). Although numerically efficient, these approaches are specialized to scenarios in which shear, stretch, and/or twist and dynamic effects are unimportant. In contrast, in the context of dynamic musculoskeletal systems (biological or robotic) operating in unstructured environments, all the above effects can be excited and become important. Hence the need for efficient, robust and more broadly applicable solvers. Thus, building on the above methods, we have developed an approach based on assemblies of Cosserat rods (45).”

Page 4, line 60 (introduction):

“The Cosserat model and its (far more popular) unstretchable, unshearable counterpart, the Kirchhoff model (46), spurred by the work of Grinspun and colleagues (40), have led to a number of graphics applications through the realistic simulation of elastic ribbons (47, 48), woven cloth (49, 50), entangled hair and fibers (41, 48, 51), wire mesh (52) and viscous threads (53). Moreover, these models found application in physics, biology and engineering to characterize polymers and DNA (54, 55), flagella (25, 56), tendrils (57), cables in automotive design (58), and soft robot arms (59).”

Page 5, line 88 (end of the introduction):

“Overall, this study advances the argument that rod models have a valuable role to play in complex, active settings, further expanding their range of application in robotics and biology.”

Additionally, we have updated the relevant sections in the main text highlighting effects of twist and shear as well as directing readers to corresponding sections in SI as follows:

Page 9, line 157 (human elbow joint):

“As an example, in the SI, we propose an assisting device (inspired by the coiled fishing lines of (68)), which converts internal twist into contraction forces to aid restoring the weightlifting abilities of an injured biceps. (b) Our compliant muscles are allowed to bend, twist and shear to respond realistically to the dynamics of the entire structure and the environment. Indeed, the investigation presented in section 6 of the SI finds that neglecting twist or shear (often assumed unimportant) can have a significant quantitative and qualitative impact, especially when the environment produces three-dimensional, fluctuating and impulsive loads.”

Page 13, line 238 (bio-hybrid walker):

“Adding to these difficulties, the slip-and-stick nature of surface friction and contact is found to be able to excite not only bending, but also twist and shear (SI), which, if neglected, can lead to up to 43% differences in output speeds (SI).”

Page 18, line 331 (soft snake):

“As a side note, we also find that snake locomotion is affected by twist and shear modes, consistent with the bio-hybrid walker study, with observed differences in speed and lateral gait displacement, up to ~10% and ~60%, respectively (SI).”

Page 20, line 385 (feathered wing):

“As an additional comment, in this case too we find that twist and shear can play a role, as illustrated by the hardening/softening study reported in the SI.”

Here are some more specific remarks:

2. Please accurately contextualize prior work on numerical models wherever possible. Two passages that stood out to me, for instance:

- (a) In the revised text, line 48, I found it slightly unnatural to say “Inspired by”, since this line of work was cited only in the revised submission.***
- (b) Citation 45 demonstrated simulation of trees, plectonemes, helices, and the model was used subsequently to simulate hair, fur, flagella, transoceanic communication cables, plant tendrils, etc., with good agreement to laboratory experiments, so “rendering of elastic ribbons” seems overly restrictive. As an aside, the line of work begun in reference 45 also included consideration of complex rod assemblies and numerical treatment of contact, e.g., “Adaptive Nonlinearity for Collisions in Complex Rod Assemblies” and related works, which may be relevant here.***

We agree with the reviewer that the contextualization of our work needs to be improved. The modifications reported in the introduction and listed in answer #1 account for these.

In the introduction, we now give more prominence to the general argument of advancing the role of rod models in engineering and biology (ans. #1), better place in context reference 45 (Bergou et al. (2008)) among the techniques aiming at filling the gap between rigid links models and FEM (ans. #1), and specify how (Bergou et al. (2008)) led to a number of demonstration in a variety of contexts (ans. #1 and additional references below). Finally, we carefully read through the text and remove language that might suggest that ours is the “best” model. This is not our intent here, and we instead better define the scenarios in which our approach is most suitable, i.e. in the case of 3D, dynamic environmental loads.

Additional references:

- D. M. Kaufman, R. Tamstorf, B. Smith, J. M. Aubry, and E. Grinspun, “Adaptive nonlinearity for collisions in complex rod assemblies,” *ACM Transactions on Graphics (TOG)*, vol. 33, no. 4, p. 123, 2014.
- D. Harmon, E. Vouga, B. Smith, R. Tamstorf, and E. Grinspun, “Asynchronous contact mechanics,” in *ACM Transactions on Graphics (TOG)*, vol. 28, p. 87, ACM, 2009.
- M. K. Jawed, N. K. Khouri, F. Da, E. Grinspun, and P. M. Reis, “Propulsion and instability of a flexible helical rod rotating in a viscous fluid,” *Physical review letters*, vol. 115, no. 16, p. 168101, 2015.
- A. Garg, A. O. S. F., B. Deng, Y. Yue, E. Grinspun, M. Pauly, and M. Wardetzky, “Wire mesh design,” *ACM Trans. Graph.*, vol. 33, no. 4, pp. 66–1, 2014.

3. The tests that have been added in the recent revision consider simultaneously the shear and twist. By stiffening both shear and twist modes, different results are obtained for the feather. This advances the argument that neglecting simultaneously both shear and twist limits the range of biological systems that can be captured. It would be interesting to determine whether (solely) shearing of the material cross sections is a necessary ingredient (as opposed to accounting for twist).

We thank the reviewer for his suggestion to test the individual as well as combined effects of shearing and twisting. These studies consolidate the argument that a solver capturing all deformation modes (including twist and shear) is an important asset to understand the full dynamics of complex, heterogenous architectures. Moreover, it brought to our attention new interesting lines of research and potential opportunities related to the deliberate use of shear and twist.

We then confirm that neglecting certain deformation modes limits the range of biological systems that can be modeled accurately, especially when complex architectures and/or environments that produce three-dimensional, unsteady and impulsive loads are considered. These studies have now been combined and reported in the new section “Impact of twist and shear modes and potential opportunities” in the SI. As requested by the reviewer, in this new section we numerically test the individual and simultaneous effects of twist and shear-hardening (and in some cases, softening) for the human elbow joint, bio-hybrid swimmer and walker, snake and feathered wing cases. Moreover, we present two proof-of-concept studies to further underscore potential opportunities related to the deliberate use of highly shearable/twistable materials.

Here we highlight and summarize our major findings:

- **Bio-hybrid walking robot (section 6.4, SI):** We observe that twist/shear hardening has a quantitative significant impact (up to ~43%) on the output walking speed. We relate the observed changes to the slip-and-stick nature of surface friction and contact, which produces three-dimensional, localized, intermittent and impulsive forces able to excite twist and shear modes. In turn, these modulate the normal forces acting on the substrate (i.e. friction response) which manifest in different propulsion velocities.
- **Soft, limbless, slithering snakes (section 6.5, SI):** Here again we relate the variation in snake's forward speed (up to ~10%) and lateral gait displacements (as much as ~60%) to the local normal forces that are modulated differently (as also observed in the walker case) due to differences in shearability of the snake.
- **Snakes slithering on uneven terrain (section 6.6, SI):** This is perhaps the most interesting study among the new set of investigations. We challenge our snake to overcome uneven terrain features (a slope in this case) to further highlight the prominence of out-of-plane (i.e. 3D) shear. The rationale is that shear effectively allows larger deflections than bending alone, thus affecting how the snake complies and adheres to the slope. As illustrated in detail in the SI and in the following, this mechanism has a dramatic effect, determining the difference between success or failure in overcoming the barrier.
- **Rehabilitation using artificial, assistive muscles (section 6.2, SI):** This testcase was already present in our previous submission, but we recall it here to emphasize potential opportunities enabled by twisting modes specifically. In this study, we mimic an injury by impairing the ability of a portion of the biceps to contract. We then show how functionality of the arm can be approximately recovered by attaching an external artificial muscle which can generate longitudinal contractile forces by releasing stored twist into stretch. This again underscores the importance of capturing bend, stretch, twist and shear physics in general settings in which artificial and biological components might be combined together.
- **Feathered wings (section 6.7, SI):** While carrying out the additional tests requested by the reviewer, we noticed that in the investigation conducted in the wing hardening section in the SI of the previous submission, by mistake we softened twist/shear instead of hardening. We apologize for this, corrected and updated the section, now reporting both the effect of hardening and softening. We see that hardening the wing by 10 times has a quantitative effect, although not particularly significant (~ 3% change in maximum joint angles). Instead, softening (even by only 5 times) significantly alters the wing kinematics (up to ~60%). An explanation for this behavior is detailed in the SI and in the following. In particular, this demonstration highlights the marked sensitivity of the system response to shear softening of the actuator groups. In the context of bio-inspired flight, shear should then be an important consideration in the engineering of feathered ornithopters. This is relevant given the recent focus in aviation on low flight, low speed (relative to planes) devices, inspired by agile biological flyers. Thus, as new materials that mimic naturally occurring structures are developed and implemented

in such devices, it is of use to engineers to know what elements are particularly susceptible to excitement of unwanted or desired deformation modes.

- The same hardening experiment on the other demonstrations presented in this paper, the human elbow joint and cell-powered swimming flagella robot case, did not highlight any significant effect of shear and twist. This was not unexpected, and an explanation for this behavior is presented in in the SI (section 6.1 and 6.3).

Overall, these examples point to the importance of accounting for twist and shear in the case of complex dynamic architecture interacting with unsteady environmental loads. Thus, as mentioned above, they provide the physical rationale for enriching the family of elastic rod models with an approach able to capture the dynamics of assemblies of Cosserat rods in which all modes of deformations can be activated.

All the above studies are reported in detail in the SI. Here for brevity, we include the most compelling examples, as well as the corrected and updated wing section.

Bio-hybrid walking robot

Page xvii, Section 6.4 (SI):

“Here we consider the walker design of (28) with twist- and shear-hardening of the entire structure (muscle and scaffold) and compare walking performances. As can be seen in Fig. S8, both twist- and shear- hardening significantly affect the walker’s forward displacement, with speed varying up to ~ 43% between the fastest (shear-hardened) and the slowest (twist-hardened) walker. We relate this to the slip-and-stick nature of surface friction and contact, which produces three-dimensional, localized, intermittent and impulsive forces. Thus, this environment has the tendency to excite all deformation modes instead of dampening them as in the case of viscous fluids, thereby introducing (or magnifying) the twist and shear modes in the structure. Shear modes in particular (red curve in Fig. S8b) are found to play an important role. We know from Timoshenko beam theory that shear causes a beam to deflect more than bending alone. This, translated in our walker setup, causes the local normal forces exerted by the walker on the substrate to be modulated differently in the shearable and unshearable cases, leading to different speeds. This is in agreement with biological (29) and robotic (30) studies in which friction manipulation is highlighted as a key factor in locomotion performance. For example, snakes lift part of their bodies to modify their grip on the ground.

Then, in the next demonstration, we further exemplify the role of twist and shear in modulating friction in our snake models, showing how these modes can affect performances both qualitatively (fail or success in overcoming terrain features) and quantitatively (different forward speeds and kinematic envelopes).”

Figure S8: **(a)** One-to-one computational replica of bio-hybrid walker of (28). The original muscle (yellow, $E = 16$ kPa, $G = 2E/3$) is hardened for twist and shear modes by a factor of 10 (i.e. $10G$). The original scaffold (purple, $E = 319.4$ kPa, $G = 2E/3$) is hardened for twist and shear modes by a factor of 10000 (i.e. $10000G$). **(b)** Resulting time evolution of the walker’s displacements for hardening twist (blue), shear (red), and both twist and shear (green) are compared with the original swimmer performance.

Snakes slithering on uneven terrain

Page xx, section 6.6 (SI):

“We further focus on the effects of shear in a slithering snake by casting it in an environment where three-dimensional deformations are important. Indeed, so far we have considered snakes propelling on perfectly flat surfaces. Now we introduce terrain features, so as to give prominence to out-of-plane (i.e. 3D) shear mode. Then, we perform a proof-of-concept study in which the barrier shown in Fig. S10 is considered, thereby challenging the snake to overcome the effect of gravity up the slope. We then consider snakes with and without shear hardening actuated by a continuous torque profile, similar to (32). Due to the difference in compliance of these snakes, we expect them to conform to the shape of the substrate differently, thus affecting their contact with the ground. Since limbless locomotion relies crucially on friction modulation, we hypothesize that these snakes would exhibit different behaviors.

Figure S10: Uneven terrain setup (in the $x - z$ plane) demonstrating the effects of shear and twist hardening on the trajectory of a snake actuated with a continuous torque profile. The setup consists of two half-planes at $z = 0$ and $z = 0.2$. These planes are connected by a sloped barrier, which is the polynomial function described by $z = -0.4(x - 3)^3 + 0.6(x - 3)^2$ for $3 \leq x \leq 4$, chosen for C^1 continuity with the half-planes at the line of intersection, while also being symmetric about $x = 3.5$. The barrier is placed at a sufficient distance from the initial position of the snake $(0, 0, 0)$ to account for initial startup transience.

Indeed, when testing these snakes in this environment (Fig. S11), we see that the hardened snake (blue) manages to slither up the slope barrier successfully, while the one without hardening (red) cannot. Hence, the presence or absence of shear determine the difference between success and failure. Upon comparing the snake centerline trajectories at different temporal snapshots (Fig. S11), we see that shear hardening of the snake (hence, making it less compliant) results in uneven contact with the substrate while slithering up (see Fig. S11 at 7.5T). This uneven contact in turn modulates net frictional forces (both propulsive and opposing) for successful locomotion over the barrier. This is again consistent with the biological observations of (30) where snakes lift part of their bodies to modulate friction, thus locomotion.

This study then demonstrates dramatic qualitative differences in the system outcomes (success/failure in overcoming barrier) when shear is accounted for. This particular scenario, underscores the risk of designing a robotic snake using for example the Kirchhoff rod model. Indeed, unless specific precautions are taken, soft robots are made of shearable materials (31). Thus, the Kirchhoff model would under-predict the snake compliance, hence it would not detect problems in climbing up the slope. In contrast, the actual, shearable robot will struggle and potentially fail. In this specific instance then, the use of Cosserat rods is more appropriate for actual engineering design, as they will be able to detect the problem and inform roboticists on the need to account for it.

Figure S11: Snake locomotion on uneven terrain: We compare the trajectories of an unhardened and shear-hardened snake across the uneven terrain of Fig. S10. Hardening the shear and twist by a factor of 100 (i.e. $100 \times G$) leads to different qualitative outcomes. While a shear-hardened snake (left, blue) can overcome the sloped barrier given enough time, the unhardened snake (right, red) simply cannot do so. The snapshots at different times demonstrate how the hardened snake lifts its body off the substrate thus modulating frictional forces to overcome the effect of gravitational forces. The parameters for the non-hardened snake used in this setup are listed below, where the quantities and symbols retain their definition from our main manuscript and Gazzola et. al, 2018. We add that no static or rolling friction is considered in this setting. Settings: length $L = 1$ m, radius $r = 0.025$ m, density $\rho = 10^3$ kg/m³, $T_m = 1$ s, Young's modulus $E = 10^7$ Pa, shear modulus $G = 2E/3$ Pa, shear/stretch matrix $\hat{S} = (4G\hat{A}/3, 4G\hat{A}/3, E\hat{A})$ N, bending/twist matrix $\hat{B} = \text{diag}(EI_1, EI_2, GI_3)$ Nm², dissipation constant $\gamma = 5$ kg/(ms), gravity $g = 9.81$ m/s², forward kinematic friction coefficient $\mu_k^f = 0.519$, friction coefficient ratios $\mu_k^f : \mu_k^b : \mu_k^r = 1 : 1.5 : 2$, ground stiffness and viscous dissipation $k_w = 1000$ kg/s² and $\gamma_w = 10^{-6}$ kg/s, discretization elements $n = 50$, timestep $\delta t = 1 \cdot 10^{-5} T_m$, wavelength $\lambda_m = 0.97L$, torque B-spline coefficients $\beta_{i=0,\dots,5} = \{0, 17.4, 48.5, 5.4, 14.7, 0\}$ Nm.

We conclude by noting that this proof-of-concept study points to a detrimental effect of shear. Nonetheless, there might be cases in which more shearable materials might help, depending on the context and envisioned tasks. In our opinion, this is a telling example, and opens up a new, very interesting avenue of investigation which we will pursue in the future."

Feathered wings

Page xxiii, section 6.7 (SI):

"Here we consider the wing from the main text, and test the potential impact of twist and shear. We start by hardening twist and shear mode of the muscles, tendons and feathers of the wing and compare output kinematics of the three joints. As can be seen in Fig. S12, hardening both twist and shear modes by a factor of 10 quantitatively alters the resulting

kinematics of the wing, although not significantly (up to $\sim 3\%$ maximum difference in joint angles). We also note that no change is observed in the kinematics upon further hardening (we tested 25, 50 and 100 times hardening factors).

Figure S12: Resulting joint angles kinematics of the wing for different hardening/softening configurations as compared with the original. Here, the original wing muscles ($E = 16$ kPa, $G = 2E/3$), tendons ($E = 500$ MPa, $G = 2E/3$), bones ($E = 11$ GPa, $G = 2E/3$) and feathers ($E = 2.5$ MPa, $G = 2E/3$) are all hardened for twist and shear modes by a factor of 10 (i.e. $10G$). For the softening case, these modes are decreased for only the muscles and tendons by a factor of 5 (i.e. $G/5$).

These small changes were surprising at first, as we expected the complexity of this structure to amplify the role of shear/twist. Nonetheless, the particular geometries and materials of the various elements justify this outcome. Indeed, bones are effectively rigid objects given their high Young's modulus (the highest among all elements), thus further hardening does not change their behavior. Barbs and rachis are made of stiff materials too. Moreover, they are very thin, rendering them less susceptible to shearing modes. A similar argument is valid for tendons as well. Then, the most sensitive elements are the muscles. Nonetheless, the stiff tendons seem to be able to keep them under sufficient tension, so that the effects of shear and twist are contained. Hence, the observed small quantitative differences.

Then to test the above explanation, we reduced shear/twist stiffnesses (i.e softened) of the muscle–tendon groups alone. The rationale being that by softening these elements,

the overall group tension might relax, thereby allowing the other modes to be excited. Thus, we softened these modes by 5 times (half of the hardening test). As illustrated in Fig. S12, softening has a significant impact with up to ~ 60% difference in the observed kinematics. Here we highlight the marked sensitivity of the system response to softening of key components such as the muscle–tendon groups. Additionally, we note that individual twist-softening has an indiscernible effect, thus making shear-softening the primary contributor in altering the kinematics, as can be seen in Fig. S12.

In the context of bio-inspired flight, shear should then be an important consideration in the engineering of feathered ornithopters. This is relevant given the recent focus in aviation on low flight, low speed (relative to aeroplanes) devices, inspired by agile biological flyers (33, 34). Thus, as new materials (35) that mimic naturally occurring structures are developed and implemented in such devices, it is of use to engineers to know what elements are particularly susceptible to the excitement of unwanted or desired deformation modes.”

A very positive contribution of the submitted work is that the source code for both the numerical model and the specific application examples will be made accessible to the readers.

We are supporters of the open-source initiative. Our software for individual rods is already publicly available at <https://github.com/mattialab/elastica>. The software implementing this work has been made available to the Editor and the Reviewers, and will be openly distributed on Github upon publication. Future improvements and updates will also be progressively published as we develop new capabilities.

Finally, we wish to thank the reviewer for his time and useful comments. In particular, the focus on shear and twist helped us to better contextualize our work, and draw our attention to new interesting lines of research which we will pursue in the future.

References used in this response:

- D. L. Hu, J. Nirody, T. Scott, and M. Shelley, “The mechanics of slithering locomotion,” *Proceedings of the National Academy of Sciences*, vol. 106, no. 25, pp. 10081–10085, 2009.
- M. K. Jawed, N. K. Khouri, F. Da, E. Grinspun, and P. M. Reis, “Propulsion and instability of a flexible helical rod rotating in a viscous fluid,” *Physical review letters*, vol. 115, no. 16, p. 168101, 2015.
- A. Rafsanjani, Y. Zhang, B. Liu, S. M. Rubinstein, K. Bertoldi, *Science Robotics* 3, eaar7555 (2018).
- H. Ceylan, J. Giltinan, K. Kozielski, and M. Sitti, “Mobile microrobots for bioengineering applications,” *Lab on a Chip*, vol. 17, no. 10, pp. 1705–1724, 2017.
- D. M. Kaufman, R. Tamstorf, B. Smith, J. M. Aubry, and E. Grinspun, “Adaptive nonlinearity for collisions in complex rod assemblies,” *ACM Transactions on Graphics (TOG)*, vol. 33, no. 4, p. 123, 2014.
- D. Harmon, E. Vouga, B. Smith, R. Tamstorf, and E. Grinspun, “Asynchronous contact mechanics,” in *ACM Transactions on Graphics (TOG)*, vol. 28, p. 87, ACM, 2009.
- R. Goldenthal, D. Harmon, R. Fattal, M. Bercovier, and E. Grinspun, “Efficient simulation of inextensible cloth,” *ACM Transactions on Graphics (TOG)*, vol. 26, no. 3, p. 49, 2007.
- M. K. Jawed, N. K. Khouri, F. Da, E. Grinspun, and P. M. Reis, “Propulsion and instability of a flexible helical rod rotating in a viscous fluid,” *Physical review letters*, vol. 115, no. 16, p. 168101, 2015.
- A. Garg, A. O. S. F., B. Deng, Y. Yue, E. Grinspun, M. Pauly, and M. Wardetzky, “Wire mesh design,” *ACM Trans. Graph.*, vol. 33, no. 4, pp. 66–1, 2014.

Response to Reviewer 3:

Modeling and simulation of complex dynamic musculoskeletal architectures

Xiaotian Zhang¹, Fan Kiat Chan¹, Tejaswin Parthasarathy¹ and Mattia Gazzola^{1,2†}

¹Mechanical Sciences and Engineering, University of Illinois at Urbana-Champaign, Urbana, IL 61801, USA

²National Center for Supercomputing Applications, University of Illinois at Urbana-Champaign, Urbana, IL 61801, USA

† To whom correspondence should be addressed; E-mail: mgazzola@illinois.edu.

We thank the reviewer for her/his valuable time, consideration and largely positive assessment. In the following, the comments of the reviewer are listed followed by our responses. All modifications to the manuscript are highlighted in red for the reviewer's convenience.

We hope that the reviewer considers our answers acceptable and the improved manuscript is suitable for publication.

Clearly the authors have made a large effort to improve the manuscript, including additional analyses and models. Generally I have a positive view of the manuscript, but I have a few remaining issues before I would recommend it for publication.

We are grateful to the reviewer for recognizing our efforts in improving the manuscript based on her/his helpful comments from the first round of reviews. To fully address the reviewer's recent comments, we have provided additional explanations and corresponding figures, especially pertaining to the modeling of the soft snake. Detailed responses to the comments can be found below.

Some of my major comments are in regard to the snake modeling:

1) From line 250: "Although snakes are equipped with a multitude of muscles to orchestrate a variety of gaits and body deformations, we speculate that only a few are necessary for effective forward slithering. We test this hypothesis by reducing the system complexity—in favor of engineering feasibility—via the symmetrical arrangement of only four antagonistic, lateral muscle–tendon groups whose location and actuation are determined using the CMA-ES optimization procedure."

I disagree with this idea. The model presented in this study represents a reduced-order model of the snake, and by itself it can indeed provide some insight into what may be fundamental to snake locomotion. But it doesn't test the idea that only certain muscles are necessary for certain gaits or body deformations. To do so would require a complete model that accurately represents every aspect of the musculoskeletal system, and then one could erode away components systematically to see what is essential. Or, one could take the real animal and do the same thing, say for instance by dissecting away muscles (not advised) or paralyzing muscles systematically to see what their effect is. Real anatomy has complex interactions that can't always be predicted from simple models, so such an approach is required to test the idea rigorously. As an example to this approach, using ablation of appendages, see:

Aerial manoeuvrability in wingless gliding ants (*Cephalotes atratus*)

Stephen P. Yanoviak, Yonatan Munk, Mike Kaspari and Robert Dudley, Proc. R. Soc. B 2010 277, 2199-2204

We believe our statement might have not properly emphasized the intent of our investigation, and therefore a clarification is necessary. We agree with the reviewer that to identify which muscles, in a specific real snake, are critically responsible for a certain gait, the most sensible approach is a top-down one. This can be implemented, as suggested, by systematically hindering or altogether removing individual muscles to observe how gaits are affected, with the goal of identifying the essential ones. Another possibility is to start with a complete computational model of the snake accounting in detail for all muscles, and then proceed as above.

This approach can be very valuable to understand how a specific architecture works, providing great biological detail. Nonetheless, it is complex as we start from the full system, and, perhaps more importantly, may lack in generality. Indeed, it is very specific to the muscular architecture we start with. In order to generalize the gained insights, we would need to repeat the same procedure for different snakes' size and species.

Instead, the section relative to limbless locomotion in our paper aims at illustrating how our model (combined with an optimization procedure) can help to synthesize biological design principles into an engineered device. As such, here we are not so much concerned with the full replica of a real animal (that is the objective of the wing section in our paper) or in identifying what exact muscles are critical to a certain gait. Rather, we are interested in distilling general architectural principles or motifs that can be broadly applied in engineering. Because of the emphasis on generally applicable design principles, instead of biological detail, a species-agnostic, bottom-up approach might be more practical.

In the following, we guide the reviewer through our logic aided by a new set of simulations in which we show how the progressive additions of muscle groups (as opposed to *removal* in the top-down approach) affects the snake gait. We emphasize here the subtle, yet critical role of optimization: indeed, this allows us to fairly compare progressively more complex models, by identifying their best performance.

Step-by-step approach:

1. **Aim.** Distilling design principles inspired by snake muscular architecture to achieve realistic slithering in engineered devices via few, simple actuators.
2. **Hypothesis.** Smooth and fast forward slithering motion can be obtained by a small number of overlapping actuators coupled to a flexible backbone.
3. **Establish a benchmark for comparison.** Independent of snakes' internal makings, we know from experimental observations that snakes are able to achieve nearly continuously actuation. Then, if we are not interested in how muscle arrangements lead to this, a simple way to model a snake is to consider it as a slender elastic body actuated via a continuous torque planar wave. Targeting fast gaits, the shape of this function (the torque wave) can be determined via CMA-ES to maximize speed. The obtained torque function is the result of thousands of simulations guided by an evolution scheme until no further improvement in forward speed is observed. This continuum model, despite its simplicity, represents the upper bound in terms of attainable speed and gait smoothness: indeed, from a mathematical perspective, the obtained torque profile is a C^∞ function, i.e. a function infinitely smooth and differentiable, and as such, it allows for the greatest level of control (torque prescribed at each infinitesimal cross section). Biological snakes clearly approach this limit. Thus, the optimized continuum snake establishes a computational reference, but it does not tell us anything about how muscular arrangements work. The goal of the next steps is to generate increasingly complex muscular snakes, and test whether they can approach the

reference's speed and kinematics, and whether muscle overlap naturally arises as a favorable solution.

4. **Build and optimize a muscular model consisting of only 1 muscle pair.** We start by considering the simplest possible muscular architecture in which only one muscle–tendon pair (i.e. two equal antagonistic muscle–tendon groups) is glued to the snake. Clearly, depending on where the pair is located along the body, its span and actuation, different gaits can be achieved. Then, to remove the human component from the process, we ask CMA-ES to identify for us these parameters so that the 1-muscle snake can move as fast as possible. The solution found is illustrated in the figure below. As immediately noticeable, and expected, this architecture is simply incapable of reproducing the reference gait and speed.
5. **Build and optimize a muscular model consisting of 2 muscle pairs.** Given our bottom-up approach, the next step is then to add complexity, in this case a second muscle pair, and see whether we can start approaching the reference. But where to put the second muscle? And should the first muscle pair stay where it was at step 4? Again, we let the optimizer answer these questions. By testing many (tens of thousands) different configurations, CMA-ES finds that the 2-muscle architecture that produces the fastest speed reported in the image below. As can be seen, the snake's speed improves (inset in panel b) and the gait starts resembling the reference's one (panel a). We also note that the optimizer finds on its own a solution in which muscles overlap (panel c—each line in the plot represents a muscle–tendon group pair: the line indicates the span and location across vertebrae (x-axis), while its y-position is the peak muscle force. We refer the reviewer to answer #2 in which the exploded rendering provides further visual intuition). We note that muscle-overlapping is not built into the model nor the optimization. The optimizer spanned candidate solutions with both overlapping and non-overlapping muscles and found the former design to be most effective in terms of locomotion speed. This procedure highlights the importance of using an optimization method to fairly compare different architectures according to the same standards: indeed, within the class of 1-muscle and 2-muscle architectures, we identified the configurations that produce the fastest gait, and we compare those.
6. **Progressively increase the number of muscles.** We then keep increasing the number of muscles to 3,4,5, etc. and see how the architecture evolves together with the associated speed and gait kinematics. Fig. S2 (reproduced below) illustrates this process. As can be seen, as the number of muscles increase, the solutions identified by CMA-ES tend to approach the speed and gait of the reference one, as well as closely resemble experimentally observed gaits (Fig. 3e in the main text). The effect of adding muscles becomes less and less conspicuous, and plateaus after 4 muscles, hence our choice in the manuscript.

Figure S2: Optimized designs for snakes equipped with different number of muscle pairs (increasing from left to right) detailing the (a) gait envelopes, (b) center-of-mass positions at $t = 3s$ and the corresponding average forward velocities (inset), and (c) optimal muscle arrangements and span across vertebrae as well as the peak contraction forces of these muscles.

The procedure above shows then that indeed a few simple actuators can produce fast, smooth gaits, and muscle overlapping naturally emerges as a favorable solution. Indeed, by scanning through all generated solutions, we observe that non-overlapping arrangements consistently underperform overlapping ones, with top speed degradation ranging from ~25% to ~60%.

As pointed out by the reviewer, a top-down approach is best suited when the goal is biological detail, while our bottom-up method is practical yet rigorous for demonstrating a useful engineering design principle, inspired by real snakes. While such a design motif might be intuitively clear to a biologist, it is certainly not to engineers, as chiefly exemplified by how robotic snakes have been built so far.

We updated the main text, referred to the suggested reference, and included a new section in the SI to reflect the above discussion.

Page 14, line 255:

“Here we employ our numerical approach to distill design principles and extract broadly applicable architectural motifs from complex biological systems (in this case, a snake with its intricate musculature layout), in favour of engineering manufacturability and biomechanical understanding.”

Page 14, line 265:

“Although snakes are equipped with a multitude of muscles to orchestrate a variety of gaits and body deformations, we speculate that only a few and, importantly, overlapping actuators are necessary for effective and smooth forward slithering. We test this hypothesis by considering a simplified snake architecture made of a small number of symmetric and antagonistic lateral muscle–tendon pairs. Then, we let CMA-ES identify locations and actuation patterns, so as to maximize the snake’s forward speed. This way architectural motifs are free to emerge, and their performance can be compared with reference simulations (66, 80, 81) and experimental recordings (77, 82). While previous reference studies were able to realistically replicate various gaits via continuously actuated elastic beams (66, 80, 81), we emphasize that our goal here is to reveal hidden architectural design principles and expose their functioning.”

Page 15, line 291:

“Given the above musculoskeletal model, we seek to determine the minimal number of actuators, their layout and activation patterns so as to closely reproduce reference simulation and experimental recordings.”

Page 17, line 305:

“By considering snakes of progressive complexity, equipped with 1 to 6 muscle pairs, and performing a separate optimization campaign for each of these snakes, we show in the SI that as few as four soft longitudinal actuators can closely approximate the idealized continuous reference above (66), which sets the attainable velocity upper bound. The 37-

generations optimization course of this four-muscle architecture is reported in Fig. 3b and illustrates how the average velocity converges to a maximum value that coincides with the upper bound provided by (66). Thus, a snake bearing merely four muscle groups is shown to perform comparably to the continuous actuation model.”

Page 17, line 315:

“Moreover, actuators’ overlap (Fig. 3d) is consistently identified as a key feature, both in this case (4 actuators) as well as for the 2-, 3-, 5-, 6-muscle architectures tested in the SI. Indeed, throughout the optimization, CMA-ES did generate non-overlapping candidate solutions, but these were systematically discarded as sub-optimal (up to 60% speed degradation).”

2) I still don’t understand the snake model. It is stated that it “allows us to test the functioning of overlapping longitudinal muscle”, which implies that there the model actuators are overlapping. I don’t see the overlap in Figure 3g. In fact I can’t even tell how many virtual muscles there are. On each side of the body, it appears to be three, and they are all in series. If so, why isn’t this described explicitly in the manuscript? Somewhere, I would like to see a more detailed figure that shows exactly what the layout is of the virtual muscles within the body. The same is true for the bird wing — it’s hard to see in figure 4b where exactly the connections are, and to what elements.

We thank the reviewer for bringing this to our attention. The layout geometric details were reported in Fig. 3d. There, each line represents a muscle pair span and its location across vertebrae (x-axis is the vertebrae index, while the y-axis expresses the peak contraction force generated by each muscle). As can be seen in Fig. 3d, the muscle pairs are four in number and are indeed overlapping. Nonetheless, we do acknowledge that this compact representation might not be particularly intuitive, therefore we improved it by highlighting muscle overlap (see new version of Fig. 3d in the main text), and added a new figure depicting an exploded view of the snake architecture. Given the size of this new figure (for clarity), we included it in the SI (Fig. S3, reproduced below for convenience), although we now refer to it in the main text. We hope that the new figure and its corresponding caption make the computational setup of the snake clearer.

Figure S3: (a) Full assembly of a four-muscle snake which exhibits characteristics of (b) overlapping arrangements in the muscle–tendon groups that each span 30-40 vertebrae. (c) An exploded view of the snake’s muscular architecture illustrates the muscle–tendon pairs’ position relative to the snake’s skeleton, providing visual clarity of the overlapping arrangement.

We also added the following text in the caption of Fig. 3d referring to the new figure in SI:

Page 17, Figure 3d caption:

“(d) Identified optimal muscle arrangement: muscle groups’ span (detailed exploded view provided in Fig. S3 of section 4 in the SI) and corresponding peak actuation F_m^i .”

Similarly, for understanding the connections and elements of the bird wing, we added a new figure in the SI (Fig. S4), which is reproduced below for convenience. To better visualize the connections and elements, we depict the geometry across four different views. Once again, we refer to this figure from the main text:

Page 21, Figure 4b caption:

“(b) Computational wing (right) consisting of 3171 filaments (close-up visualization from multiple perspectives provided in Fig. S4 of section 5.3 in the SI) that mimics the illustration (left) adapted from Cheng Li’s artwork (89).”

Figure S4: (a) The full assembly of a feathered wing composed of bones (humerus, ulna, radius—purple), muscles (supra-coracoideus, biceps, pectoralis, scapulo-triceps—yellow) and feathers (green). A close-up view of the wing from the (b) front, (c) bottom and (d) back view (with feathers removed to avoid visual occlusion).

3) Perhaps a slightly more minor point, but the comparison to the snake in Hu & Shelley 2012 isn't the greatest — that book chapter only provides the most superficial of analyses of the snake's movements. What is described as 'sprinting' in that chapter is not even characterized in the simplest of terms (as far as I can tell, the forward speed, amplitudes, and even snake numbers and lengths are not provided). As such, there's no way to evaluate what they mean by 'sprinting' or 'leisurely', which by the way, are not accepted terms in snake locomotion. (Sprinting is a running, used by animals with legs. How slow or fast is 'leisurely'? I naturally walk fast, and my 'leisurely' stroll might be a brisk walk for someone else.) It would be a disservice to propagate the term 'sprinting' for these reasons. This also brings up an ancillary point — Hu and Shelley's book chapter might have been more biologically informed if they had involved a biologist in the work; similarly, this current study might have been better served if a biologist had been involved. (I also advise the converse: I have seen biologists whose work delves into engineering, who don't involve engineers and whose work would be better served if they had crossed over disciplines to collaborate.)

We reported the gait kinematics of our optimized (for forward velocity) snake in comparison with those of (Hu et al., 2012) in response to an earlier comment by the reviewer: "This concern arises because for both the snake model and the wing model, the

kinematics don't look very realistic. The snake model looks to me like the roundworm C. elegans. I believe that the new approach in this manuscript could indeed produce snake-like behavior, but this isn't it."

Our intent was to show that our approach did produce kinematics comparable to observed snake kinematics. We are not in the position to assess the competences in biology of David Hu and Michael Shelley, but we have no reason to doubt their experimental recordings of midline kinematics. Nonetheless, we agree that the language used to describe the various observed gaits might have been colloquial, and not entirely rigorous. Therefore, we replaced in our manuscript every occurrence of the terms "sprinting" and "leisure", with a more factual description such as "the fastest gait observed among various snake specimens characterized by similar Froude number". The new main text then reads:

Page 18, line 324:

"For comparison, we additionally report experimentally recorded midline gaits of a corn snake, the fastest recorded in (82) among various species characterized by $Fr = 0.1$. The observed gait is found to closely resemble our models (Fig. 3e)."

Other specific comments by line:

We thank the reviewer for her/his diligent and careful reading. We address the reviewer's specific comments as follows.

Page 1 Change 'feedbacks' to 'feedback'.

We have updated the text accordingly.

Page 1 "we present here a versatile and robust numerical approach to the simulation of arbitrary musculoskeletal architectures" Delete arbitrary—if one were to use this approach, one would not choose the architectures arbitrarily.

We have updated the text accordingly.

Line 5 Change 'rigid body robots' to 'rigid-body robots'.

We have updated the text accordingly.

Line 14 "to a zoo of robotic creatures" A zoo is a specific thing, which is not a collection of robots; what is meant here is 'range'?

We replaced 'zoo' with 'range'.

Line 24 "as mechanical structures comprised of springs" Here and elsewhere throughout the manuscript, change this either to 'composed of' or 'comprising'. See here for an explanation of the difference:

<https://www.grammarly.com/blog/comprise-vs-compose/>

We thank the reviewer for pointing this out. We have updated the text accordingly.

Line 34 Remove the capital letter from ‘Finite Element Method’ and add ‘the’ at the beginning.

We have updated the text accordingly.

Line 37 Change ‘discretization elements’ distortion’ to ‘distortion of discretization elements’.

We have updated the text accordingly.

Line 42 “This represents...” Here and throughout the rest of the manuscript, the word ‘this’ should usually not be used as a noun, because it is ambiguous. Here, does the word mean paper, Euler beams, 3D lattices, or something else?

We apologize for using the word ‘this’ loosely. We meant “3D lattices of masses and Euler beams”. We have updated the text as follows.

Page 3, line 41:

“An attempt to fill the space between rigid-body and FEM models is represented by the 3D lattices of masses and Euler beams of (36), which represent a balanced compromise between accuracy, robustness and computational costs, although specialized to monolithic soft bodies.”

Line 50 There’s no need to italicize ‘assemblies’.

We have updated the text accordingly.

Line 65 “We capitalize on our previous work” I believe that ‘build’ is intended rather than ‘capitalize’, unless you are referring to making money off of your previous work.

Indeed, ‘build’ is a more suitable word in this context, and we have updated the text accordingly.

Line 82 “to study and understand in-silico biophysical functioning of natural creatures.” In silico should not be hyphenated, here and elsewhere in the manuscript. Also, the term should fall at the end in this sentence.

We have updated the text accordingly.

Line 89 “It also serves the purpose of illustrating our representation’s level of detail, which can be leveraged to address patient-specific kinesiological needs.” Because a large portion of this paper deals with non-human organisms and this is the first mention of patients, it should be noted that this refers to human patients.

We have updated the text accordingly.

Line 93 “viscoelastic Cosserat rods or filaments with each rod representing motor units (SI).” Does each rod always represent more than one motor unit? Explain more detail here.

The reviewer is right that in our human elbow joint model, each rod represents more than one motor unit. Each head of the biceps brachii is then modeled as a bundle of 18 Cosserat rods with each representing 20 motor units. This in total provides each head with 360 motor units, which is consistent with the human muscle within the estimated range for number of motor units in the biceps brachii as reported in Brown (1988). While the number of motor units is a physiological, well-defined quantity, their grouping into 18 contractile model filaments that can be autonomously recruited is a modeling approximation. The level of course-graining can be adjusted as needed and can even match all 360 physical units, although this will increase computational costs.

We now briefly report this in the main text and refer to the SI.

Page 6, line 99:

“We reproduce the biceps brachii (Fig. 1b, orange elements) in silico, each head modeled as a bundle of 18 viscoelastic Cosserat rods or filaments with each rod representing 20 motor units, for a total of 360 (hence, 720 for two heads), in agreement with average physiological measurements (63). We note here that the number of motor units per rod can be varied depending on the desired level of granularity (SI).”

Page ii, line 24 (SI):

“Each head is made of 360 motor units based on (9) and equally-distributed among $N = 18$ filaments. While the number of motor units is a physiological, well-defined quantity, their grouping into 18 contractile model filaments that can be autonomously recruited is a modeling approximation. The level of course-graining can be adjusted depending on the desired level of detail and can even match all 360 physical units, although this will increase computational costs.”

Line 94 Remove the capitalization from Sections.

We have updated the text accordingly.

Figure 1b. Change ‘Level of muscle contraction’ to ‘Level of muscle shortening’. Same in figure 3.

We have updated the figures accordingly.

Line 127 Change to ‘confirming a good match’.

We have updated the text accordingly.

Line 139 Remove the capitalization from Sections.

We have updated the text accordingly.

Line 140 What is a collision-check, and why is it hyphenated?

The collision check is implemented in our solver to ensure that the rods are not interpenetrating one another, which is an unphysical behavior. The check is implemented by measuring the local distances between filaments' centerlines and test whether these are smaller than the sum of the two filaments' local radii. The method by which interpenetration is avoided is detailed in the Methods section under "Rod collision". We have updated the text and removed the hyphen.

Page 8, line 147:

"Our model accounts for incompressibility (Poisson ratio $\nu = 0.5$) through the local dilatation factor e of Eqs. 3, 4 (Methods section—mathematical derivation can be found in (66)), and prevents rods' interpenetration by checking for collisions among them (Eq. 7)."

Line 158 Remove the capitalization from Polydimethylsiloxane.

We have updated the text accordingly.

Line 185 "The above parameter ranges are determined to account for actual manufacturability (23)." Manufacturing by the authors, in this paper? Or for future work? Distinguish. Also, why does this statement need a citation? Do we need to read that paper to understand the use of parameter ranges in manufacturing?

We set our parameter range to reflect the geometric features (for example, minimum thickness) attainable through capillary molding (a typical technique employed together with PDMS). The reference provided represents an example of the use of capillary molding in a bio-hybrid setting, and reports some of these constraints. We did not manufacture the biobot reported in that paper, and the reference is not critical to understand the meaning of ranges in manufacturing. Thus, we removed it.

Line 201 Change to 'a previous demonstration'.

We have updated the text accordingly.

Line 205 Change to '14 mm,'

We have updated the text accordingly.

Line 209 Most researchers in biomechanics report muscle contractile stress in units of MPa. Please convert.

We have updated the text accordingly.

Line 217 Change to 'leg length'.

We have updated the text accordingly.

Line 235 "to accurately capture the physics of cell- and muscle-powered soft robotic systems"

We have updated the text as follows:

Page 14, line 249:

“We have shown through these studies that our computational approach is able to capture accurately the physics of cell- and muscle-powered soft robotic systems and further optimize their design for desired performance.”

Line 240 I recommend changing ‘exploit’ to ‘use’ or ‘employ’.

We have updated the text accordingly.

Line 263 “represented by a single group—one muscle bundle of radius 6 mm spanning across 12 vertebrae and intervened between two tendons of radii 3 mm and adjustable lengths (Fig. 3a).” The usage of ‘intervened’ seems off, though I can’t put my finger on exactly why so. Perhaps it’s because the only usage I’ve seen of this in anatomical lingo is as ‘intervene’, as in the groups intervenes between two tendons. One suggestion is something like, “represented by a single group—one muscle bundle of radius 6 mm spanning across 12 vertebrae; this group intervenes between two tendons of radii 3 mm and adjustable lengths (Fig. 3a).” **Also, change ‘antagonistic muscle segments intervened between tendons’ to ‘antagonistic muscle segments that intervene between tendons’.**

We have updated the manuscript so that it now reads:

Page 15, line 279:

“In our simulation, the three major lateral muscle–tendon groups responsible for locomotion (semispinalis-spinalis (SSP-SP), longissimus dorsi (LD) and iliocostalis (IC)) are lumped into a single group—one muscle bundle of radius 6 mm spanning across 12 vertebrae that intervenes between two tendons of radii 3 mm and adjustable lengths (Fig. 3a).”

Figure 3. I don’t think it’s necessary to name Bruce Jayne in the figure and the legend. Instead, indicate that it is modified with permission from the publisher (assuming that has been done).

We have updated the figure as suggested by the reviewer, where we now removed Bruce Jayne’s name from the figure (reproduced here for convenience) and instead indicate in the caption that the image is a modified figure (with permission—requested and obtained—from the publisher) of Bruce Jayne’s illustration.

Page 16, Figure 3a caption:

“(Middle) Sketch of the snake lateral muscle anatomy (adapted and modified with permission from Bruce Jayne’s illustration (83) of the epaxial muscle segment comprised of multiple muscles and tendons).”

Line 293 “The identified optimal design exhibits muscle groups that span roughly 30–40 vertebrae (Fig. 3d). This is in reasonable agreement with biological observations where the snake’s major epaxial muscle segments in total span ~27 vertebrae (79).” I scanned through the cited paper, and can’t seem to understand where this number 27 comes from. That paper shows very clearly that there is a large amount of variation among snake species in the extent of coverage of the epaxial musculature; it would be better to simply make this statement and provide a range reported in the paper.

We apologize for the lack of clarity in the cited paper. The reviewer is right in that the cited paper (Jayne (1982)) compares snakes of different species, thereby illustrating an extensive range of epaxial musculature length. Instead, we now cite Jayne (1988), which shows a musculature comprising 27 vertebrae, consistent with the snake lateral muscle anatomy (also illustrated in Figure 3a which is redrawn from the same paper).

We updated the text as follows:

Page 17, line 313:

“This is in reasonable agreement with biological observations (83) where the snake’s major epaxial muscle segments in total span ~27 vertebrae (Fig. 3a)”

Line 301 “We additionally show experimentally recorded midline gaits” ‘Show’ seems to be an odd word choice here; it seems that the previous data are being compared to results in this study.

We apologize for the odd choice of word here, and have modified the sentence to read as follows:

Page 18, line 324:

“For comparison, we additionally report experimentally recorded midline gaits of a corn snake, the fastest recorded in (82) among various species characterized by $Fr = 0.1$. The observed gait is found to closely resemble our models (Fig. 3e).”

Line 306 “four servomotors that would exhibit a rather clunky motion.” ‘rather clunky’ is subjective; what is more precisely meant?

We agree with the reviewer that ‘clunky’ is an informal term. The message that we wish to convey is that a snake robot actuated by only four servos will generate a significantly less smooth gait. We then rephrased the sentence accordingly.

Page 18, line 328:

“This is in stark contrast with a rigid snake robot counterpart equipped with only four servomotors that would otherwise exhibit a less refined and less smooth motion.”

Line 311 “This is shown to approximate the idealized continuous actuation case, highlighting an ingenious natural solution based on overlapping longitudinal actuators.” The use of ‘ingenious’ to describe one’s own work is a bit self-congratulatory; for the authors’ own reputational sake, I’d suggest to tone this down.

We have removed the word ‘ingenious’ and replaced the sentence as follows:

Page 18, line 338:

“This is shown to approximate the idealized continuous actuation case, highlighting the role of a natural solution based on overlapping longitudinal actuators.”

Line 322 “Several studies have been conducted to understand the different biophysical aspects of bird flight” A large number studies have been conducted, not just a few.

We have updated the text as follows:

Page 19, line 349:

“Numerous studies have been conducted to understand the different biophysical aspects of bird flight, from muscular activation patterns for different flight modes (84) to geometrical and mechanical properties of feathers (85), in relation to thrust generation, drag reduction and sound suppression (86).”

Line 324 Change to ‘to thrust generation’.

We have updated the text accordingly.

Line 327 Change to ‘and model the rachis’.

We have updated the text accordingly.

Line 329 Change to ‘producing feathers of total area 28 cm², similar’.

We have updated the text accordingly.

Line 330 “is set to match area” The area of what?

This sentence refers to the feather surface area. We realize that we have mentioned it in the previous sentence and therefore, we removed it as it is redundant. The main text now reads:

Page 19, line 357:

“Each computational barb represents approximately five real ones, and its radius (1.5 mm) is set to match the estimated aggregate bending stiffness (88) (SI).”

Line 336 Change ‘dorso-ventral’ to ‘dorsoventral’, here and elsewhere.

We have updated the text and image throughout the text accordingly.

Line 380 Change to ‘feedback’.

We have updated the text accordingly.

Line 517,520 Capitalize ‘Bioinspiration’ and ‘Chrysopelea’. And the species name should be italicized. Please check the remaining citations for errors; there are errors in the supplementary references as well, which might not be caught by the journal’s editors.

We thank the reviewer for bringing this to our attention. We went over the references and have made the necessary changes.

Finally, we wish to sincerely thank the reviewer for the careful reading and numerous suggestions which helped us to significantly improve the manuscript. We believe that this is now a much stronger version than the original submission.

References used in this response:

- M. Gazzola, L. Dudte, A. McCormick, and L. Mahadevan. Forward and inverse problems in the mechanics of soft filaments. *Royal Society Open Science*, 5(6):171628, 2018.
- D. L. Hu, J. Nirody, T. Scott, and M. Shelley, “The mechanics of slithering locomotion,” *Proceedings of the National Academy of Sciences*, vol. 106, no. 25, pp. 10081–10085, 2009.
- D. L. Hu and M. Shelley. Slithering locomotion. In *Natural locomotion in fluids and on surfaces*, pages 117–135. Springer, 2012.
- H. Marvi, C. Gong, N. Gravish, H. Astley, M. Travers, J. Hatton, R.L. and Mendelson, H. Choset, D. Hu, and D. Goldman, “Sidewinding with minimal slip: Snake and robot ascent of sandy slopes,” *Science*, vol. 346, no. 6206, pp. 224–229, 2014.
- C. Wright, A. Johnson, A. Peck, Z. McCord, A. Naaktgeboren, P. Gianfortoni, M. Gonzalez-Rivero, R. Hatton, and H. Choset, “Design of a modular snake robot,” in *Intelligent Robots and Systems, 2007. IROS 2007. IEEE/RSJ International Conference on*, pp. 2609–2614, IEEE, 2007.
- W. F. Brown, M. J. Strong, and R. Snow. Methods for estimating numbers of motor units in biceps- brachialis muscles and losses of motor units with aging. *Muscle & nerve*, 11(5):423–432, 1988.
- B. C. Jayne, “Muscular mechanisms of snake locomotion: an electromyographic study of lateral undulation of the florida banded water snake (*nerodia fasciata*) and the yellow rat snake (*elaphe obsoleta*),” *Journal of Morphology*, vol. 197, no. 2, pp. 159–181, 1988
- B. C. Jayne, “Comparative morphology of the semispinalis-spinalis muscle of snakes and correlations with locomotion and constriction,” *Journal of Morphology*, vol. 172, no. 1, pp. 83–96, 1982.

Reviewers' Comments:

Reviewer #2:

Remarks to the Author:

I enjoyed reading the revised manuscript. In my opinion, the additional discussion, numerical experiments, and new observations all serve to better contextualize the argument advanced in the paper, and make the paper much stronger. I am grateful to the authors for their considerable effort in making these revisions.

Eitan Grinspun

Reviewer #3:

Remarks to the Author:

The authors have made another very large effort to improve the manuscript. I have no further major concerns, but still have a number of minor suggestions to fix. I believe that this manuscript will be an important contribution in the literature, as it should spur many new efforts to model biological systems, across disciplines and fields.

Main text:

52 It is unclear what 'all the above effects can be excited' means.

274 To further clarify this point (brought up in more depth in the response document), I'd suggest changing the end of this sentence to be: "...and expose their function for engineering purposes, rather than for explicitly testing biological hypotheses."

Figure 3d Change "Vertebrate Number" to "Vertebra number".

Figure 3 legend I still think that, for using modified figures from published sources, the convention is to not name authors explicitly. The names are listed in the References section, and so don't need to be repeated here. Also, the parenthesis should be moved from the end of the sentence to after the numeric citation.

308 Change to '37-generation'.

316 Change to 'actuators' overlap'.

Many errors remain in the References sections, both in the main document and the supplement. This issue was mentioned in the previous review. Please check carefully, as these errors can get propagated in the literature. I found potential errors in the following references, and there may be more:

[main text] 11, 23, 25, 29, 30, 46, 51, 54, 63, 64, 71, 82, 87, 88, 91

[supplement] 1, 3, 4, 7, 8, 9, 10, 11, 13, 14, 19, 20, 21, 22, 23, 28, 33, 36, 38

Supplement:

133 Change 'makings' to 'anatomy'.

134 Change 'continuously' to 'continuous'.

149 As this was not a physical model, I'd suggest changing 'glued to the snake' to 'included in the snake model'.

162 There's a missing parenthesis somewhere.

313 As mentioned previously, the comparison is most relevant to particular species, because there is so much variation in muscle morphology among snakes. So, I suggest changing:

"This is in reasonable agreement with biological observations (83) where the snake's major epaxial muscle segments in total span ~27 vertebrae."

to

"This result is in reasonable agreement with morphological observations of some species of snakes; for example, in the rat snake *Pantherophis obsoletus* (formerly *Elaphe obsoleta*), the major epaxial muscle groups span X vertebrae (83)."

Please note, I could not find the result mentioned in this sentence in this paper, but perhaps I did not look closely enough. I think it's important to get this anatomy right, and therefore suggest that the authors correspond directly with Bruce Jayne <bruce.jayne@uc.edu> about this matter. He will be able to identify the snake species and the correct number of vertebrae spanned by the muscle group.

321 Add the appropriate article to 'out-of-plane'.

330 Change to 'determines'.

349 Best to delete 'very interesting', as this is a subjective opinion.

Response to Reviewer 2:

Modeling and simulation of complex dynamic musculoskeletal architectures

Xiaotian Zhang¹, Fan Kiat Chan¹, Tejaswin Parthasarathy¹ and Mattia Gazzola^{1,2†}

¹Mechanical Science and Engineering, University of Illinois at Urbana-Champaign, Urbana, IL 61801, USA

²National Center for Supercomputing Applications, University of Illinois at Urbana-Champaign, Urbana, IL 61801, USA

† To whom correspondence should be addressed; E-mail: mgazzola@illinois.edu.

We thank the reviewer for his valuable time, consideration as well as constructive comments that led to the improved manuscript.

Response to Reviewer 3:

Modeling and simulation of complex dynamic musculoskeletal architectures

Xiaotian Zhang¹, Fan Kiat Chan¹, Tejaswin Parthasarathy¹ and Mattia Gazzola^{1,2†}

¹Mechanical Science and Engineering, University of Illinois at Urbana-Champaign, Urbana, IL 61801, USA

²National Center for Supercomputing Applications, University of Illinois at Urbana-Champaign, Urbana, IL 61801, USA

† To whom correspondence should be addressed; E-mail: mgazzola@illinois.edu.

We thank the reviewer for his valuable time, consideration and positive assessment. In the following, the comments of the reviewer are listed followed by our responses.

The authors have made another very large effort to improve the manuscript. I have no further major concerns, but still have a number of minor suggestions to fix. I believe that this manuscript will be an important contribution in the literature, as it should spur many new efforts to model biological systems, across disciplines and fields.

We are grateful to the reviewer for recognizing our efforts in improving the manuscript based on his helpful comments from the previous rounds of reviews, as well as the contribution in the modeling of biological systems this manuscript presents. We address the suggestions from the reviewer as below.

Main text:

52 It is unclear what ‘all the above effects can be excited’ means.

We rephrased the above sentence to clarify its meaning:

“In general, instead, it is not possible to rule out a priori any particular mode of deformation, especially when complex architectures interact with unstructured and dynamic environments.”

274 To further clarify this point (brought up in more depth in the response document), I’d suggest changing the end of this sentence to be: “...and expose their function for engineering purposes, rather than for explicitly testing biological hypotheses.”

We agree with the reviewer that a further clarification is needed. However, we have reworded the suggested sentence for consistency with the rest of the text and updated it as follows:

“While previous reference studies were able to realistically replicate various gaits via continuously actuated elastic beams (64, 78, 79), we emphasize that our goal here is to reveal hidden architectural design principles and expose their function for engineering purposes. This is achieved here via a generic, species-agnostic approach, rather than dissecting in detail the functioning of any specific snake architecture (Supplementary Section 4).”

Figure 3d Change “Vertebrate Number” to “Vertebra number”.

We have updated the text accordingly.

Figure 3 legend I still think that, for using modified figures from published sources, the convention is to not name authors explicitly. The names are listed in the References section, and so don’t need to be repeated here. Also, the parenthesis should be moved from the end of the sentence to after the numeric citation.

Following the reviewer’s suggestions, we have updated the caption as follows:

“(Middle) Sketch of the snake lateral muscle anatomy highlighting the epaxial muscle segment comprised of multiple muscles and tendons (adapted and modified with permission from (80))”

308 Change to ‘37-generation’.

We have updated the text accordingly.

313 As mentioned previously, the comparison is most relevant to particular species, because there is so much variation in muscle morphology among snakes. So, I suggest changing: “This is in reasonable agreement with biological observations (83) where the snake’s major epaxial muscle segments in total span ~27 vertebrae.” to “This result is in reasonable agreement with morphological observations of some species of snakes; for example, in the rat snake *Pantherophis obsoletus* (formerly *Elaphe obsoleta*), the major epaxial muscle groups span X vertebrae (83).” Please note, I could not find the result mentioned in this sentence in this paper, but perhaps I did not look closely enough. I think it’s important to get this anatomy right, and therefore suggest that the authors correspond directly with Bruce Jayne about this matter. He will be able to identify the snake species and the correct number of vertebrae spanned by the muscle group.

We agree with the reviewer that there is much variation in muscle morphology among snakes, and hence comply to the suggested changes. We draw the approximation that the muscle segments in total spans ~27 vertebrae for *Elaphe obsoleta quadrivittata* from Fig. 2 in Jayne (1988) (figure reproduced below for convenience). In the figure, the span from one end of the longissimus dorsi (LD) to the other end of anterior tendon (AT) is ~27 vertebrae. Since in our simplified muscular snake model we lump the major muscle–tendon groups (as illustrated in the figure below) into one, we considered the entire span of these groups in order to correlate and compare with the resulting muscle–tendon group length obtained from the optimization campaign.

Fig. 2. Simplified right lateral view of the major epaxial muscle segments of *Elaphe obsoleta quadrivittata*. Anterior is to the right. SP and SSP, respectively, indicate the spinalis and semispinalis portions of the M. semispinalis-spinalis, and AT is the anterior tendon of

the SSP-SP. LD represents the M. longissimus dorsi, and MT, TA, and LT are the medial tendon, tendinous arch, and lateral tendon of LD. MIC and LIC, respectively, are the medial and lateral heads of the M. iliocostalis, and IT is its intermediate tendon.

316 Change to ‘actuators’ overlap’.

We have updated the text accordingly.

Many errors remain in the References sections, both in the main document and the supplement. This issue was mentioned in the previous review. Please check carefully, as these errors can get propagated in the literature. I found potential errors in the following references, and there may be more:

[main text] 11, 23, 25, 29, 30, 46, 51, 54, 63, 64, 71, 82, 87, 88, 91

[supplement] 1, 3, 4, 7, 8, 9, 10, 11, 13, 14, 19, 20, 21, 22, 23, 28, 33, 36, 38

We originally thought that the errors mentioned by the reviewer referred to the chapter/pages numbers and cross-checked all our references with Google Scholar and Isi Web of Knowledge. We then realized that the issue pointed out by the reviewer relates to having initial capitals for journal/book titles (for example *Physical review letters* instead of the correct version: *Physical Review Letters*), as well as volume numbers with the subsequent comma that appears in bold.

We thank the reviewer for pointing this out and have implemented the relevant changes accordingly. The list of references pointed out by the reviewer now reads:

Main text:

[11] Onal, C. D. & Rus, D. Autonomous undulatory serpentine locomotion utilizing body dynamics of a fluidic soft robot. *Bioinspiration and Biomimetics* **8**, 026003 (2013).

[23] Williams, B., Anand, S., Rajagopalan, J. & Saif, M. A self-propelled biohybrid swimmer at low reynolds number. *Nature Communications* **5**, 3081 (2014).

[25] Jawed, M. K., Khouri, N. K., Da, F., Grinspun, E. & Reis, P. M. Propulsion and instability of a flexible helical rod rotating in a viscous fluid. *Physical Review Letters* **115**, 168101 (2015).

[29] Winter, D. A. *Biomechanics and motor control of human movement Ch. 9* (John Wiley & Sons, 2009).

[30] Delp, S. L. *et al.* Opensim: open-source software to create and analyze dynamic simulations of movement. *IEEE Transactions on Biomedical Engineering* **54**, 1940–1950 (2007).

[46] Kirchhoff, G. Ueber das gleichgewicht und die bewegung eines unendlich dünnen elastischen stabes. *Journal für die reine und angewandte Mathematik* **56**, 285–313 (1859).

[51] Durville, D. Numerical simulation of entangled materials mechanical properties. *Journal of Materials Science* **40**, 5941–5948 (2005).

[54] Yang, Y., Tobias, I. & Olson, W. Finite element analysis of dna supercoiling. *The Journal of Chemical Physics* **98**, 1673–1686 (1993).

- [63] Brown, W. F., Strong, M. J. & Snow, R. Methods for estimating numbers of motor units in biceps-brachialis muscles and losses of motor units with aging. *Muscle & Nerve* **11**, 423–432 (1988).
- [64] Henneman, E. Organization of the spinal cord. *Medical Physiology* **12**, 1717–32 (1968).
- [71] Hansen, N. & Ros, R. Benchmarking a weighted negative covariance matrix update on the bbob-2010 noiseless testbed. In *Proceedings of the 12th Annual Conference Companion on Genetic and Evolutionary Computation*, 1673–1680 (ACM, 2010).
- [82] Hu, D. L. & Shelley, M. Slithering locomotion. In *Natural Locomotion in Fluids and on Surfaces*, 117–135 (Springer, 2012).
- [88] Sullivan, T. N. *et al.* A lightweight, biological structure with tailored stiffness: The feather vane. *Acta Biomaterialia* **41**, 27–39 (2016).
- [91] Antman, S. *The theory of rods* (Springer, 1973).

Supplementary Information:

- [1] Winter, D. A. *Biomechanics and motor control of human movement Ch. 9* (John Wiley & Sons, 2009).
- [3] Cofer, D. *et al.* Animatlab: a 3d graphics environment for neuromechanical simulations. *Journal of Neuroscience Methods* **187**, 280–288 (2010).
- [7] Conwit, R. *et al.* The relationship of motor unit size, firing rate and force. *Clinical Neurophysiology* **110**, 1270–1275 (1999).
- [8] Milner-Brown, H., Stein, R. & Yemm, R. The orderly recruitment of human motor units during voluntary isometric contractions. *The Journal of Physiology* **230**, 359–370 (1973).
- [9] Brown, W. F., Strong, M. J. & Snow, R. Methods for estimating numbers of motor units in biceps-brachialis muscles and losses of motor units with aging. *Muscle & Nerve* **11**, 423–432 (1988).
- [10] Klein, C. S., Marsh, G. D., Petrella, R. J. & Rice, C. L. Muscle fiber number in the biceps brachii muscle of young and old men. *Muscle & Nerve* **28**, 62–68 (2003).
- [19] Cox, R. The motion of long slender bodies in a viscous fluid part 1. general theory. *Journal of Fluid Mechanics* **44**, 791–810 (1970).
- [20] Yanoviak, S. P., Munk, Y., Kaspari, M. & Dudley, R. Aerial manoeuvrability in wingless gliding ants (*Cephalotes atratus*). *Proceedings of the Royal Society B: Biological Sciences* **277**, 2199–2204 (2010).
- [22] Bachmann, T. *et al.* Morphometric characterisation of wing feathers of the barn owl *Tyto alba pratincola* and the pigeon *Columba livia*. *Frontiers in Zoology* **4**, 23 (2007).
- [23] Sullivan, T. N. *et al.* A lightweight, biological structure with tailored stiffness: The feather vane. *Acta Biomaterialia* **41**, 27–39 (2016).
- [28] Williams, B., Anand, S., Rajagopalan, J. & Saif, M. A self-propelled biohybrid swimmer at low reynolds number. *Nature Communications* **5**, 3081 (2014).

[33] Lentink, D., Jongerius, S. R. & Bradshaw, N. L. The scalable design of flapping micro-air vehicles inspired by insect flight. In *Flying Insects and Robots*, 185–205 (Springer, 2009).

[36] Du, Y., Plante, E., Janicki, J. S. & Brower, G. L. Temporal evaluation of cardiac myocyte hypertrophy and hyperplasia in male rats secondary to chronic volume overload. *The American Journal of Pathology* **177**, 1155–1163 (2010).

[38] Jacot, J. G., Martin, J. C. & Hunt, D. L. Mechanobiology of cardiomyocyte development. *Journal of Biomechanics* **43**, 93–98 (2010).

Supplement:

133 Change ‘makings’ to ‘anatomy’.

We have updated the text accordingly.

134 Change ‘continuously’ to ‘continuous’.

We have updated the text accordingly.

149 As this was not a physical model, I’d suggest changing ‘glued to the snake’ to ‘included in the snake model’.

We have updated the text accordingly.

162 There’s a missing parenthesis somewhere.

We have fixed the error accordingly.

321 Add the appropriate article to ‘out-of-plane’.

We have fixed the error accordingly.

330 Change to ‘determines’.

We have fixed the error accordingly.

349 Best to delete ‘very interesting’, as this is a subjective opinion.

We have fixed the error accordingly.

Again, we thank the reviewer for his careful comments, that overall lead us to significantly improve the manuscript.